# HIERARCHICAL PERIODIC STATIONARIZATION FOR NON-STATIONARY TIME SERIES FORECASTING

## ABSTRACT

Time series forecasting (TSF) has advanced rapidly through benchmark-driven competition. However, we find that state-of-the-art models struggle to predict even a simple long-period sine wave, despite ample training data. One reason is that existing benchmarks underrepresent the non-stationary characteristics prevalent in real-world time series, leading to misleading evaluations. Moreover, standard stationarization methods inherently introduce substantial information loss during the stationarization process. To investigate this, we introduce *controlled* datasets that expose information loss incurred by standard z-normalization-based stationarization methods, widely used in TSF models. To address this limitation, we propose Hipeen, a hierarchical periodic stationarization method that achieves stationarization through representing the value into multiple periodic components, minimizing information loss. Hipeen, with a linear backbone, successfully forecasts highly non-stationary signals— *controlled* datasets and large-scale stock datasets—substantially outperforming current SOTA models (8 stationarization methods and 8 baselines), while maintaining strong performance on conventional benchmarks. Our results highlight the importance of preserving critical information during stationarization and provide a new approach for robust TSF in non-stationary environments. All code and models will be released in the final version.

## 1 INTRODUCTION

Time series forecasting (TSF) has advanced rapidly through benchmark-driven competition on datasets designed to represent real-world signals (Wu et al., 2023). Yet, our analysis reveals that even the latest state-of-the-art (SOTA) models perform unexpectedly poorly on a seemingly simple case: forecasting a long-period sine wave with Gaussian noise (Figure 1A), despite the ample training data covering multiple full cycles. This raises two natural questions: Why do benchmark-leading models fail on such simple signals, and do current benchmarks adequately reflect real-world time series?

To address these questions, we examine the **stationarity** in time series. Changes in a data's distribution over time—known as **distribution shift** or **non-stationarity**—cause train and test distributions to diverge, reducing model performance (Li et al., 2023). Fan et al. (2023) further demonstrated that non-

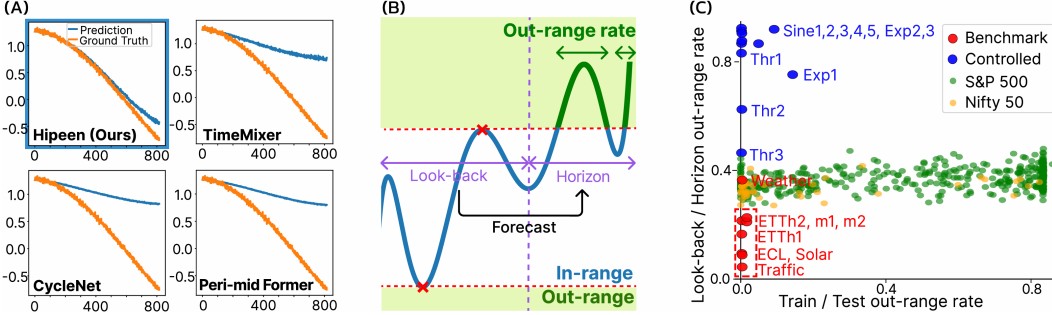

Figure 1: **(A)** Latest SOTA models, including Timemixer (Wang et al., 2024b), CycleNet (Lin et al., 2024), and Peri-midformer (Wu et al., 2024), fail on the long-range sine wave forecasting. **(B)** The out-range rate is shown as an intuitive proxy for the degree of non-stationarity. **(C)** Four types of datasets are positioned according to their Train/Test and Look-back/Horizon out-range rates.

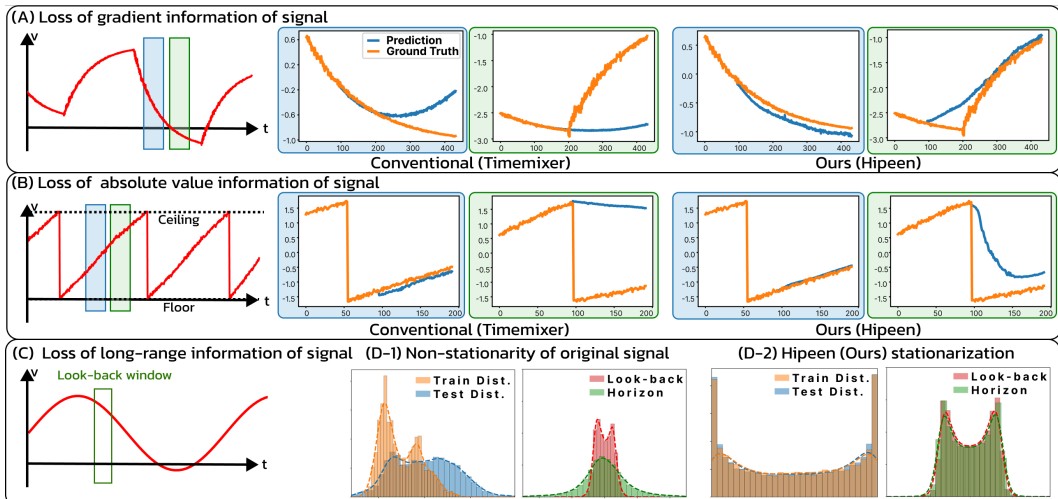

Figure 2: **(A)** Visualization of the Exponential task. At each look-back position (blue, green), the right figures show how the TimeMixer and Hipeen(Ours) forecast the following signal. **(B)** Visualization of the Threshold task. same as (A). **(C)** Visualization of the Sine wave task. **(D)** Hipeen stationarizes not only across train-test splits but also within each sample between look-back and horizon.

stationarity within a sample—between look-back and horizon windows—also impairs performance. In line with this research, we introduce the "**out-range rate**" as an intuitive proxy to quantify the degree of non-stationarity in a dataset (Figure 1B). This metric measures the percentage of values in a sequence B that fall outside the $[\min(A), \max(A)]$ range of another sequence A.

Figure 1C reveals a stark contrast between TSF benchmarks (red) and the long-periodic sine wave (blue; Sine1). In this figure, the x and y-axis represent the train/test and look-back/horizon out-range rate, respectively, mapping the space where all time series data can be positioned. While the Sine1 and real-world stock datasets (*S&P500* and *Nifty50*) span broader regions, the benchmarks are clustered narrowly around the origin. This suggests that current benchmarks underrepresent non-stationary real-world time series, making it plausible that models optimized for these benchmarks would fail to predict even a simple long-periodic sine wave that exhibits high look-back/horizon non-stationarity.

This leads to further questions. Are current SOTA models incapable of handling non-stationarity? And why does their performance falter on non-stationary data? The first question can be answered in the negative. As will be detailed in *related works*, stationarization methods such as RevIN (Kim et al., 2021), Dish-TS (Fan et al., 2023), and SAN (Liu et al., 2024b) employ z-normalization to align distributions effectively, yielding low out-range rates after processing. Indeed, most SOTA models incorporate RevIN as a default component Wang et al. (2024b), ensuring that even highly non-stationary signals are supplied in a stationarized form. Therefore, in response to the second question, we argue that **the critical issue lies not in how well stationarization aligns distributions, but in the extent of information loss it introduces.**

To substantiate our claim, we introduce a *controlled* **dataset** where forecasting requires information—gradients or absolute values—that z-normalization discards. First, the **Exponential (Exp.)** dataset contains exponential functions that flip when reaching a specific gradient (Figure 2A). Second, the **Threshold (Thr.)** dataset involves a strictly increasing function whose slope lies within a prescribed range and resets to zero upon reaching a predetermined threshold (Figure 2B). Finally, the **Sine wave (Sine)** requires both absolute value and gradient information to ascertain its current position within the long-range pattern (Figure 2C). As can be seen in Figure 1C, these datasets exhibit substantially higher look-back/horizon out-range rates compared to the benchmark. Our experiments show that the latest SOTA models (as well as older models that do not use z-normalization) all fail to predict these *controlled* datasets, thereby confirming that the information essential for forecasting—specifically gradients and absolute values—is indeed lost in practice.

Many real-world systems, such as battery charging or HVAC systems, rely on gradient or threshold dynamics, making their loss during stationarization problematic. Furthermore, as demonstrated with the sine wave, current models perform poorly in identifying long-range periodic patterns. To address

these shortcomings, we propose a novel **Hierarchical Periodic Ensemble (Hipeen)** stationarization method, which does not rely on z-normalization and thus mitigates the loss of essential information. Analogous to representing a single real number as multiple digits in a decimal expansion, Hipeen performs stationarization by projecting a signal's value into multiple hierarchical periodic components, transforming non-stationary value variations into stationary, fixed-range periodic motions, achieving high stationarity (Figure 2D). Remarkably, Hipeen, when paired with a simple linear backbone, is the sole method to succeed in forecasting our *controlled* dataset.

We further extend our experiments to a broad real-world stock datasets characterized by simultaneously high look-back/horizon and train/test out-range rates. We show that Hipeen, with only a linear backbone, outperforms current SOTA models on these datasets, clearly demonstrating both the limitations of existing stationarization approaches and the effectiveness of Hipeen on real-world datasets. Finally, despite being designed to address pronounced non-stationarity, Hipeen also demonstrates more favorable performance compared to other stationarization methods on the stationary benchmark dataset. To sum up, Hipeen is the first method capable of processing highly non-stationary signals without significant information loss, paving the way for future advancements in stationarization. It achieves state-of-the-art performance on non-stationary signals (both *controlled* and stock datasets) while also demonstrating robust capabilities on the stationary benchmarks.

In summary, our main contributions are as follows:

- Revealing the benchmark gap: We show that widely used TSF benchmarks underrepresent non-stationary characteristics found in simple signals (e.g., long-period sine wave) and real-world data (e.g., stocks), explaining why existing SOTA models fail on such tasks.
- Controlled datasets for analysis: We introduce new *controlled* datasets (Exponential, Threshold, and Sine wave) that isolate gradient and absolute-value information. These datasets expose information loss in current stationarization approaches.
- Hipeen method: It minimizes information loss while achieving high stationarity. Hipeen, even with a linear backbone, outperforms SOTA models on both highly non-stationary datasets (*controlled* and stock) and is comparable on the standard stationary benchmarks.

## 2 RELATED WORKS

**Addressing non-stationarity in TSF models.** Real-world time series are often non-stationary, with distribution shifts over time due to changing environments, hindering their predictability (Wu et al., 2023; Kim et al., 2025a; 2021). While early methods rely on domain adaptation (e.g., DDG-DA (Li et al., 2022)) or distribution matching (e.g., AdaRNN (Du et al., 2021)), the most widely used approach today is to apply normalization and de-normalization around the forecaster. The pivotal method, RevIN (Kim et al., 2021), applies instance-wise normalization by removing the time-domain mean and variance, then restoring them after forecasting. This line of research evolved to handle distribution shifts more dynamically: Dish-TS (Fan et al., 2023) predicts future statistics, while SAN (Liu et al., 2024b) introduced slice-level normalization to capture local distributional changes.

Recognizing the limitations of purely time-domain statistics, the latest approaches leverage the frequency domain. FAN (Ye et al., 2024) employs the Fourier transform to identify and normalize instance-wise dominant frequency components, explicitly modeling evolving trends and seasonalities. Similarly, DDN (Dai et al., 2024) utilizes wavelet transforms to dynamically capture and normalize multi-scale non-stationary factors in both the time and frequency domains.

Although these frameworks are widely adopted across SOTA TSF and foundation models (Wang et al., 2024a;b; Das et al., 2024; Goswami et al., 2024), they all share a fundamental limitation: normalization discards critical information. Specifically, the original signal's absolute magnitude, gradient, and higher-order statistics are lost in the process of achieving stationarity. Other attempts to bypass this, such as NST (Liu et al., 2022b) incorporating non-stationary dynamics into its architecture or DLinear (Zeng et al., 2023) using the raw signal. However, these methods either still depend on the lossy statistics or lack robust mechanisms for raw signal. Our approach, Hipeen, is fundamentally different in that it achieves stationarity representationally—not through normalization—by projecting values into a hierarchical periodic space. This process preserves the critical absolute value and gradient information that normalization-based methods inherently discard. For a detailed discussion on recent TSF models, please refer to Appendix A.

## 3 METHODS

**Problem Statement.** We follow the standard multivariate TSF formulation (Wu et al., 2023; Liu et al., 2024a). At time $t$, the length $L$ look-back window $\boldsymbol{X}_t = \{\boldsymbol{x}_{t-L+1}, \cdots, \boldsymbol{x}_t\} \in \mathbb{R}^{L \times N}$ is given to predict consecutive length $K$ horizon $\boldsymbol{Y}_t = \{\boldsymbol{x}_{t+1}, \cdots, \boldsymbol{x}_{t+K}\} \in \mathbb{R}^{K \times N}$, where $N$ denotes the number of channels. Section 3.1 describes how Hipeen transforms $\boldsymbol{X}_t$ and $\boldsymbol{Y}_t$ into projections, and conducts training in this projection space. Section 3.2 explains how projections are converted back to signal values via a loss-minimizing estimator during inference.

**Motivation behind Hipeen (Conceptual).** First, "periodicity" in Hipeen is not about the signal's repeating patterns over time (temporal periodicity), but about embedding value into periodic digit-based representation. Therefore, this is a concept entirely different from approaches that leverage the temporal periodicity of time series (e.g. DDN (Dai et al., 2024), CycleNet (Lin et al., 2024)).

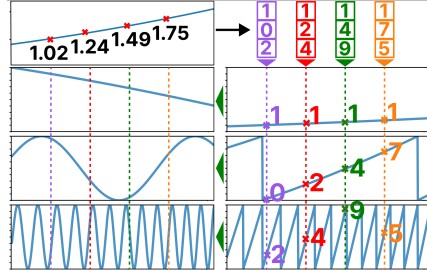

Figure 3: Conceptual visualization of Hipeen: representing each value as digits projects a simple increasing function into diverse periodic patterns.

Hipeen is a function that converts a scalar into a vector by decomposing its decimal digits; for example, 1.6712 becomes [1,6,7,1,2]. This allows stationarization without the information loss associated with normalization.

Stationarity is achieved as follows: For low-order digits, even small changes in the original value cause rapid fluctuations, with digits 0-9 appearing at a uniform frequency, thus achieving high stationarity. For the high-order digits, they naturally remain stationary for a long period. For the middle digits, we add random angular bias to achieve stationarity.

Consider a signal with a long-range pattern beyond the look-back window. With a small window, you'd only observe a non-periodic segment of the signal (Figure 3). Hipeen addresses this by decomposing a simple monotonic value change into multiple hierarchical periodic signals. The lower-order digits undergo multiple periodic cycles (as the digit wraps around 0 to 9 multiple times) with small changes in original value, thereby encoding fine-grained gradient variations through frequency changes. Also, high-order digits capture global trends and absolute values of the signal. These hierarchical projections serve as multiple views of a single value, effectively forming an ensemble.

*Technical note*: In reality, Hipeen follows a binary representation with hierarchical radii based on powers of 2. And the transformation is not a simple quantified split, but rather something analogous to: $1.6712... \rightarrow [0.167, 0.671, 0.712, 0.12, ...]$

### 3.1 HIPEEN PROJECTION

Hipeen replaces traditional normalization-based stationarization (Kim et al., 2021; Fan et al., 2023; Liu et al., 2024b)—which typically loses the original signal's mean and variance information—by projecting the input values into multiple periodic components organized in a hierarchical structure.

Figure 4(A) illustrates the schematic process of the Hipeen projection, where a raw value is mapped as $V \in \mathbb{R} \rightarrow \boldsymbol{\theta} \in [0, 2\pi)^H \rightarrow \boldsymbol{P} \in [-1, 1]^{2H}$. Here, $V$ denotes a real-valued scalar, $\boldsymbol{\theta} = (\theta_1, \ldots, \theta_H)$ denotes its $H$-dimensional angular representation ($H$=number of hierarchy levels). Each angle $\theta_h$ is then expressed as its sine–cosine pair, thereby producing the projection vector $\boldsymbol{P} = (\sin\theta_1, \cos\theta_1, \ldots, \sin\theta_H, \cos\theta_H) \in [-1, 1]^{2H}$.

Specifically, the Hipeen projection is defined by three components: the scale parameter $M \in \mathbb{R}$, the number of hierarchy levels $H$, and a bias matrix $\boldsymbol{B} \in [0, 2\pi)^{N \times H}$ sampled from the uniform distribution $\mathcal{U}(0, 2\pi)$. These components are fixed before training. For each hierarchy level $h \in \{1, \ldots, H\}$, we set the radius as $r_h = M \cdot 2^h$. This exponential growth of radii allows the projection to capture both fine-scale and large-scale variations of the signal simultaneously, providing a multi-resolution view of the input. Let $V$ be the value from the $n$-th channel at a particular time step. Its angular representation at hierarchy level $h$ is obtained as follows:

$$\theta_h = \left(\frac{V}{r_h} + B_{n,h}\right) \bmod 2\pi, \quad \boldsymbol{\theta} = (\theta_1, \ldots, \theta_H) \in [0, 2\pi)^H, \tag{1}$$

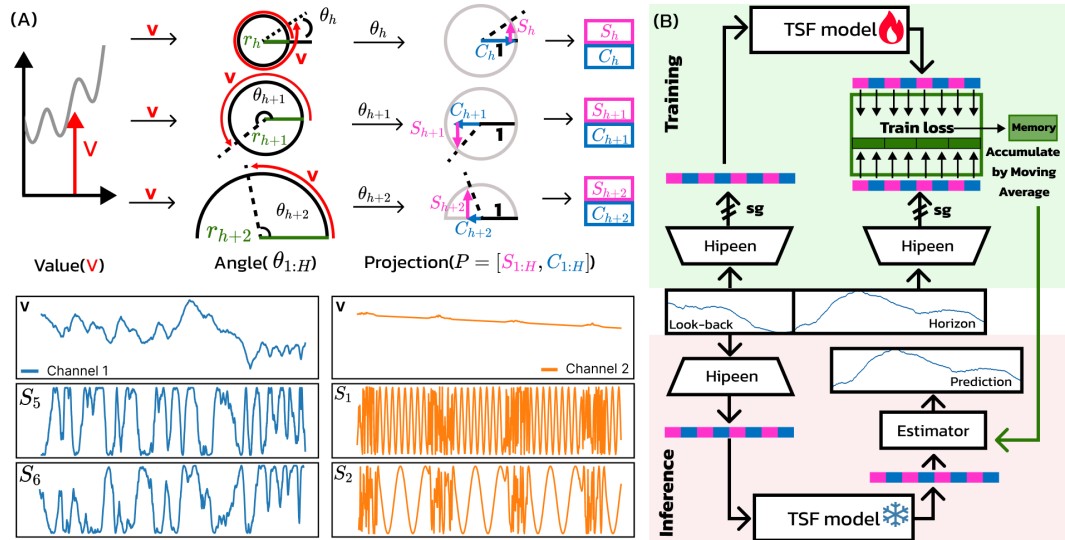

Figure 4: (A) Top: Time series value is converted into multiple periodic angles $\theta$ with exponentially increasing $r$, then into sine $S$ and cosine $C$. Bottom: Example of this transformation on weather data ($V$ to $S$). (B) Overview of the model's training and inference process (sg: stop gradient).

where $B_{n,h}$ denotes the $(n,h)$-th entry of the bias matrix $\boldsymbol{B}$.

This angular representation $\theta_h$ effectively transforms unbounded real values into periodic coordinates. Each $\theta_h$ is converted into a sine–cosine pair, forming the Hipeen projection vector $\boldsymbol{P}$.

$$\boldsymbol{P}_{[2h:2h+1]} = [\sin(\theta_h), \cos(\theta_h)], \quad \boldsymbol{P} \in [-1, 1]^{2H}. \tag{2}$$

As a result, the Hipeen projection is a $2H$-dimensional bounded vector $\boldsymbol{P}$ for each scalar input $V$. This transformation resolves the discontinuity at $0$ and $2\pi$ of the angular representation. It preserves both the continuity and differentiability properties of the original time series.

Moreover, since the projection involves no learnable parameters, it is computationally efficient and can be seamlessly integrated into any TSF model architecture, making it inherently model-agnostic.

***Training phase.*** Since the reverse mapping of the Hipeen projection does not admit a closed-form solution, training is performed in the projection space (Figure 4(B)). To this end, both the look-back $\boldsymbol{X} \in \mathbb{R}^{L \times N}$ and the horizon $\boldsymbol{Y} \in \mathbb{R}^{K \times N}$ are projected using Hipeen, resulting in $\boldsymbol{X}_{\text{hip}} \in [-1, 1]^{L \times N \times 2H}$ and $\boldsymbol{Y}_{\text{hip}} \in [-1, 1]^{K \times N \times 2H}$. For notational simplicity, we omit the time index $t$ in both $\boldsymbol{X}$ and $\boldsymbol{Y}$. The projection dimensions can be interpreted as channels with strong interdependencies, and the backbone TSF model $f(\cdot)$ learns to map $\boldsymbol{X}_{\text{hip}}$ to $\boldsymbol{Y}_{\text{hip}}$.

To train the model in the projection space, we define the loss between the prediction $\hat{\boldsymbol{Y}}_{\text{hip}} := f(\boldsymbol{X}_{\text{hip}})$ and the target $\boldsymbol{Y}_{\text{hip}}$. To capture hierarchical periodicity, we optimize each of the $H$ $(\sin, \cos)$ pairs independently with cosine distance, rather than all $2H$ dimensions jointly:

$$\mathcal{L} = \frac{1}{KNH} \sum_{k=1}^{K} \sum_{n=1}^{N} \sum_{h=1}^{H} 2 \cdot d_{\cos}\left(\hat{Y}_{\text{hip}[2h:2h+1]}^{k,n}, Y_{\text{hip}[2h:2h+1]}^{k,n}\right), \tag{3}$$

where $d_{\cos}(a, b) = 1 - \frac{a \cdot b}{\|a\| \|b\|}$ denotes the cosine distance, and $\boldsymbol{Y}_{\text{hip}[2h:2h+1]}^{k,n}$ denotes the sine–cosine pair of the $n$-th channel at horizon step $k$ and level $h$, with $\hat{Y}_{\text{hip}[2h:2h+1]}^{k,n}$ its prediction. This ensures that each sub-period is aligned in phase, effectively capturing hierarchical periodicity.

Since $\cos(\theta)$ approximates $1 - 0.5 \cdot \theta^2$ when $\theta$ is small, minimizing the loss is equivalent to minimizing the squared angular difference. A loss before averaging: $\boldsymbol{Q} \in \mathbb{R}^{K \times N \times H}$ is maintained in memory for the estimation phase. This tensor is progressively updated throughout training via exponential moving averaging (EMA). We fixed the smoothing factor of the EMA to 0.005 for all experiments.

***Inference phase.*** The model prediction $\hat{\boldsymbol{Y}}_{\text{hip}}$ in the Hipeen projection space is transformed back to the original space $\hat{\boldsymbol{Y}} \in \mathbb{R}^{K \times N}$ using the Hipeen estimator, described in the following section.

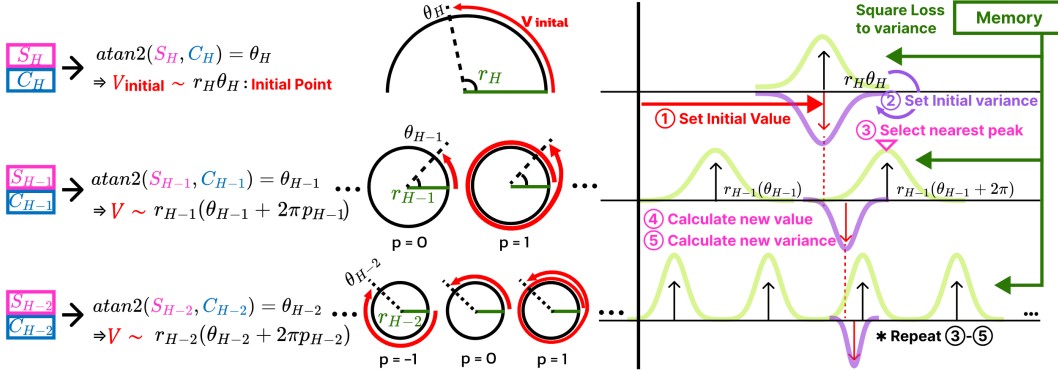

Figure 5: The Hipeen estimator sequentially ensembles projections of various periods along the $H$ dimension. It calculates the number of full rotations ($2\pi p$) to add to $\theta$ based on the previous $V_{\text{est}}$, and calculates $V_h$ with $p$. Then updates $V_{\text{est}}$ to have minimal variance utilizing the $V_h$ and stored loss.

## 3.2 HIPEEN ESTIMATOR

The initial reverse mapping from $\boldsymbol{P}$ to the $\boldsymbol{\theta}$ can be efficiently computed using the two-argument arctangent function, atan2, which preserves quadrant information.

$$\boldsymbol{P} \to \boldsymbol{\theta}: \qquad \theta_h = \text{atan2}(P_{2h}, P_{2h+1}), \qquad \boldsymbol{\theta} \in [0, 2\pi)^H. \qquad (4)$$

However, analytically retrieving the most probable value from a vector of angles ($\boldsymbol{\theta} \to V$) requires solving a degree-$H$ polynomial equation, which is intractable. To address this, we leverage the hierarchical structure of Hipeen and perform a chain of estimations to progressively reconstruct the final value $V_{\text{est}}$, as illustrated in Figure 5. This hierarchical estimation procedure, the Hipeen estimator, performs inverse mapping from $\boldsymbol{\theta}$ to $V$ during inference with $O(H)$ computational complexity.

$\boldsymbol{\theta} \to V$ estimation starts from the assumption that the absolute value of $V_{\text{est}}$ is less than $\pi \cdot r_H$. Since the data is normalized with training data statistics (Wu et al., 2023), and $r_H$ increases exponentially with $H$, the assumption holds with a reasonable choice of $H$. An initial estimate $V_{\text{est}}$ is computed from $\theta_H$, mapping $\theta_H - B_{n,H}$ to $[-\pi, \pi]$. And initial variance $v_{\text{est}}$ comes from the $Q_H$.

$$\text{Init}: \qquad r_H \cdot ((\theta_H - B_{n,H} + \pi) \bmod 2\pi - \pi) \to V_{\text{est}}, \qquad Q_H \cdot (r_H)^2 \to v_{\text{est}}. \qquad (5)$$

The angular squared loss $Q_H$ is scaled by the squared radius to reflect variance in the length. Subsequently, $V_{\text{est}}$ is iteratively refined descending through the $H$ dimension. The challenge with smaller radii $r$ lies in the ambiguity of how many full rotations ($2\pi p$) are missing in the angle $\theta$. Therefore, we first determine the number of cycles $p$ that makes $V_h$ closest to $V_{\text{est}}$ (step 3 in Figure 5).

$$\text{Calculate } p: \qquad p_h = \text{round}((1/2\pi) \times (V_{\text{est}}/r_h - (\theta_h - B_{n,H}))). \qquad (6)$$

Then, based on $p$, the new $V_h$ is calculated. To minimize the variance of $V_{\text{est}}$, we apply inverse-variance weighting to compute a weighted average of the observations. The corresponding variance estimate, $v_{\text{est}}$, is updated accordingly (step 4,5 in Figure 5).

$$\text{Update } V_{\text{est}}, v_{\text{est}}: \qquad r_h((2\pi p_h + \theta_h - B_{n,h}) \to V_h, \qquad Q_h \cdot (r_h)^2 \to v_h, \qquad (7)$$

$$(V_{\text{est}} * v_h^n + V_h * v_{\text{est}})/(v_{\text{est}} + v_h) \to V_{\text{est}}, \quad (v_{\text{est}} * v_h)/(v_{\text{est}} + v_h) \to v_{\text{est}}. \qquad (8)$$

The final estimate $V_{\text{est}}$ is obtained by iteratively applying Equations (7)–(9), offering a simple yet accurate method for estimating $V$. Computation takes less than 1ms/step in real-world practice, making it negligible. Refer to Appendix C.1 for further details on the Hipeen projection and estimator.

**Backbone TSF model** is a linear architecture, deliberately chosen to isolate and highlight the effectiveness of the Hipeen stationarization, excluding improvements that could arise from architectural advancements. Convolutional layers without non-linear activations were used to minimize the number of learnable parameters. To enhance the expressiveness of Hipeen under this linear mapping constraint, we introduce an extra ensemble that generates multiple Hipeen projections per sample, offering diverse views. This is achieved by multiplying a scaling factor $W \sim \mathcal{U}(0.5, 1.5)$ to the radius $r$, resulting in period-adjusted windows. All extra ensemble views share the same backbone model, and no additional parameters are introduced. Moreover, these extra ensemble dimensions are merged into the batch dimension, allowing efficient parallel computation. For more details on the backbone architecture and extra ensemble, please refer to Appendix C.2.

Table 1: Results on the three *controlled* datasets. 16 recent baseline models were compared with Hipeen. We report the average performance across four forecasting horizons {96, 192, 336, 720} and three random seeds. The best results are highlighted in red and the second-best in blue. The extended table and standard deviation results are provided in Appendix E.1.

| | Models | | Exponential | | | | Threshold | | | | Sine wave | | | | | |
|---|---|---|---|---|---|---|---|---|---|---|---|---|---|---|---|---|
| | | | 300-350 | 400-450 | 500-550 | Rank | 5-20(e-4) | 10-40(e-4) | 15-60(e-4) | Rank | 2k-3k | 3k-4k | 4k-5k | 5k-6k | 6k-7k | Rank |
| Stationarization methods | NST (2022b) | MSE | 0.566 | 0.372 | 0.802 | 8.0 | 1.412 | 2.177 | 1.713 | 14.7 | .1291 | .0272 | .0183 | .0039 | .0071 | 5.4 |
| | | MAE | 0.505 | 0.376 | 0.529 | 7.7 | 0.715 | 1.029 | 0.956 | 11.3 | .1908 | .0862 | .0703 | .0413 | .0526 | 6.8 |
| | DLinear (2023) | MSE | 1.327 | 0.631 | 0.627 | 13.7 | 0.689 | 0.736 | 0.737 | 2.6 | .1586 | .1032 | .0859 | .0425 | .0374 | 13.2 |
| | | MAE | 0.710 | 0.535 | 0.534 | 14.0 | 0.608 | 0.664 | 0.680 | 2.6 | .2550 | .1922 | .1911 | .1338 | .1288 | 13.6 |
| | RLinear (2023) | MSE | 0.633 | 0.571 | 0.526 | 10.3 | 1.317 | 1.419 | 1.216 | 9.3 | .1076 | .0429 | .0166 | .0135 | .0072 | 7.2 |
| | | MAE | 0.526 | 0.460 | 0.404 | 9.7 | 0.774 | 0.897 | 0.836 | 9.3 | .1619 | .1047 | .0678 | .0655 | .0499 | 7.8 |
| | Dish-TS (2023) | MSE | 2.146 | 1.463 | 0.660 | 14.7 | 0.795 | 1.016 | 0.936 | 3.0 | .5572 | 1.499 | 2.632 | .2635 | .2596 | 15.0 |
| | | MAE | 1.120 | 0.861 | 0.586 | 15.0 | 0.672 | 0.800 | 0.801 | 3.0 | .5004 | .7361 | 1.114 | .3819 | .3314 | 15.0 |
| | SAN (2024b) | MSE | 0.482 | 0.425 | 0.392 | 2.7 | 1.285 | 1.273 | 1.134 | 4.0 | .1000 | .0363 | .0118 | .0079 | .0045 | 3.6 |
| | | MAE | 0.481 | 0.429 | 0.387 | 5.0 | 0.824 | 0.912 | 0.849 | 13.0 | .1468 | .0913 | .0561 | .0506 | .0413 | 3.4 |
| | Leddam (2024) | MSE | 0.474 | 0.493 | 0.482 | 3.0 | 1.256 | 1.458 | 1.272 | 11.7 | .0774 | .0359 | .0162 | .0134 | .0069 | 4.4 |
| | | MAE | 0.462 | 0.428 | 0.390 | 4.0 | 0.796 | 0.958 | 0.895 | 15.3 | .1350 | .0915 | .0655 | .0642 | .0481 | 4.6 |
| | DDN (2024) | MSE | 0.792 | 0.709 | 0.614 | 13.7 | 1.356 | 1.549 | 1.207 | 11.7 | .2346 | .0864 | .0276 | .0165 | .0196 | 14.0 |
| | | MAE | 0.651 | 0.601 | 0.512 | 13.7 | 0.779 | 0.922 | 0.818 | 10.7 | .2924 | .1737 | .1015 | .0796 | .0856 | 13.8 |
| | FAN (2024) | MSE | 1.340 | 0.599 | 0.557 | 13.0 | 0.515 | 0.835 | 0.656 | 2.3 | .0663 | .0359 | .0162 | .0134 | .0069 | 9.8 |
| | | MAE | 0.682 | 0.485 | 0.496 | 12.7 | 0.486 | 0.780 | 0.629 | 2.3 | .4207 | .0915 | .0655 | .0642 | .0481 | 10.0 |
| Latest SOTA baselines | TimMixer (2024b) | MSE | 0.553 | 0.512 | 0.483 | 4.7 | 1.356 | 1.410 | 1.197 | 8.0 | .0600 | .0390 | .0199 | .0141 | .0081 | 7.8 |
| | | MAE | 0.466 | 0.413 | 0.371 | 2.7 | 0.764 | 0.883 | 0.814 | 4.7 | .1157 | .0934 | .0692 | .0631 | .0488 | 5.0 |
| | iTransformer (2024a) | MSE | 0.579 | 0.559 | 0.524 | 8.7 | 1.327 | 1.437 | 1.212 | 11.0 | .1511 | .0730 | .0319 | .0244 | .0149 | 11.4 |
| | | MAE | 0.536 | 0.473 | 0.424 | 11.7 | 0.760 | 0.897 | 0.839 | 8.3 | .1989 | .1367 | .0923 | .0840 | .0670 | 11.4 |
| | PatchTST (2023) | MSE | 0.548 | 0.514 | 0.485 | 5.0 | 1.323 | 1.413 | 1.199 | 8.3 | .0719 | .0398 | .0170 | .0130 | .0078 | 6.8 |
| | | MAE | 0.478 | 0.430 | 0.390 | 6.0 | 0.779 | 0.898 | 0.825 | 9.7 | .1310 | .0999 | .0683 | .0646 | .0516 | 7.4 |
| | TiDE (2023) | MSE | 0.637 | 0.574 | 0.529 | 11.3 | 1.322 | 1.424 | 1.223 | 10.7 | .1099 | .0454 | .0172 | .0125 | .0074 | 7.8 |
| | | MAE | 0.532 | 0.464 | 0.409 | 10.7 | 0.779 | 0.900 | 0.841 | 12.0 | .1664 | .1086 | .0689 | .0630 | .0505 | 7.8 |
| | TimesNet (2023) | MSE | 0.664 | 0.623 | 0.589 | 12.7 | 1.437 | 1.617 | 1.339 | 14.3 | .3088 | .1431 | .0698 | .0439 | .0311 | 13.6 |
| | | MAE | 0.601 | 0.523 | 0.486 | 12.7 | 0.852 | 0.987 | 0.899 | 14.3 | .3566 | .2318 | .1626 | .1196 | .1066 | 13.4 |
| | CycleNet (2024) | MSE | 0.583 | 0.542 | 0.505 | 8.0 | 1.315 | 1.411 | 1.195 | 6.0 | .0716 | .0368 | .0165 | .0128 | .0072 | 4.8 |
| | | MAE | 0.491 | 0.441 | 0.388 | 6.7 | 0.768 | 0.888 | 0.815 | 6.3 | .1282 | .0939 | .0663 | .0636 | .0497 | 5.0 |
| | Peri-midformer (2024) | MSE | 0.614 | 0.575 | 0.521 | 10.0 | 1.329 | 1.423 | 1.211 | 10.3 | .1756 | .0526 | .0263 | .0422 | .0191 | 11.8 |
| | | MAE | 0.511 | 0.454 | 0.397 | 8.7 | 0.772 | 0.891 | 0.830 | 8.3 | .2026 | .1126 | .0787 | .1022 | .0691 | 11.6 |
| | FRNet (2024) | MSE | 0.564 | 0.537 | 0.490 | 6.3 | 1.313 | 1.410 | 1.197 | 5.7 | .0645 | .0367 | .0171 | .0139 | .0073 | 6.2 |
| | | MAE | 0.475 | 0.431 | 0.374 | 4.7 | 0.765 | 0.887 | 0.816 | 6.0 | .1227 | .0941 | .0672 | .0654 | .0501 | 6.2 |
| | Hipeen (**Ours**) | MSE | **0.436** | **0.183** | **0.238** | **1.0** | **0.394** | **0.560** | **0.624** | **1.0** | **.0072** | **.0040** | **.0019** | **.0016** | **.0015** | **1.0** |
| | | MAE | **0.438** | **0.284** | **0.293** | **1.0** | **0.354** | **0.510** | **0.572** | **1.0** | **.0488** | **.0390** | **.0309** | **.0294** | **.0292** | **1.0** |

## 4 Experiments

Section 4.1 describes the *controlled* dataset, which requires gradient and raw value information for forecasting, and shows that only Hipeen can forecast it effectively. Section 4.2 evaluates Hipeen on over 500 real-world stock datasets, achieving SOTA, and demonstrates its comparable performance also on current benchmarks. Hipeen does not require a hyperparameter search. For *controlled* and Stock datasets, we fixed M=0.25, H=10, and the learning rate at 0.001. The look-back window was fixed at 96 throughout this study. Training details and baseline models are provided in Appendix D.

### 4.1 Experiments on the Controlled Datasets

To validate that current stationarization methods discard gradient and raw value information, we constructed three *controlled* datasets specifically designed to require this information for successful prediction. **Exponential** (requires grad. info.): New flipped exponential function begins when reaching a specific gradient. To prevent value-based prediction, the value of each flip point was varied. Experiments were conducted using three flipping intervals: [300, 350], [400, 450], and [500, 550]. **Threshold** (requires raw value info.): An increasing function with a gradient within a specified range that resets to 0 upon reaching 1. Owing to the discontinuous nature of the signals, which cannot be modeled by a linear backbone, two additional non-linear layers were introduced only for this dataset. We evaluated the function using three gradient ranges: [0.0005, 0.002], [0.001, 0.004], and [0.0015, 0.006]. **Sine wave** (requires both): To infer the current position on the long-range pattern, both the raw value and gradient information are required. We evaluated the model using five different periods: [2k, 3k], [3k, 4k], . . . , [6k, 7k]. All controlled datasets above consist of five independently generated channels of length 10k. The data is split into train, validation, and test sets in a 7:1:2 ratio (Wu et al., 2023; Wang et al., 2024c). For more details on controlled datasets, refer to Appendix B.1.

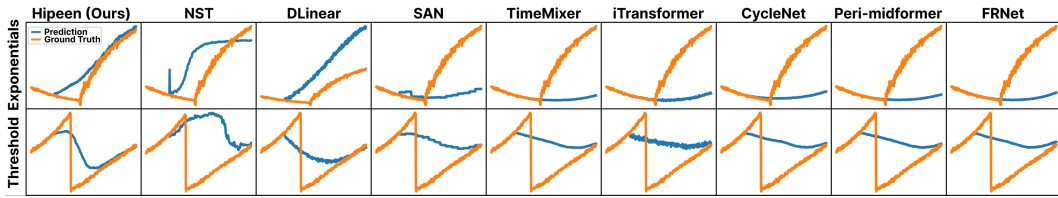

Figure 6: Ground truth (orange) and predictions (blue) for the Exponentials and Threshold tasks. All models except Hipeen failed, including various stationarization approaches. Additional illustrations, including Sine wave, are provided in Appendix E.1.

Table 1 demonstrates that Hipeen achieves the best performance on the *controlled* datasets with a significant margin. Notably, on the Sine wave dataset, Hipeen attains an MSE that is eight times lower than the second-best model. This substantial performance gap supports our hypothesis that conventional stationarization discards critical information—namely, gradients and raw values—necessary for forecasting. Figure 6 further illustrates this point: while Hipeen makes predictions based on both gradients and values, existing models fail to capture critical points altogether.

## 4.2 EXPERIMENTS ON REAL-WORLD DATASETS

We used the *S&P500* dataset (MVD, 2025) (Jan. 4, 2010 – Dec. 19, 2024) and the *Nifty50* dataset (Rao, 2021) (Jan. 1, 2000 – Apr. 30, 2021), both of which feature high look-back/horizon and train/test out-of-range rates. refer to Appendix B.1.2 for more details.

**S&P500.** After removing entries with missing values, 430 stocks remain. Since unpredictable non-stationary data (e.g., random walks) can yield high MSE, we applied three conditions: minimum baseline MSE $\leq 1$, $\leq 2$, and all datasets. Hipeen consistently achieves the best performance under all three criteria, showing a substantial average rank gap over the second-best baseline (Table 2). This robustness persists even when the default $H = 10$ is varied to 9 or 8. These results suggest that Hipeen outperforms existing models on real-world non-stationary time series and that its solution to the limitations of conventional stationarization also holds in practical scenarios.

**Nifty50.** To build a compact and predictable dataset, we applied an inclusion criterion of MSE $\leq 2$ to *Nifty50*. For fairness, inclusion was based on the baseline models (excluding Hipeen). Table 3 shows that Hipeen achieves the best performance on *Nifty50* across MAE, MAPE, and RMSE, attaining first rank on MAE in over 66% of the 48 combinations. We further assess the models in a trading scenario, including SMamba (Shi, 2024) and Stock-Transformer; STF (Mozaffari & Zhang, 2024) designed for stock forecasting. Hipeen achieves SOTA in Revenue, Sharpe, Sortino, and Calmar scores, delivering high returns with strong risk-adjusted performance. As Drawdown measures peak-to-trough decline, lower-return models often show better Drawdown. Trading methodology is provided in Appendix E.2.

Table 2: S&P500 dataset experiment, we reported the average rank of each model. Forecasting horizon = 96, averaged over three random seeds. The extended table is presented in Appendix E.2.

| Subsets | **All** (430) | | **MSE≤2** (402) | | **MSE≤1** (364) | |
|---|---|---|---|---|---|---|
| Metric | MSE Rank | MAE Rank | MSE Rank | MAE Rank | MSE Rank | MAE Rank |
| **Hipeen** | 5.23 | **3.50** | 4.98 | **3.28** | 4.98 | **3.32** |
| **Hipeen** (H:9) | **5.02** | 4.62 | **4.85** | 4.48 | **4.86** | 4.58 |
| **Hipeen** (H:8) | 5.46 | 6.13 | 5.34 | 6.06 | 5.38 | 6.12 |
| NST | 16.84 | 16.71 | 16.94 | 16.80 | 16.87 | 16.76 |
| Dlinear | 11.79 | 12.92 | 11.57 | 12.73 | 11.50 | 12.68 |
| Rlinear | 9.43 | 9.33 | 9.71 | 9.52 | 9.87 | 9.65 |
| Dish-TS | 14.46 | 15.17 | 14.53 | 15.21 | 14.56 | 15.21 |
| SAN | 6.89 | 7.72 | 6.53 | 7.44 | 6.26 | 7.24 |
| Leddam | 5.95 | 6.06 | 5.91 | 6.00 | 5.86 | 5.94 |
| TimeMixer | 14.29 | 13.57 | 14.46 | 13.75 | 14.58 | 13.88 |
| iTransformer | 10.41 | 10.62 | 10.33 | 10.53 | 10.33 | 10.52 |
| PatchTST | 9.67 | 8.93 | 9.90 | 9.11 | 9.87 | 9.05 |
| TiDE | 7.13 | 7.26 | 7.02 | 7.15 | 6.94 | 7.05 |
| TimesNet | 11.59 | 11.64 | 11.83 | 11.87 | 12.01 | 12.01 |
| CycleNet | 9.82 | 9.08 | 10.09 | 9.34 | 10.22 | 9.47 |
| Peri-midformer | 9.05 | 8.85 | 9.20 | 8.94 | 9.28 | 8.97 |
| FRNet | 6.98 | 6.44 | 7.06 | 6.52 | 7.00 | 6.48 |

Table 3: Results on the Nifty50 dataset (inclusion criteria: MSE≤2) averaged over 12 stocks, {12, 24, 48, 96} horizons and three seeds. The weakest models (NST, Dish-TS, and iTransformer) are omitted. Descriptions for each metric and the full table are provided in Appendix E.2. (R.: averaged rank)

| Models | **Hipeen** | Dlinear | RLinear | SAN | Leddam | DDN | FAN | TimeMixer | PatchTST | TiDE | TimesNet | CycleNet | Peri-midf. | FRNet | SMamba | STF |
|---|---|---|---|---|---|---|---|---|---|---|---|---|---|---|---|---|
| MAE | **0.198** | 0.294 | 0.205 | 0.212 | 0.210 | 0.248 | 0.252 | 0.213 | 0.209 | 0.217 | 0.242 | 0.212 | 0.206 | 0.206 | 0.912 | 0.818 |
| MAPE | **0.402** | 0.538 | 0.456 | 0.438 | 0.437 | 0.527 | 0.467 | 0.418 | 0.418 | 0.471 | 0.488 | 0.445 | 0.447 | 0.416 | 0.882 | 1.008 |
| RMSE | **0.274** | 0.386 | 0.282 | 0.288 | 0.287 | 0.330 | 0.336 | 0.291 | 0.288 | 0.293 | 0.324 | 0.288 | 0.283 | 0.282 | 1.021 | 0.928 |
| Revenue R. | **5.38** | 9.60 | 9.81 | 8.42 | 9.02 | 9.73 | 8.63 | 8.21 | 5.92 | 10.71 | 10.52 | 7.69 | 10.21 | 8.50 | 7.48 | 6.19 |
| Drawdown R. | 7.19 | 11.06 | 7.71 | 9.13 | **6.21** | 9.46 | 10.23 | 9.08 | 8.00 | 8.38 | 8.54 | 7.17 | 9.60 | 7.40 | 8.42 | 8.44 |
| Sharpe R. | **5.63** | 9.29 | 9.83 | 8.38 | 8.92 | 9.77 | 8.25 | 8.25 | 6.10 | 11.65 | 10.67 | 7.71 | 10.04 | 8.52 | 6.69 | 6.29 |
| Sortino R. | **5.65** | 9.54 | 9.75 | 8.35 | 9.02 | 9.83 | 8.29 | 8.21 | 6.10 | 11.54 | 10.75 | 7.67 | 9.98 | 8.35 | 6.77 | 6.19 |
| Calmar R. | **5.58** | 9.60 | 9.81 | 8.46 | 8.98 | 9.44 | 9.02 | 7.83 | 6.00 | 10.81 | 10.67 | 7.81 | 9.94 | 8.42 | 7.33 | 6.29 |

Table 4: Benchmark results on stationarization methods, averaged across 4 horizon lengths:{96, 192, 336, 720} and 3 seeds. The last row shows the number of inherent learnable parameters beyond the backbone (in Traffic 96; Note that the total number of parameters in DLinear is 19k in this case). The extended Table and standard deviation results are provided in Appendix E.3

| Model | Hipeen(Ours) | | NST | | DLinear | | RLinear* | | Dish-TS*† | | SAN*† | | Leddam*† | | DDN*† | | FAN*† | |
|---|---|---|---|---|---|---|---|---|---|---|---|---|---|---|---|---|---|---|
| Metric | MSE | MAE | MSE | MAE | MSE | MAE | MSE | MAE | MSE | MAE | MSE | MAE | MSE | MAE | MSE | MAE | MSE | MAE |
| Exchange | 0.335 | **0.397** | 0.461 | 0.454 | 0.354 | 0.414 | 0.412 | 0.431 | 0.511 | 0.507 | **0.330** | 0.398 | 0.398 | 0.420 | 0.499 | 0.454 | 0.423 | 0.450 |
| Weather | **0.224** | **0.261** | 0.288 | 0.314 | 0.265 | 0.315 | 0.244 | 0.268 | 0.239 | 0.303 | 0.251 | 0.296 | 0.240 | 0.270 | 0.268 | 0.302 | 0.241 | 0.292 |
| Solar | **0.205** | **0.257** | 0.350 | 0.390 | 0.330 | 0.401 | 0.260 | 0.304 | 0.208 | 0.286 | 0.313 | 0.338 | 0.254 | 0.281 | 0.292 | 0.348 | 0.255 | 0.280 |
| ETTm1 | **0.388** | **0.393** | 0.481 | 0.456 | 0.404 | 0.408 | 0.393 | 0.400 | 0.500 | 0.496 | 0.404 | 0.404 | 0.390 | 0.397 | 0.413 | 0.421 | 0.408 | 0.416 |
| ETTm2 | 0.301 | 0.332 | 0.306 | 0.347 | 0.354 | 0.402 | **0.283** | 0.333 | 1.364 | 0.779 | 0.284 | 0.340 | 0.289 | **0.329** | 0.288 | 0.333 | 0.323 | 0.373 |
| ETTh1 | 0.452 | **0.431** | 0.570 | 0.537 | 0.461 | 0.458 | **0.442** | 0.439 | 0.613 | 0.570 | 0.579 | 0.527 | 0.448 | 0.441 | 0.451 | 0.437 | 0.478 | 0.468 |
| ETTh2 | 0.435 | 0.428 | 0.526 | 0.516 | 0.563 | 0.519 | 0.410 | 0.422 | 3.176 | 1.248 | 0.395 | 0.420 | **0.385** | **0.406** | 0.432 | 0.434 | 0.508 | 0.489 |
| ECL | 0.197 | **0.294** | 0.193 | 0.296 | 0.225 | 0.319 | 0.203 | 0.302 | 0.237 | 0.344 | 0.270 | 0.364 | **0.191** | **0.294** | 0.260 | 0.356 | 0.205 | 0.301 |
| Traffic | 0.630 | **0.317** | 0.624 | 0.340 | 0.625 | 0.383 | 0.601 | 0.386 | 0.619 | 0.417 | 0.604 | 0.376 | 0.571 | 0.375 | 0.645 | 0.409 | **0.569** | 0.373 |
| Inh. Param. | 0 | | 0 | | 0 | | 0 | | 15081k | | 114k | | 3415k | | 5539k | | 59k | |

\* Replaced the backbone with a linear model to evaluate each stationarization, removing architectural influence.
† However, some methods inherently contain multiple non-linear layers, offering extra architectural gains.

**Conventional Benchmark.** Hipeen also maintains competitive performance on relatively stationary benchmarks. Table 4 compares Hipeen with major stationarization methods. Using a learning-free stationarization module with no internal parameters, Hipeen outperforms other learning-free approaches such as NST, DLinear, and RLinear (RevIN). However, as shown in the last row, some methods incorporate multiple layers and non-linear activations within their stationarization modules, gaining architectural advantages that hinder a fair comparison. Notably, Leddam contains 180× more parameters than DLinear (excluding the backbone), raising concerns about practicality. Even so, Hipeen achieves strong performance, demonstrating its effectiveness on relatively stationary signals.

**Analysis.** Conceptually, scale parameter $M$ sets the smallest decimal place and hierarchy level $H$ the total number of digits; e.g., $M = 0.1$ and $H = 2$ can represent values from 0.1 to 9.9. Experiments on benchmark datasets varying $H$ and $M$ show that Hipeen is robust across a wide range of $H$ if $M$ is small enough (Figure 7), analogous to that representing 1.63 as 01.630 does not enhance representational accuracy. We also analyzed the bias term $B$ in Equation 1. Comparing Hipeen with no bias, bias on $N$-dim. ($[0, 2\pi)^{N \times 1}$), and on $H$-dim. ($[0, 2\pi)^{1 \times H}$). Table 5 shows that adding a bias term is crucial, and channel-wise bias ($N$-dim) is especially important.

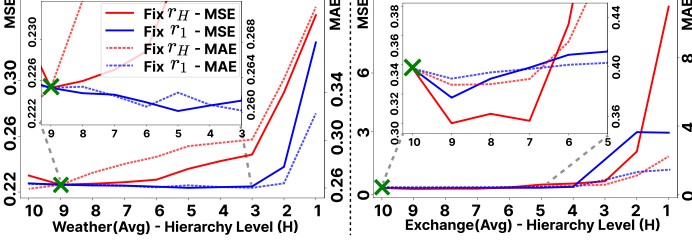

Figure 7: Hipeen performance with varying hierarchy level $H$ and scale $M$. Red: Fix $r_H$, Blue: Fix $r_1$

Table 5: Experiments on different bias settings, averaged over 3 datasets and 3 seeds. Each value represents the MSE. Full results are provided in Appendix E.4.

| Horizon | Ours | N-dim. | H-dim. | No B |
|---|---|---|---|---|
| 96 | **0.207** | 0.207 | 0.216 | 0.382 |
| 192 | **0.269** | 0.274 | 0.354 | 0.936 |
| 336 | **0.374** | 0.382 | 0.510 | 1.023 |
| 720 | **0.516** | 0.534 | 0.885 | 4.778 |

## 5 CONCLUSION

We demonstrate that widely used TSF benchmarks underrepresent real-world non-stationarity and that conventional stationarization methods can cause critical information loss. To address this, we introduce Hipeen, a novel stationarization method that preserves essential gradient and absolute value information by projecting signals into a hierarchical periodic representation. Hipeen is the only model to succeed on our controlled datasets designed to highlight this information loss. Moreover, it substantially outperforms state-of-the-art models on highly non-stationary real-world stock datasets while remaining competitive on standard benchmarks, underscoring the importance of information-preserving stationarization for robust time series forecasting. **Limitation & Future Work**. While we have identified the limitations of existing TSF benchmarks, we do not provide representative non-stationary datasets to address these shortcomings. Future work should focus on systematically evaluating the extent of non-stationarity in current benchmarks and on developing datasets that better reflect the complexities of real-world non-stationary signals.

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

## A  DETAILED RELATED WORKS

### A.1  TIME SERIES MODELING.

Deep learning has substantially advanced time series forecasting by introducing architectures that more effectively capture temporal dynamics and inter-variable dependencies (Hyndman & Athana-sopoulos, 2018; Liu et al., 2024a; Wang et al., 2024b;a). Recent models can be broadly categorized into several key paradigms: Transformer-based, CNN-based, and MLP/linear-based architectures, with a growing trend towards general-purpose foundation models (Kim et al., 2025b).

Transformer-based models have become prominent due to their capacity to model long-range de-pendencies. Autoformer (Wu et al., 2021) and FEDformer (Zhou et al., 2022) both incorporate decomposition into trend and seasonal components, with the latter enhancing efficiency through Fourier-based attention. PatchTST (Nie et al., 2023) introduces a patching strategy that segments time series into fixed-length patches for Transformer input, while modeling each variable independently to improve generalization. Crossformer (Zhang & Yan, 2022) proposes cross-dimension attention to jointly capture temporal and feature-wise dependencies. The Non-stationary Transformer (Liu et al., 2022b) introduces a two-part framework comprising series stationarization and de-stationary attention, which normalizes input statistics and restores non-stationary information lost in traditional attention mechanisms, thereby improving robustness to distribution shifts. iTransformer (Liu et al., 2024a) reformulates the input structure by treating each variable as a token, offering an inverted perspective on Transformer-based time series modeling.

CNN-based approaches exploit multi-scale feature extraction to capture temporal patterns. SCINet (Liu et al., 2022a) adopts a recursive downsample–convolve–interact design to model complex temporal dynamics through hierarchical resolution. TimesNet (Wu et al., 2023) transforms time series into 2D representations based on learned periods and applies inception-style convolutional blocks to capture both intra- and inter-period variations, achieving strong performance on various forecasting benchmarks.

Simpler architectures based on MLPs and linear layers have also demonstrated competitive perfor-mance. DLinear (Zeng et al., 2023) applies lightweight linear projections to decomposed components for efficient forecasting. TimeMixer (Wang et al., 2024b) extends this design with shift-based mixing and channel-wise MLPs, enabling scalable modeling without attention. TiDE (Das et al., 2023) employs a dense MLP-based encoder–decoder to effectively handle covariates and non-linear rela-tionships, showing strong results in long-horizon forecasting tasks. Some models aim for broader applicability beyond forecasting. TimeMixer++ (Wang et al., 2024a) generalizes time series modeling through the Time Series Pattern Machine, which transforms sequences into multi-resolution temporal images and integrates axis-aware decomposition with multi-scale feature fusion, supporting tasks such as classification, imputation, and anomaly detection alongside forecasting.

### A.2  STOCK PRICE PREDICTION.

Despite the rapid progress of general time series forecasting, stock price prediction research remains comparatively conservative, with many studies still grounded in traditional or narrowly focused deep learning models. For instance, (Mozaffari & Zhang, 2024) evaluate LSTM against a Transformer-based model for stock index prediction and show that Transformers provide gains mainly by better capturing temporal dependencies. To mitigate non-stationarity, decomposition-based hybrids have been proposed. SVMD–LSTM (Agarwal et al., 2025) decomposes stock series into intrinsic mode functions before applying LSTMs, demonstrating more stable forecasts than standalone recurrent models. However, the predictive head still follows a relatively simple architecture.

Other works explores modern sequence modeling approaches. SMamba (Shi, 2024) adapts Mamba (Gu & Dao, 2024) to stock data, showing improved accuracy through efficient long-range dependency modeling. PMANet (Zhu et al., 2024) enhances attention mechanisms and multi-scale convolution to better handle long input sequences and anomaly points, yet it remains a domain-specific design optimized for hand-crafted financial features.

Overall, while time series forecasting architectures diversify, stock price prediction remains grounded in narrowly scoped, task-specific designs rather than the unified and scalable approaches emerging in the broader field. Also, existing TSF models are largely benchmark-driven and have not been

thoroughly evaluated on stock datasets. Using stocks as a representative non-stationary dataset, we show that Hipeen achieves superior performance in MAE, RMSE, and MAPE. Furthermore, when used for prediction-based trading, Hipeen attains the highest returns and risk-adjusted performance, demonstrating its applicability to the stock domain.

# B  DATASETS, BASELINE MODELS, AND IMPLEMENTATION DETAILS

## B.1  DATASETS

Table 6: Detailed descriptions of datasets. The look-back window for all data is 96. The dataset size is organized in (Train, Validation, Test).

| Tasks | Dataset | Dim | Horizon Length | Dataset Size | Frequency | Non-stationarity∗ | Information |
|---|---|---|---|---|---|---|---|
| | Sine1 | 5 | {96, 192, 336, 720} | (6905, 1001, 2001) | 1 step | 0.92/0.00 | Synthetic |
| | Sine2 | 5 | {96, 192, 336, 720} | (6905, 1001, 2001) | 1 step | 0.91/0.00 | Synthetic |
| | Sine3 | 5 | {96, 192, 336, 720} | (6905, 1001, 2001) | 1 step | 0.90/0.00 | Synthetic |
| | Sine4 | 5 | {96, 192, 336, 720} | (6905, 1001, 2001) | 1 step | 0.87/0.00 | Synthetic |
| | Sine5 | 5 | {96, 192, 336, 720} | (6905, 1001, 2001) | 1 step | 0.86/0.00 | Synthetic |
| Controlled | Exp.1 | 5 | {96, 192, 336, 720} | (6905, 1001, 2001) | 1 step | 0.75/0.14 | Synthetic |
| | Exp.2 | 5 | {96, 192, 336, 720} | (6905, 1001, 2001) | 1 step | 0.86/0.05 | Synthetic |
| | Exp.3 | 5 | {96, 192, 336, 720} | (6905, 1001, 2001) | 1 step | 0.92/0.09 | Synthetic |
| | Thr.1 | 5 | {96, 192, 336, 720} | (6905, 1001, 2001) | 1 step | 0.83/0.00 | Synthetic |
| | Thr.2 | 5 | {96, 192, 336, 720} | (6905, 1001, 2001) | 1 step | 0.62/0.00 | Synthetic |
| | Thr.3 | 5 | {96, 192, 336, 720} | (6905, 1001, 2001) | 1 step | 0.46/0.00 | Synthetic |
| | ETTh1 | 7 | {96, 192, 336, 720} | (8545, 2881, 2881) | 15 min | 0.16/0.00 | Temperature |
| | ETTh2 | 7 | {96, 192, 336, 720} | (8545, 2881, 2881) | 15 min | 0.21/0.01 | Temperature |
| | ETTm1 | 7 | {96, 192, 336, 720} | (34465, 11521, 11521) | 15min | 0.21/0.00 | Temperature |
| Benchmark | ETTm2 | 7 | {96, 192, 336, 720} | (34465, 11521, 11521) | 15min | 0.22/0.01 | Temperature |
| Datasets | Weather | 21 | {96, 192, 336, 720} | (36792, 5271, 10540) | 10 min | 0.36/0.00 | Weather |
| | Solar-Energy | 137 | {96, 192, 336, 720} | (36601, 5161, 10417) | 10min | 0.09/0.00 | Electricity |
| | Electricity | 321 | {96, 192, 336, 720} | (18317, 2633, 5261) | Hourly | 0.08/0.00 | Electricity |
| | Traffic | 862 | {96, 192, 336, 720} | (12185, 1757, 3509) | Hourly | 0.04/0.00 | Transportation |
| | Exchange | 8 | {96, 192, 336, 720} | (5120, 665, 1422) | Daily | 0.69/0.27 | Finance |
| | ADANIPORTS | 9 | {12, 24, 48, 96} | (2230, 334, 665) | Daily | 0.32/0.00 | Stock |
| | BAJAJ-AUTO | 9 | {12, 24, 48, 96} | (2146, 322, 641) | Daily | 0.32/0.21 | Stock |
| | HDFC | 9 | {12, 24, 48, 96} | (3619, 532, 1062) | Daily | 0.32/0.00 | Stock |
| | HEROMOTOCO | 9 | {12, 24, 48, 96} | (3619, 532, 1062) | Daily | 0.31/0.38 | Stock |
| Nifty50 | HINDALCO | 9 | {12, 24, 48, 96} | (3619, 532, 1062) | Daily | 0.34/0.00 | Stock |
| Stock | LT | 9 | {12, 24, 48, 96} | (2833, 421, 837) | Daily | 0.34/0.00 | Stock |
| Datasets | MARUTI | 9 | {12, 24, 48, 96} | (3003, 445, 886) | Daily | 0.37/0.77 | Stock |
| | NTPC | 9 | {12, 24, 48, 96} | (2766, 411, 818) | Daily | 0.27/0.00 | Stock |
| | POWERGRID | 9 | {12, 24, 48, 96} | (2256, 338, 672) | Daily | 0.27/0.09 | Stock |
| | TATASTEEL | 9 | {12, 24, 48, 96} | (3619, 532, 1062) | Daily | 0.35/0.02 | Stock |
| | TECHM | 9 | {12, 24, 48, 96} | (2449, 365, 728) | Daily | 0.37/0.00 | Stock |
| | TITAN | 9 | {12, 24, 48, 96} | (3619, 532, 1062) | Daily | 0.37/0.00 | Stock |

∗ The Non-stationarity is obtained by measuring the out-range rate between the look-back/horizon and train/test.

A summary of the entire training dataset is provided in Table 6. This table presents the number of channels (Dim) in the data, the lengths of the trained horizons, the sizes of the train, validation, and test sets, the sampling frequency, the degree of non-stationarity, and the types of data.

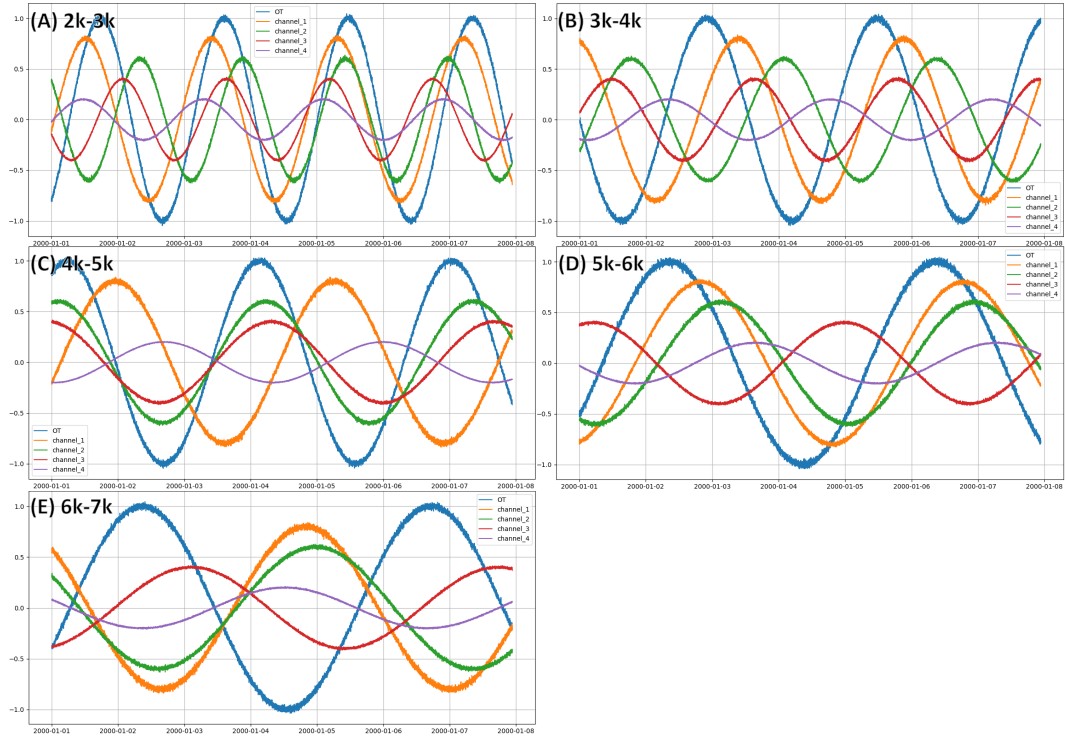

Figure 8: The full 10k timesteps of the `Sine wave` datasets are shown. Each color represents a channel that was independently generated. To enhance visual clarity, each channel was plotted with a different amplitude; however, since the data undergo global normalization based on training statistics during preprocessing, this has the same effect as using identical amplitudes across channels.

### B.1.1 CONTROLLED DATASETS

All controlled datasets are multivariate time series consisting of 5 channels and 10k timesteps. Each channel is independently generated from a specified distribution.

**The Sine wave dataset** represents the most basic form of time series, generated by adding Gaussian noise to long-period Sine waves. The standard deviation of the added Gaussian noise is sampled from Uniform[0.01, 0.02], and the period length is sampled from the following ranges: (1) Uniform[2k, 3k], (2) Uniform[3k, 4k], (3) Uniform[4k, 5k], (4) Uniform[5k, 6k], and (5) Uniform[6k, 7k]. In this way, five types of `Sine wave` datasets were created. The resulting five datasets are visualized in Figure 8.

**The Exponentials dataset** is designed to model systems in which changes in the time series are triggered by reaching a certain gradient (rate of change). To simulate such behavior, exponential decay functions are generated, and once a function reaches a predefined gradient, a new exponential decay function—flipped vertically—is initiated. While the trigger gradient for flipping is fixed for each function, the initial gradient of the new function after the flip is randomly sampled within a range. This makes the value of each flipping point vary and prevents the model from learning the flip timing based on value rather than gradient. The base of the exponential function is sampled from Uniform[1.004, 1.007] and is kept constant throughout the series. For each exponential decay segment, the end value is fixed at +200 to ensure consistent flipping gradients, but the start value varies to induce diverse initial slopes. These flipped segments are concatenated to form the entire time series. The duration of each segment (i.e., the flipping interval defined as start value - end value) is sampled from: (1) Uniform[300, 350], (2) Uniform[400, 450], and (3) Uniform[500, 550]. The resulting three datasets are visualized in Figure 9.

**The Threshold dataset** is designed to simulate systems in which changes are triggered when the time series value reaches a specific value. Once an increasing function with a certain gradient range

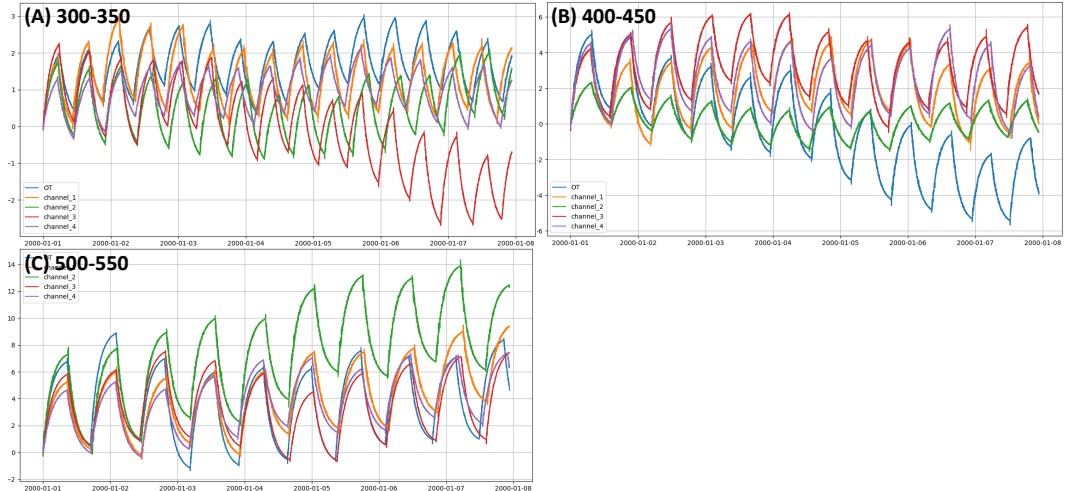

Figure 9: The full 10k timesteps of the `Exponentials` datasets are shown. Each color represents a channel that was independently generated. As the length of each exponential segment increases, the frequency of flipping decreases.

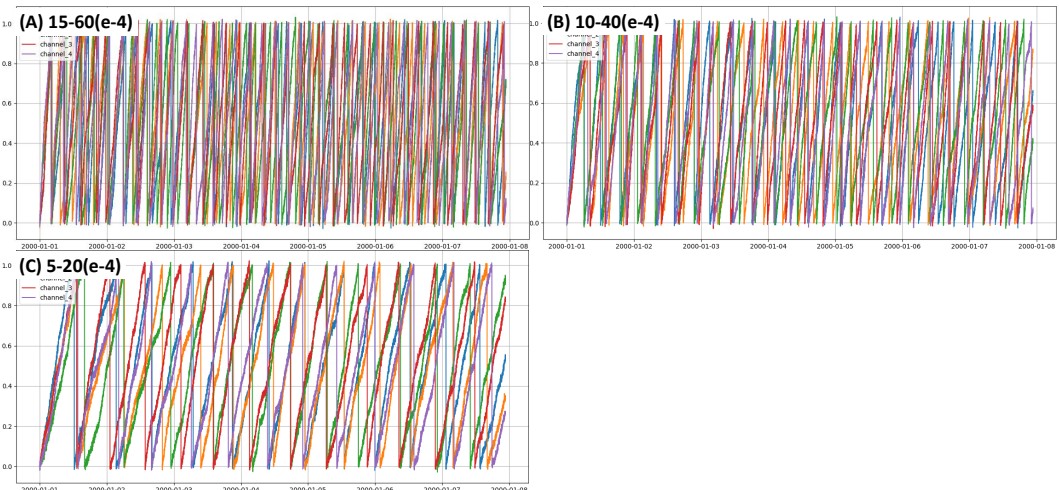

Figure 10: The full 10k timesteps of the `Threshold` datasets are shown. Each color represents a channel that was independently generated. As the gradient of linear segment decreases, the frequency of reaching Threshold decreases.

reaches the value 1, the value is reset by subtracting 1, and the process repeats. The increasing function is composed of piecewise linear segments, where each segment has an x-length sampled from Uniform[50, 100] and a gradient sampled from a specified range. To control the period at which the function reaches the threshold, we sample gradients for each segment from the following ranges: (1) Uniform[0.0005, 0.002], (2) Uniform[0.001, 0.004], and (3) Uniform[0.0015, 0.006]. The resulting three datasets are visualized in Figure 10.

All controlled datasets are created from csv file using the Dataset_Custom class from the Time Series Library (Wu et al., 2023; Wang et al., 2024c), following the same procedures used for processing benchmark datasets such as `Weather` and `Traffic`. This class includes a default preprocessing step of global normalization based on training set statistics, which is applied uniformly across both custom and benchmark datasets. For a summary of each dataset, refer to the "Scenarios" section of Table 6.

### B.1.2 REAL-WORLD STOCK DATASETS

We utilized a publicly available *S&P500* (MVD, 2025) and *NIFTY50* (Rao, 2021) Stock Market dataset under the CC0 (Public Domain) license. The dataset comprises daily price and trading volume information for the 500 constituent stocks of the S&P 500 index, which represents large-cap companies listed on U.S. stock exchanges, and for the 50 constituent stocks of the NIFTY 50 index, sourced from the National Stock Exchange (NSE) of India. Each stock's data is stored in a separate .csv file, along with a metadata file containing high-level information about each company. Given the high non-stationarity typically observed in stock market time series, most

| Feature | Description |
|---|---|
| Prev Close | Closing price of the previous day |
| Open | Opening price of the day |
| High | Highest price of the day |
| Low | Lowest price of the day |
| Last | Last traded price of the day |
| Close | Official closing price |
| VWAP | Volume Weighted Avg. Price |
| Volume | Shares traded |
| Turnover | Volume × Price |

Table 7: Stock feature descriptions

tasks focus on short-term forecasting over several months. Due to this and the limited sequence lengths of many stocks, we set the prediction horizon to 96.

1. S&P 500 dataset.

The dataset spans from January 4, 2010, to December 19, 2024, and contains six columns: Adj Close, Close, High, Low, Open, and Volume, providing comprehensive daily price and trading volume information for each stock. After excluding any rows with missing values, a total of 430 stocks were used in the analysis.

2. NIFTY 50 dataset.

The data spans over two decades, from January 1, 2000, to April 30, 2021. To ensure data quality, we excluded stocks with less than 3000 days of historical records, as they produce an insufficient amount of data for the validation set with the conventional 7:1:2 dataset split. Additionally, to eliminate stocks where past data offers little predictive value (i.e., nearly random series), we excluded those where all 16 recent baseline models (except Hipeen) yielded test MSEs greater than 2.0. After filtering, 12 stocks remained: ADANIPORTS, BAJAJ-AUTO, HDFC, HEROMOTOCO, HINDALCO, LT, MARUTI, NTPC, POWERGRID, TATASTEEL, TECHM, and TITAN. These showed an average minimum MSE of 1.1 across the 9 models, compared to 101.9 for the excluded group. Each time series is multivariate with 9 input channels: Stock market data contains multiple price and volume-related features that reflect daily trading behavior. To help interpret the multivariate inputs used in our models, Table 7 summarizes the meaning of each feature.

Dataset construction followed the same procedure as with the controlled data. Specifically, we used the Dataset_Custom class from the Time Series Library (Wang et al., 2024c), which is also employed for handling benchmark datasets such as Weather and Traffic. Please refer to the "Stock Datasets" section of Table 6 for detailed characteristics of each dataset.

### B.1.3 REAL-WORLD BENCHMARK DATASETS

We used nine public benchmarks that are widely adopted in time series forecasting research: Weather, Solar-Energy, Electricity, Traffic, Exchange, ETTh1, ETTh2, ETTm1, and ETTm2. (There is an ongoing debate about whether exchange datasets should be used as benchmarks (Bergmeir, 2024), and recent studies differ in whether they include them.; As an example, while the CycleNet (Lin et al., 2024) excluded these datasets, they were included in the Peri-Midformer (Wu et al., 2024) paper. However, we included them to enable a more comprehensive comparison.) The datasets were sourced from the Time Series Library (Wang et al., 2024c) and the TimeMixer++ paper (Wang et al., 2024a). Data splitting and preprocessing were conducted using the Dataset_ETT_minute class (for ETTm1 and ETTm2), Dataset_ETT_hour class (for ETTh1 and ETTh2), and Dataset_Custom class (for the remaining datasets) provided by the Time Series Library (Wang et al., 2024c). Please refer to the "Benchmark Datasets" section of Table 6 for the characteristics of each dataset.

### B.2 BASELINE MODELS

To evaluate and demonstrate the effectiveness of Hipeen across the diverse sets of forecasting tasks, we compare it against 16 state-of-the-art baseline models and 2 stock forecasting models encompassing a broad spectrum of architectural paradigms. These include Transformer-based models such as iTransformer (Liu et al., 2024a), PatchTST (Nie et al., 2023), Peri-midformer (Wu et al., 2024), and Non-Stationary Transformer (Liu et al., 2022b); CNN-based models including TimesNet (Wu et al., 2023); MLP-based models such as TimeMixer (Wang et al., 2024b), CycleNet (Lin et al., 2024), TiDE (Das et al., 2023), and DLinear (Zeng et al., 2023); hybrid architectures like FRNet (Zhang et al., 2024), and RLinear (Zeng et al., 2023); Stationarization methods such as Dish-TS (Fan et al., 2023), SAN (Liu et al., 2024b), Leddam (Yu et al., 2024), DDN (Dai et al., 2024), and FAN (Ye et al., 2024); Stock forecasting models such as SMamba (Shi, 2024) and STF Mozaffari & Zhang (2024). These baselines represent the current best-performing models in time series forecasting and serve as a strong foundation for comparative evaluation.

### B.3 IMPLEMENTATION DETAILS

All code implementations are based on the Time Series Library (Wu et al., 2023; Wang et al., 2024c). Using the dataset classes provided by the library, we preprocessed all the controlled datasets, benchmark, and stock data. We also utilized the model architecture, training, and evaluation pipelines provided by the library for all baseline models, ensuring consistency and reproducibility across experiments. For the benchmark datasets, we adopted the default hyperparameters specified by the library for each baseline model. In cases where the library did not provide hyperparameter settings—such as for non-benchmark datasets—we used the hyperparameters from ETTh1 as the default configuration. Additional experiments to determine more suitable hyperparameters for these datasets are underway, and their results will be incorporated into the final version of the manuscript.

## C HIPEEN AND LINEAR BACKBONE

### C.1 HIPEEN PROJECTION AND ESTIMATOR

#### C.1.1 EMA MEMORY IN THE TRAINING PHASE

During the training process of Hipeen, the training loss is stored in an internal memory. Each loss is computed by treating a pair of $S$ and $C$ as a vector representing $\theta$, and calculating the cosine distance between the model's prediction and the Hipeen projection of the label. The resulting loss values form a tensor of shape $B \times K \times N \times H$, where $B$ denotes the batch dimension. This tensor is averaged over the batch dimension to yield a $K \times N \times H$ tensor $Q$, which is then stored in memory for use in the estimation phase. The memory is updated using an exponential moving average (EMA) defined as:

$$Q_{\text{memory}} = (1 - \text{sm}) \cdot Q_{\text{memory}} + \text{sm} \cdot Q_{\text{new}},$$

where sm is the smoothing factor. At the initial stage of training, $Q_{\text{memory}}$ is simply set to $Q$. Through this process, training progressively accumulates meaningful loss statistics across all time dimensions $K$, channel dimensions $N$, and hierarchy dimensions $H$ over the entire training set. For simplicity, we fix the smoothing factor to 0.005 throughout all experiments. However, it is advisable to adjust this value according to the number of training samples. As a first choice, we recommend setting the smoothing factor to (training batch size)/(training sample size)

#### C.1.2 HIPEEN ESTIMATOR PEAK FILTERING

In addition to the estimation method in the main text, we applied a simple peak filtering technique to the Hipeen estimator to enhance its robustness. This method is designed to prevent the final result $V$ from being significantly affected by one or two outliers during the ensemble process of $H$ estimations along the hierarchy dimension. In Equation 7 of the main text, the number of rotations $p_h$ added to $\theta_h$ is determined by finding the peak closest to the previous $V_{\text{est}}$. We consider a peak to be an outlier if its distance from the previous $V_{\text{est}}$ exceeds $\frac{1}{2}\pi r_h$. (Since the distance between adjacent peaks is $2\pi r_h$, the distance to the nearest peak from $V_{\text{est}}$ can range from 0 to $\pi r_h$.) For such outliers, the $h$-th $V_h$ and $v_h$ is excluded from the update, and the process moves on to the next $h + 1$. This filtering helps mitigate the performance degradation caused by outliers in the ensemble process.

## C.2 LINEAR BACKBONE ARCHITECTURE AND EXTRA ENSEMBLE

### C.2.1 LINEAR BACKBONE USED IN HIPEEN

The backbone architecture used in Hipeen, referred to as `Linear_model`, is designed to process multivariate time-series data. We decompose the $2H$ ensemble dimension into $H \times 2$; consequently, the input has the shape $(B, L, C, H, 2)$, where $B$ is the batch size, $L$ is the input sequence length, $C$ is the number of channels (features), $H$ is the hierarchy level (half of the ensemble dimension), and the last dimension of size 2 represents a sine and cosine projection. The model uses a simple 3-layer convolutional architecture and includes neither non-linear activation functions nor dropout.

The model is composed of the following three consecutive layers:

- **Temporal Mixing layer (`Time_mix`)**: Applies 3D convolutions along the temporal and ensemble axes while preserving the channel structure.
- **Channel Mixing layer (`Channel_mix`)**: Applies 3D convolutions across the channel and ensemble axes while preserving the temporal structure.
- **Final Temporal Mixing layer (`Time_mix_fin`)**: Converts the sequence from input look-back window length $L$ to output horizon $K$. This layer shares the module with `Time_mix`.

Input/Output Shape Summary

- **Input:** $(B, L, C, H, 2)$
- **Output:** $(B, K, C, H, 2)$

**Linear_model Architecture**    The full model is summarized as follows:

$$x \leftarrow x + \texttt{Time\_mix}(x)$$
$$x \leftarrow x + \texttt{Channel\_mix}(x)$$
$$x \leftarrow \texttt{Time\_mix\_fin}(x)$$

This architecture resembles the simplest version of TSMixer (Ekambaram et al., 2023) without activation and dropout, composed of spatial/channel-wise feature mixing, and finally projects to the desired output length.

**Time_mix Module**    This module applies a 3D convolution across the $(L, H, 2)$ dimensions after normalizing each spatial unit using GroupNorm:

- **Normalization:** GroupNorm fuctions as `LayerNorm` on the $(L, H, 2)$ axes.
- **3D Convolution:** The input and output length $(L_{in}, L_{out})$ are fully connected to each other. And a convolutional kernel of size $(9, 3)$ is used, with padding to preserve the spatial resolution; 9 along the $H$ dimension and 3 along the final dimension of length 2.

The output has shape $(B, L_{\text{out}}, C, H, 2)$, preserving the channel structure.

**Channel_mix Module**    This block focuses on channel-level interactions:

- **Layer Normalization:** `LayerNorm` is applied directly to the $(C, H, 2)$ dimensions without reshaping.
- **3D Convolution:** The input and output channel ($C$; identical dimension) are fully connected to each other. And a convolutional kernel of size $(9, 3)$ is used with padding to preserve the spatial resolution of ensemble; 9 on $H$ dimension and 3 on the last dimension of length 2.

The output shape remains $(B, L, C, H, 2)$, preserving the temporal structure.

All convolutions use appropriate zero-padding to maintain spatial alignment. The temporal mapping layer, `Time_mix_fin`, which uses the **Time_mix module**, changes the input-sequence length from L to the desired forecasting horizon K, enabling the model to predict future values.

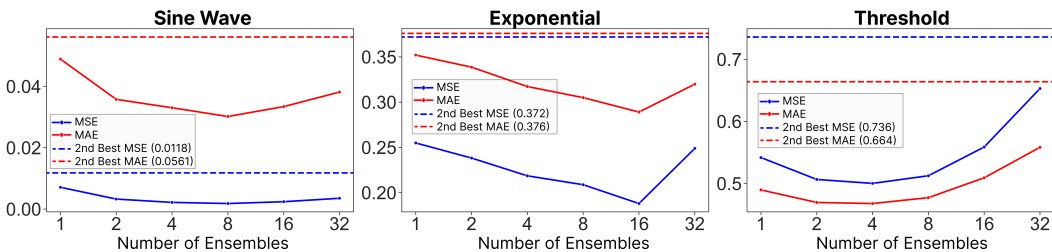

Figure 11: **Performance as a function of the extra ensemble dimension.** Results are reported for the Sine Wave (4k–5k), Exponential (400–450), and Threshold (0.001–0.004) datasets as the extra ensemble dimension increases from 1 to 32. The dashed line denotes the performance of the second-best model.

### C.2.2 EXTRA ENSEMBLE

To enhance Hipeen's representational capacity without increasing the number of learnable parameters, we propose an **extra ensemble** mechanism. This approach preserves Hipeen's original non-learning nature and supports efficient parallel computation. The extra ensemble mechanism enables Hipeen to incorporate additional periodic diversity at each hierarchical level of its original projection. This enhances the model's ability to capture richer and more varied temporal patterns within each frequency hierarchy.

In the original Hipeen formulation, a scalar value $V$ is projected to an $H$-dimensional vector $\boldsymbol{\theta} \in \mathbb{R}^H$ using a fixed radius vector $\mathbf{r} \in \mathbb{R}^H$. In contrast, our extra ensemble introduces an additional ensemble dimension $E$ by rescaling the radius vector $\mathbf{r}$ with $W \in [0.5, 1.5]^{E \times H}$ resulting in an expanded radius matrix $\mathbf{r}^{ext} \in \mathbb{R}^{E \times H}$ .

$$\mathbf{r}_h = M \times 2^h \,; h \in \{1, ..., H\}, \quad \mathbf{r}_{e,h}^{ext} = \mathbf{r}_h \cdot W_{e,h}, \quad \text{where } W_{e,h} \sim \text{Uniform}(0.5, 1.5)$$

Then angular tensor $\theta$ is calculated using $\mathbf{r}^{ext}$ and extended bias $B^{ext} \in [0, 2\pi)^{N \times E \times H}$

$$V \to \boldsymbol{\theta} : \qquad \theta_{e,h}^n = \left( \frac{V^n}{\mathbf{r}_{e,h}^{ext}} + B_{e,h}^n \right) \bmod 2\pi, \quad \boldsymbol{\theta}^n \in [0, 2\pi)^{E \times H}$$

This results in an angular projection output with extra ensemble dimension $\boldsymbol{\theta} \in [0, 2\pi)^{E \times H}$. The extra ensemble introduces period variations within the same hierarchy level. The ensemble dimension $E$ is folded into the batch dimension, enabling all $E$ projections to be processed in parallel without modifying the backbone model or increasing its parameters. During inference, predictions from the $E$ ensembles are averaged to obtain the final estimation:

This strategy may be similar in spirit to batch ensembles (Wen et al., 2020) but is more efficient due to zero additional learnable parameters. It enables Hipeen to model a wider range of periodic components more flexibly and expressively. We fix $E = 16$ for all experiments, except for `Traffic`, `ECL`, and `Solar-Energy` datasets when the horizon is 720, where $E = 8$ is used to reduce the computation. Although this mechanism enhances expressiveness when a linear backbone is used, it is not mandatory when Hipeen is paired with more expressive backbones.

Figure 11 shows how performance varies as the number of extra ensemble components increases from 1 to 32. Across all datasets, Hipeen consistently outperforms the second-best model. The performance curve exhibits a U-shaped trend—initially decreasing and then improving as the ensemble size grows. In practice, selecting 4–8 ensembles appears to be a good balance. Notably, these extra ensembles add no learnable parameters and are computed efficiently along the batch dimension.

## D TRAINING AND HYPERPARAMETER SEARCH

### D.1 COMPUTATION RESOURCE AND ENVIRONMENT

All experiments were conducted on either a single NVIDIA L40 GPU (48 GB VRAM) or an NVIDIA A100 GPU (80 GB VRAM). We used PyTorch (Paszke, 2019) 2.7.0 in a Python 3.11 environment,

along with the following additional packages, identical to those used in the Time Series Library (Wu et al., 2023; Wang et al., 2024c): einops, local-attention, matplotlib, numpy, pandas, patool, reformer-pytorch, scikit-learn, scipy, sktime, sympy, tqdm, and PyWavelets. All auxiliary packages were employed in their most recent versions available at the time of experimentation.

## D.2 Training & evaluation details

The training and evaluation of the model were based on the training and evaluation code from the Time Series Library (Wu et al., 2023; Wang et al., 2024c). The evaluation metrics used in the experiments—Mean Squared Error (MSE) and Mean Absolute Error (MAE)—follow the standard metrics commonly used in time series forecasting (TSF) literature (Liu et al., 2024a; Nie et al., 2023; Zeng et al., 2023) and are consistent with those implemented in the Time Series Library. During training, the optimizer Adam (Kingma & Ba, 2014) with default hyperparameter was used. A custom learning-rate schedule was employed: the initial learning rate was kept for the first three epochs and then multiplied by 0.8 at each subsequent epoch to ensure a gradual decrease. A batch size of 32 was used during training, which is the default setting in the Time Series Library, except for the 720-horizon training of the Traffic, Electricity, and Solar-Energy datasets, where a batch size of 16 was used. Training was conducted for up to 30 epochs with early stopping based on the validation MSE loss. The best model was not saved during the first three epochs. (Training was configured to run for a minimum of four epochs.)

## D.3 Hyperparameter search

We conducted a hyperparameter search only for the benchmark dataset and used the fixed hyperparameters for the rest of the experiments. The search was performed on only two parameters: (1) the learning rate and (2) combinations of the scale $M$ and hierarchy level $H$. The full search space for the learning rate is [0.002, 0.001, 0.0005], and the search space for combinations of $M$ and $H$ is [1,8], [0.5,9], [0.25,10]. Since Hipeen's performance is not highly sensitive to the choice of hyperparameters, $M$ and $H$ can be fixed at 0.25 and 10, respectively, without significant loss in performance, although a search can still be performed if desired.

For each hyperparameter setting, we averaged the validation loss over three random seeds and selected the configuration with the lowest average validation loss. Due to computational constraints, the hyperparameter search space was further reduced to the subspace of the defined search space, based on a sequence length of 96. We plan to explore the full search space and conduct additional tuning in extended search regions in the final version. Table 8 presents the selected hyperparameters for each experiment.

For the Controlled and Stock datasets, we did not perform hyperparameter searches, following the protocol of other baseline models to ensure a fair comparison. For the baselines, hyperparameters were fixed based on the non-stationarity values, aligning with similar datasets from the Benchmark set. When an exchange setting was provided, we used it; otherwise, we followed the order of Weather and ETTm1, as specified by the Time Series Library (Wu et al., 2023; Wang et al., 2024c). For Hipeen, the learning rate was fixed at 0.001, with M=0.25 and H=10 across all cases.

# E Results in Details

## E.1 Controlled datasets

We present a detailed overview of the experimental results obtained on the controlled datasets. Figures 12 and 14 present the extended visualizations of the time series ground truth and model predictions, following the initial overview shown in Figure 1 of the main text. Specifically, Figure 12 illustrates predictions on the Sine wave dataset when the look-back window corresponds to the ascending, plateau, and descending phases of a long-period sine wave. Accurately forecasting such long-term patterns—especially those that extend beyond the look-back window—requires a solid understanding of the global shape of the time series. Notably, only Hipeen successfully captures the long-term trend of the time series, whereas the baseline models clearly fail to represent the global shape. In the Exponential and Threshold tasks as well, Figure 14

Table 8: Hyperparameter search results for each dataset and horizon length.

| Tasks | Dataset | Horizon | Learning Rate | M&H |
|---|---|---|---|---|
| | Sine wave | {96, 192, 336, 720} | 0.001 | (0.25,10) |
| Scenarios | Exponentials | {96, 192, 336, 720} | 0.001 | (0.25,10) |
| | Threshold | {96, 192, 336, 720} | 0.001 | (0.25,10) |
| | ETTh1 | 96 | 0.001 | (1,8) |
| | | 192 | 0.001 | (1,8) |
| | | 336 | 0.001 | (1,8) |
| | | 720 | 0.001 | (1,8) |
| | ETTh2 | 96 | 0.0005 | (0.5,9) |
| | | 192 | 0.0005 | (0.5,9) |
| | | 336 | 0.0005 | (0.5,9) |
| | | 720 | 0.0005 | (1,8) |
| | ETTm1 | 96 | 0.001 | (1,8) |
| | | 192 | 0.001 | (1,8) |
| | | 336 | 0.001 | (1,8) |
| | | 720 | 0.001 | (1,8) |
| | ETTm2 | 96 | 0.0005 | (0.5,9) |
| | | 192 | 0.001 | (0.5,9) |
| | | 336 | 0.0005 | (0.25,10) |
| Benchmark | | 720 | 0.0005 | (0.25,10) |
| Datasets | Weather | 96 | 0.001 | (0.5,9) |
| | | 192 | 0.002 | (0.5,9) |
| | | 336 | 0.002 | (0.5,9) |
| | | 720 | 0.002 | (0.5,9) |
| | Solar-Energy | 96 | 0.0005 | (1,8) |
| | | 192 | 0.0005 | (1,8) |
| | | 336 | 0.0005 | (1,8) |
| | | 720 | 0.001 | (1,8) |
| | Electricity | 96 | 0.001 | (1,8) |
| | | 192 | 0.0005 | (1,8) |
| | | 336 | 0.0005 | (1,8) |
| | | 720 | 0.001 | (1,8) |
| | Traffic | 96 | 0.001 | (1,8) |
| | | 192 | 0.001 | (1,8) |
| | | 336 | 0.001 | (1,8) |
| | | 720 | 0.001 | (1,8) |
| | Exchange | 96 | 0.0005 | (0.25,10) |
| | | 192 | 0.001 | (0.25,10) |
| | | 336 | 0.001 | (0.25,10) |
| | | 720 | 0.0005 | (0.25,10) |
| All Stock | Datasets | {12, 24, 48, 96} | 0.001 | (0.25,10) |

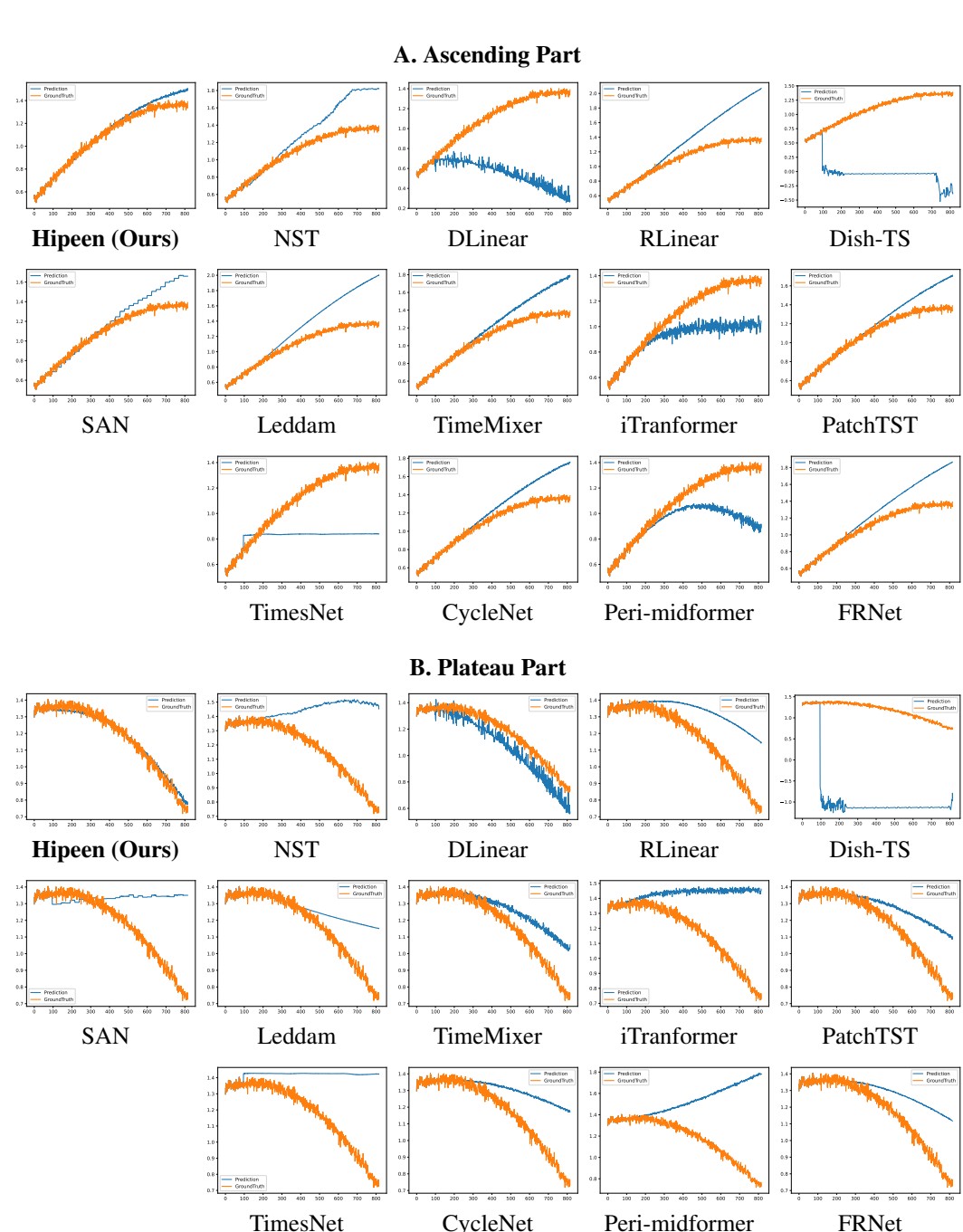

Figure 12: **(Part 1)** `Sine wave` dataset, 4k-5k period, 720 horizon. Performance comparison across three phases of long-period sine wave: (A) Ascending, (B) Plateau, and (C) Descending. Each row shows results from Hipeen and baseline models. Orange line is ground truth and blue line is model prediction. Cont'd to Table 13

**C. Descending Part**

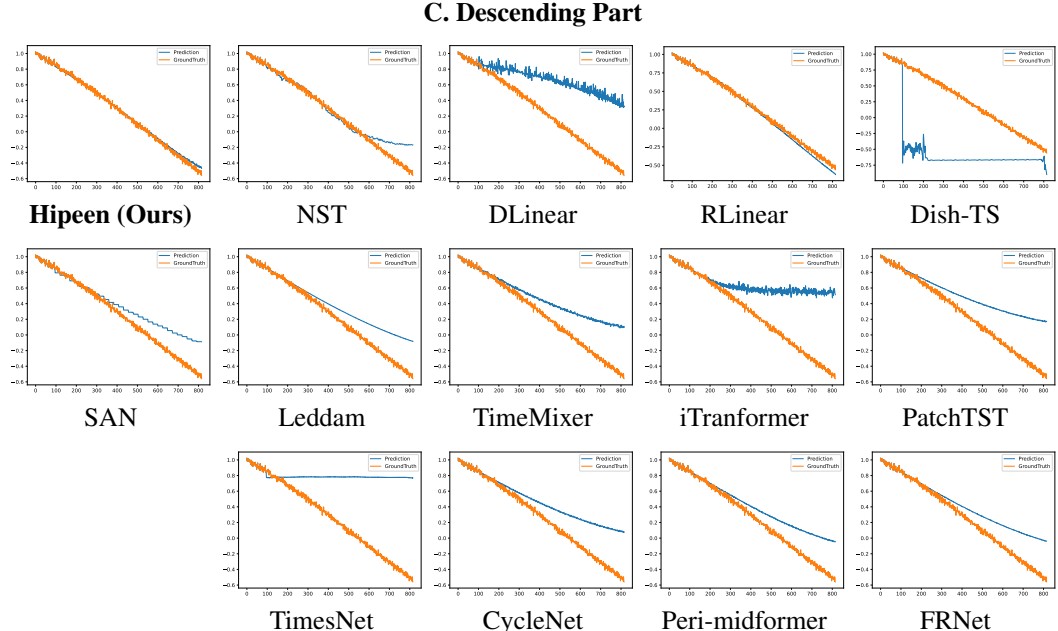

Figure 13: (**Part 2** of Table 12). `Sine wave` dataset, 4k-5k period, 720 horizon. (C) Descending. Each row shows results from Hipeen and baseline models. Orange line is ground truth and blue line is model prediction.

shows that Hipeen achieves more accurate predictions than the baselines by effectively leveraging both gradient and absolute value cues.

Table 9 and 10 presents the full results corresponding to Table 1 in the main text. In addition to the Sine wave datasets (2k–3k and 3k–4k), it includes all horizon values in {96,192,336,720}. Consistent with the main-text results, Hipeen outperforms the baseline models, demonstrating superior performance on our realistic controlled datasets. In addition, the standard deviations (std) across three random seeds for each experiment are reported in Table 11 and 12. Hipeen shows a lower standard deviation than TimeMixer, indicating more stable performance.

### E.2 REAL-WORLD STOCK DATASETS

#### E.2.1 *S&P 500* DATASETS

Table 2 reports the forecasting performance on the *S&P500* dataset. In the extended Table 13, for brevity, we present only the top 30 stocks in alphabetical order from the full set of 500. The proposed Hipeen consistently achieved superior performance compared to both classical linear approaches and recent transformer-based architectures. In particular, Hipeen delivered the lowest error values across the majority of stocks, with especially strong robustness on highly volatile equities such as AMD, AMAT, and AES, where traditional baselines (e.g., DLinear, RLinear) exhibited significant error inflation. While transformer variants such as iTransformer and PatchTST occasionally performed competitively on technology-related stocks (e.g., AAPL, ADBE, AMZN), their results were less stable across the broader set. Simpler models like DLinear showed reasonable accuracy on stable, low-volatility stocks (e.g., ABT, ADP, AMGN), but their generalization deteriorated sharply under complex dynamics. Overall, these results highlight the advantage of Hipeen, demonstrating both strong predictive accuracy and greater consistency across heterogeneous stock behaviors, making it a more reliable solution for large-scale financial time series forecasting.

#### E.2.2 *Nifty 50* DATASETS

We extended the prediction horizons in the *Nifty50* experiments to {12, 24, 48, 96}. We also simulated actual trading based on the model predictions and expanded the evaluation to include

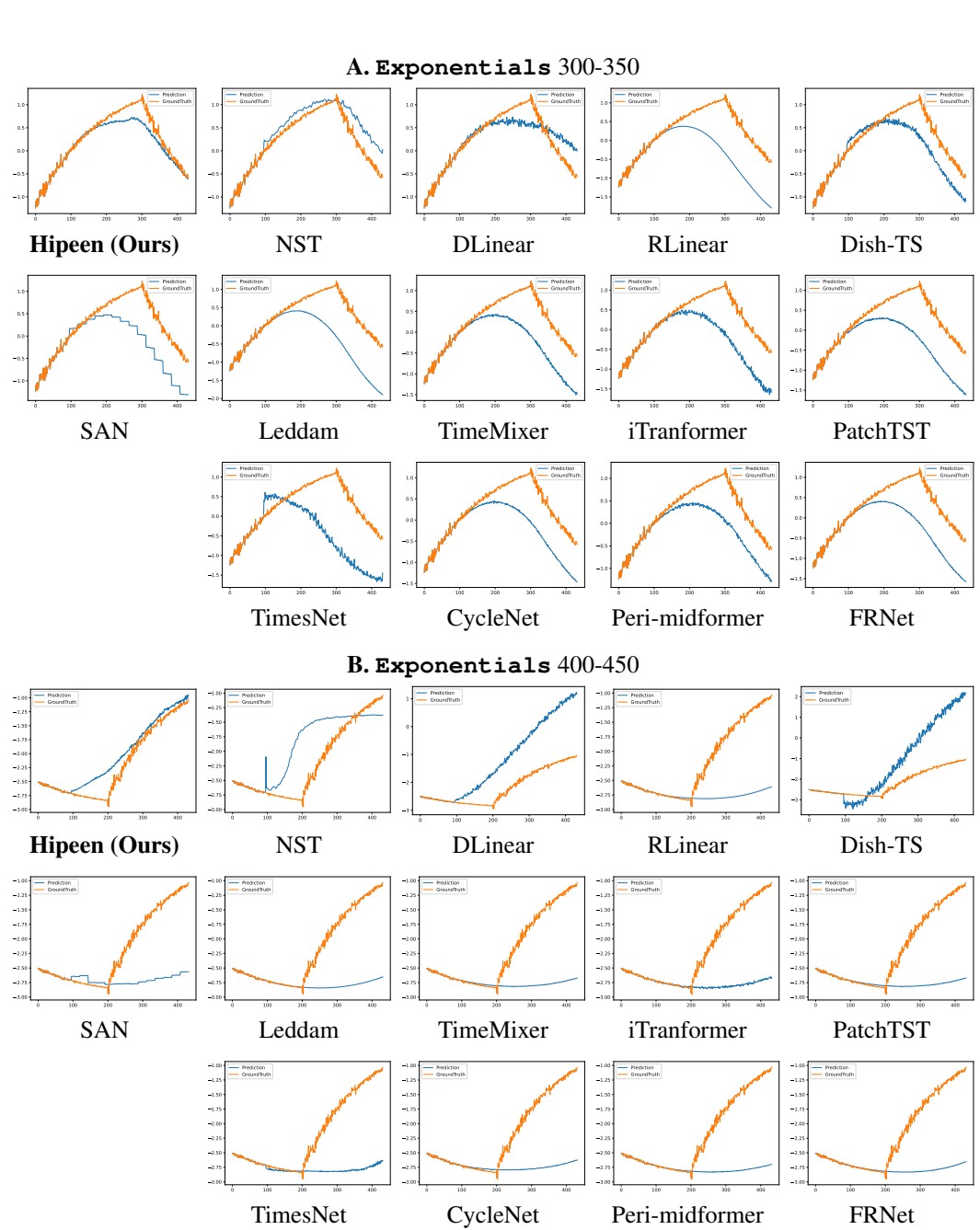

Figure 14: (**Part 1**). Prediction patterns of Hipeen and the baselines in the transition regions of time series under the `Exponentials` and `Threshold` scenario tasks. Orange indicates the ground truth, and blue represents the model predictions. (Cont'd in Table 15)

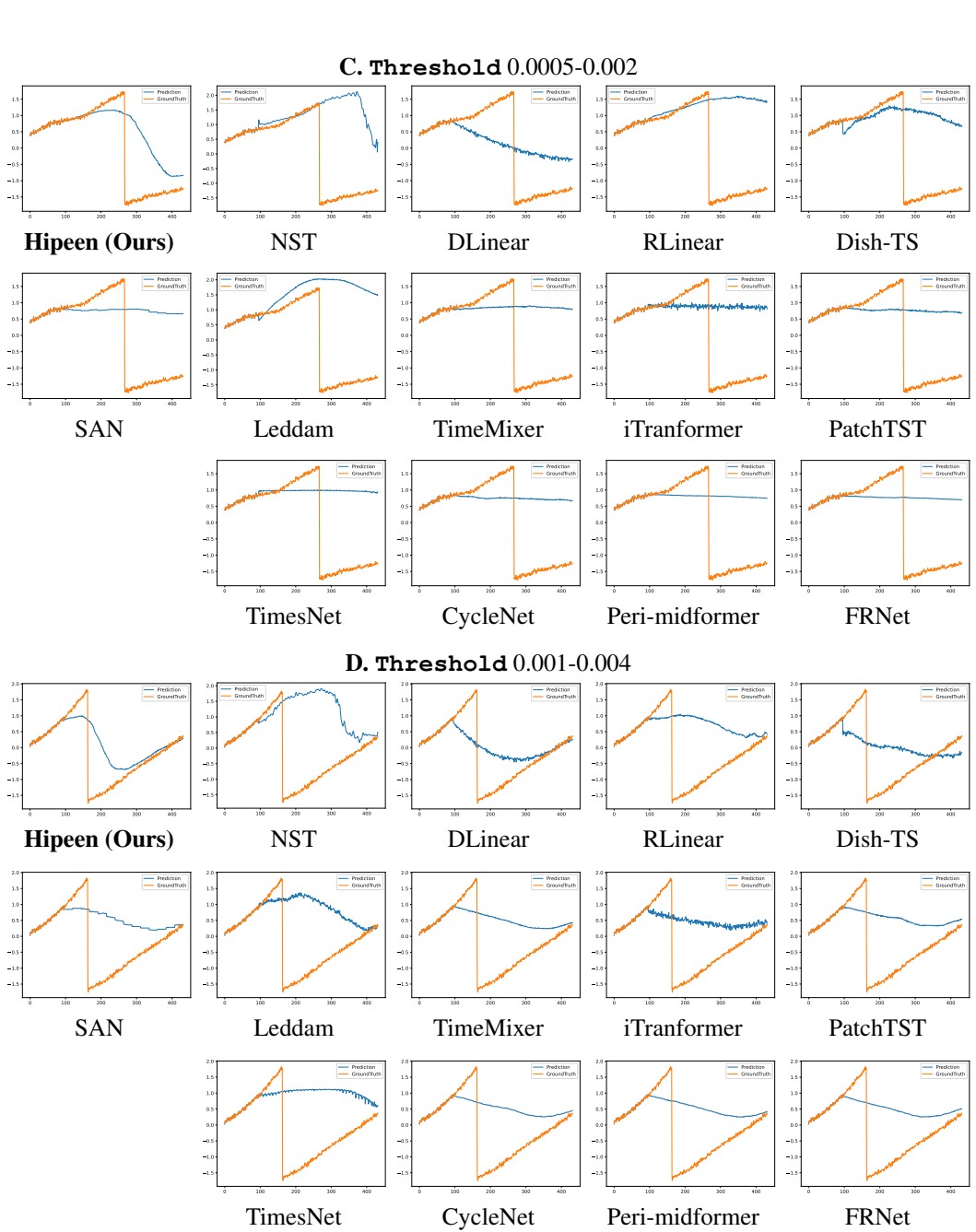

Figure 15: (**Part 2** of Table 14.) Prediction patterns of Hipeen and the baselines in the transition regions of time series under the `Exponentials` and `Threshold` scenario tasks. Orange indicates the ground truth, and blue represents the model predictions.

advanced, return-related metrics. This experiment was conducted using the following 12 stock datasets, listed in alphabetical order, as described in Appendix B.1.2: `ADANIPORTS`, `BAJAJ-AUTO`, `HDFC`, `HEROMOTOCO`, `HINDALCO`, `LT`, `MARUTI`, `NTPC`, `POWERGRID`, `TATASTEEL`, `TECHM`, and `TITAN`. For details on the inclusion criteria, please refer to Appendix B.1.2.

In addition to the MAE, MAPE, and RMSE used in TSF benchmarks, we compute the profit generated by executing trades based on the model's predictions. Trading is conducted as follows: for a model with horizon length $K$, the model predicts the next $K$ future values at each timestep. If the predicted value at $t + K$ is greater than the current value, we open a long position with $1/K$ of the capital for $K$ days. Conversely, if the predicted value at $t + K$ is lower than the current value, we open a short position with $1/K$ of the capital for $K$ days. Detailed definitions of each metric are provided below.

**Mean Absolute Error (MAE).** MAE measures the average magnitude of prediction errors. Lower values indicate better predictive accuracy.

$$\text{MAE} = \frac{1}{N} \sum_{t=1}^{N} |y_t - \hat{y}_t|.$$

**Mean Absolute Percentage Error (MAPE).** MAPE evaluates the relative prediction error as a percentage of the true value and is scale-independent. For stability, datasets where baseline models produced a MAPE larger than 10 were excluded from analysis.

$$\text{MAPE} = \frac{100}{N} \sum_{t=1}^{N} \left| \frac{y_t - \hat{y}_t}{y_t} \right|.$$

**Root Mean Squared Error (RMSE).** RMSE penalizes large prediction errors by squaring the deviations before averaging. Lower values are preferred.

$$\text{RMSE} = \sqrt{\frac{1}{N} \sum_{t=1}^{N} (y_t - \hat{y}_t)^2}.$$

**Revenue (Cumulative Return).** Revenue represents the cumulative return obtained from a trading strategy built on model predictions. Higher values indicate better performance.

$$\text{Revenue} = \prod_{t=1}^{N} (1 + r_t) - 1,$$

where $r_t$ denotes the daily strategy return.

**Drawdown.** Average drawdown measures the mean decline from the historical peak of the equity curve, reflecting the overall risk exposure of the strategy. Lower values are better.

$$\text{AvgDD} = \frac{1}{K} \sum_{k=1}^{K} \left( \frac{P_{t_k} - P_{t_k}^*}{P_{t_k}^*} \right),$$

where $P_t$ is the equity curve and $P_t^* = \max_{\tau \leq t} P_\tau$.

**Sharpe Ratio.** The Sharpe ratio quantifies the excess return per unit of total volatility. Higher values indicate better risk-adjusted performance.

$$\text{Sharpe} = \frac{\mathbb{E}[r_t - r_f]}{\sigma(r_t)},$$

assuming a risk-free rate $r_f = 0$.

**Sortino Ratio.** The Sortino ratio is similar to the Sharpe ratio but uses downside volatility instead of total volatility, penalizing only negative deviations. Higher values are preferred.

$$\text{Sortino} = \frac{\mathbb{E}[r_t - r_f]}{\sigma(r_t \,|\, r_t < 0)}.$$

**Calmar Ratio.** The Calmar ratio measures the annualized return relative to the maximum drawdown, capturing the trade-off between growth and extreme losses. Higher values are better.

$$\text{Calmar} = \frac{\text{AnnualReturn}}{|\text{MaxDrawdown}|}.$$

Tables 14, 15, and 16 present extended results for the MAE, MAPE, and RMSE experiments. Table 17 reports the results of the trading simulation, evaluated using Revenue, Drawdown, Sharpe Ratio, Sortino Ratio, and Calmar Ratio.

The results show that Hipeen achieves the best predictive performance on real-world stock datasets and attains state-of-the-art performance even in real-world trading scenarios based on these predictions.

Another noteworthy observation is that stock forecasting models such as SMamba and STF exhibit relatively low performance on the standard forecasting metrics (MAE, MAPE, RMSE), yet achieve strong results in the trading simulations.

### E.3 BENCHMARK DATASETS

Table 18 provides the complete results corresponding to Table 4 in the main text, including all horizon values in {96,192,336,720}. Only the benchmark dataset experiment was obtained using a prototype estimation approach, where Q was not stored during training and $V_{est}$ was computed by assuming that $v_h$ equals $v_{est}$ in each estimation step. In the final version of the manuscript, these results will be updated using the latest estimation method that incorporates the stored $Q$ values.

### E.4 ANALYSIS

Figure 16 presents the full results corresponding to Figure 7 in the main text. We analyzed how performance changes with varying hierarchy levels $H$ on three benchmark datasets: `ETTh1`, `Weather`, and `Exchange`. When fixing $r_H$, performance generally declined as $H$ decreased. In contrast, when fixing $r_1$, performance was maintained or even improved up to a certain point, after which it sharply deteriorated.

Table 19 provides the full results corresponding to Table 5 in the main text. Similarly, we conducted experiments on `ETTh1`, `Weather`, and `Exchange` datasets to evaluate the impact of varying the bias term added to $\theta$. Our results indicate that adding random angular bias to both the channel dimension $N$ and the hierarchy dimension $H$ is crucial for improving performance.

### E.5 COMPUTATIONAL COST

As shown in Table 20, Hipeen demonstrates strong computational efficiency, ranking among the top methods across both runtime and memory usage. Despite incorporating an ensemble dimension, Hipeen maintains lightweight training (5.1 ms per step) and inference (3.3 ms per step), with VRAM consumption comparable to the most efficient baselines. This efficiency advantage arises from its design, which scales batch size without introducing additional parameters, allowing Hipeen to retain near-linear efficiency while offering substantially stronger predictive accuracy. In contrast, transformer-based architectures incur significantly higher computational costs, highlighting Hipeen's favorable trade-off between scalability and accuracy for large-scale time series forecasting.

## F LLM USAGE CLARIFICATION

During the preparation of this manuscript, we utilized Google's Gemini (https://gemini.google.com) and OpenAI's ChatGPT (https://chat.openai.com), both Large Language Models, for proofreading and refining the writing. Our interactions with these tools were iterative and limited solely to enhancing the clarity and quality of the text. We confirm that the LLMs functioned only as assistive tools and did not contribute to the research ideas, experimental design, or data analysis in this paper. The final scientific content and all conclusions remain entirely the responsibility of the authors.

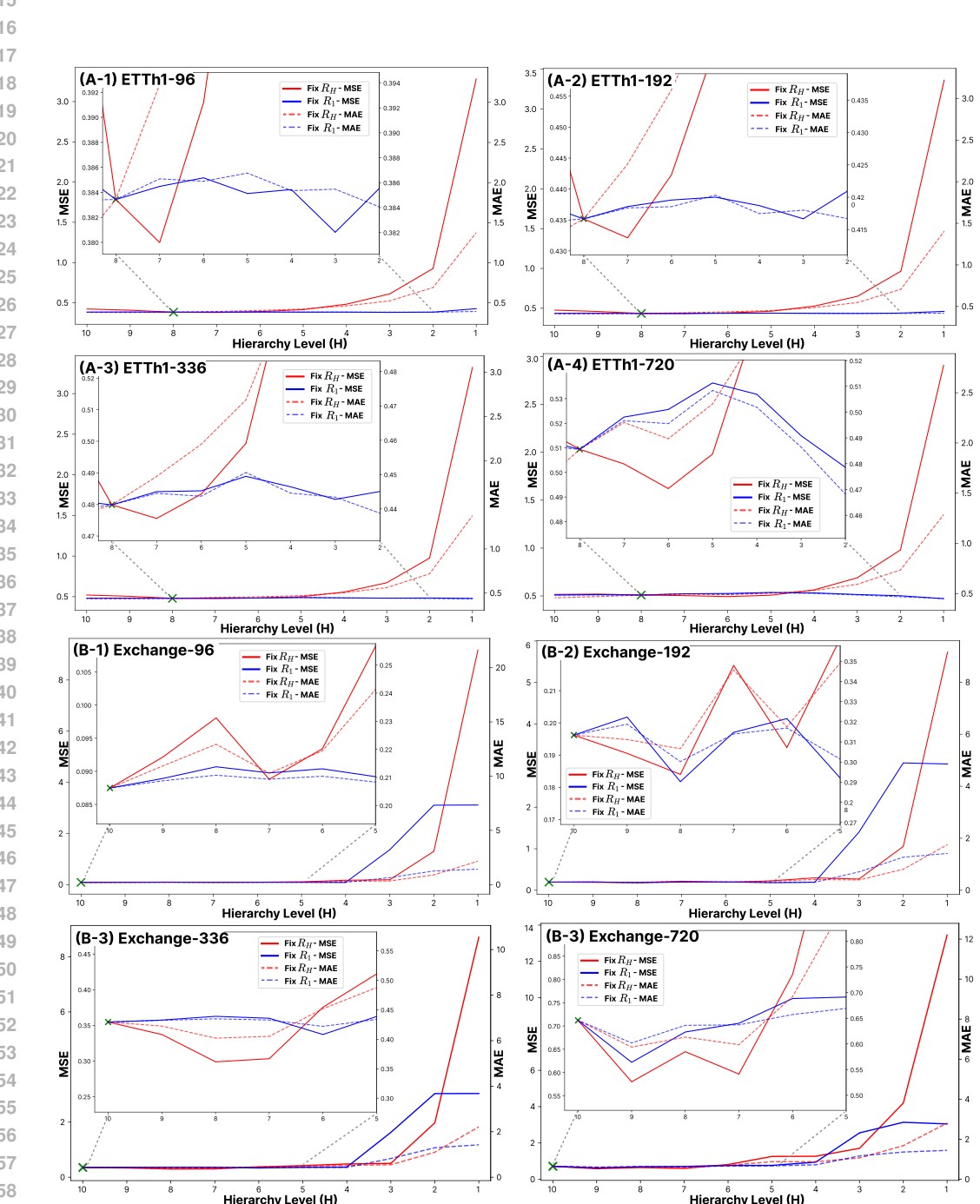

Figure 16: continued in the next figure (1/2)

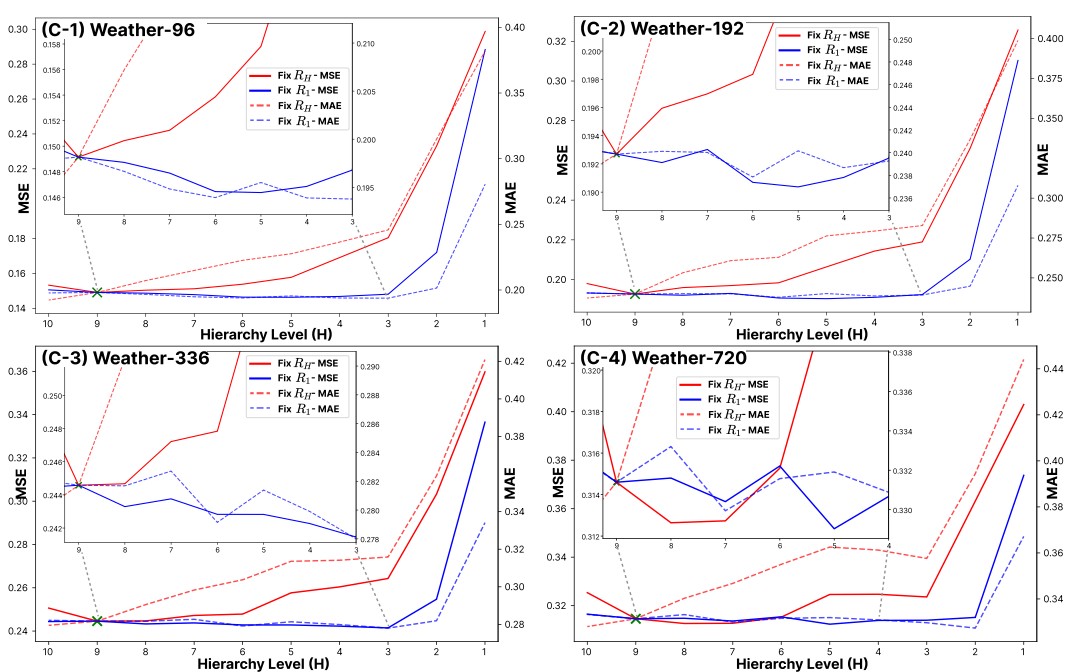

Figure 6 (continued; 2/2): We evaluated performance on the ETTh1 (A), Exchange (B), and Weather (C) datasets across horizons of 96, 192, 336, and 720 by varying the hierarchy level $H$. The red line indicates the case where $r_H$ is fixed, and the blue line indicates the case where $r_1$ is fixed. Solid lines represent MSE, while dashed lines represent MAE. The model consistently maintained high performance over a relatively wide range of $H$, while performance degradation was observed when $H$ became too small.

Table 9: Full results for the scenario datasets–Exponentials, Threshold. We compare Hippen with extensive competitive baseline models under different horizon lengths using 3 random seeds. *Avg* is averaged from all four horizon lengths: {96, 192, 336, 720}.

| Dataset | Pred_len | Hippen MSE | Hippen MAE | NST MSE | NST MAE | DLinear MSE | DLinear MAE | RLinear MSE | RLinear MAE | Dish-TS MSE | Dish-TS MAE | SAN MSE | SAN MAE | Leddam MSE | Leddam MAE | TimeMixer MSE | TimeMixer MAE | iTransformer MSE | iTransformer MAE | PatchTST MSE | PatchTST MAE | TiDE MSE | TiDE MAE | TimesNet MSE | TimesNet MAE | CycleNet MSE | CycleNet MAE | Peri-mid MSE | Peri-mid MAE | FRNet MSE | FRNet MAE |
|---|---|---|---|---|---|---|---|---|---|---|---|---|---|---|---|---|---|---|---|---|---|---|---|---|---|---|---|---|---|---|---|
| Exp. 300-350 | 96 | 0.079 | 0.176 | 0.231 | 0.279 | 0.276 | 0.363 | 0.123 | 0.199 | 0.876 | 0.684 | 0.090 | 0.210 | 0.095 | 0.183 | 0.089 | 0.153 | 0.140 | 0.237 | 0.094 | 0.175 | 0.129 | 0.208 | 0.182 | 0.302 | 0.101 | 0.172 | 0.105 | 0.179 | 0.093 | 0.161 |
| | 192 | 0.291 | 0.366 | 0.459 | 0.445 | 1.056 | 0.675 | 0.516 | 0.479 | 2.040 | 1.132 | 0.435 | 0.460 | 0.394 | 0.420 | 0.436 | 0.411 | 0.493 | 0.502 | 0.445 | 0.437 | 0.527 | 0.487 | 0.525 | 0.550 | 0.470 | 0.443 | 0.486 | 0.458 | 0.442 | 0.417 |
| | 336 | 0.687 | 0.598 | 0.790 | 0.620 | 2.343 | 0.984 | 1.053 | 0.747 | 3.158 | 1.394 | 0.796 | 0.668 | 0.769 | 0.649 | 0.928 | 0.665 | 0.925 | 0.738 | 0.918 | 0.676 | 1.061 | 0.757 | 1.052 | 0.808 | 0.971 | 0.700 | 1.014 | 0.722 | 0.928 | 0.668 |
| | 720 | 0.689 | 0.611 | 0.785 | 0.673 | 1.632 | 0.820 | 0.840 | 0.678 | 2.508 | 1.270 | 0.608 | 0.587 | 0.637 | 0.598 | 0.759 | 0.633 | 0.757 | 0.665 | 0.735 | 0.623 | 0.831 | 0.675 | 0.898 | 0.746 | 0.788 | 0.650 | 0.852 | 0.686 | 0.791 | 0.654 |
| | Avg. | 0.436 | 0.438 | 0.566 | 0.505 | 1.327 | 0.710 | 0.633 | 0.526 | 2.146 | 1.120 | 0.482 | 0.481 | 0.474 | 0.462 | 0.553 | 0.466 | 0.579 | 0.536 | 0.548 | 0.478 | 0.637 | 0.532 | 0.664 | 0.601 | 0.583 | 0.491 | 0.614 | 0.511 | 0.564 | 0.475 |
| Exp. 400-450 | 96 | 0.047 | 0.134 | 0.081 | 0.166 | 0.114 | 0.244 | 0.074 | 0.134 | 0.402 | 0.421 | 0.052 | 0.148 | 0.046 | 0.109 | 0.050 | 0.102 | 0.068 | 0.145 | 0.054 | 0.120 | 0.077 | 0.141 | 0.088 | 0.175 | 0.062 | 0.118 | 0.065 | 0.121 | 0.057 | 0.110 |
| | 192 | 0.101 | 0.213 | 0.246 | 0.299 | 0.367 | 0.448 | 0.322 | 0.330 | 0.719 | 0.653 | 0.209 | 0.317 | 0.261 | 0.304 | 0.267 | 0.281 | 0.295 | 0.330 | 0.274 | 0.305 | 0.327 | 0.336 | 0.351 | 0.392 | 0.298 | 0.308 | 0.308 | 0.317 | 0.284 | 0.296 |
| | 336 | 0.204 | 0.320 | 0.480 | 0.453 | 0.863 | 0.665 | 0.818 | 0.611 | 2.256 | 1.145 | 0.627 | 0.575 | 0.708 | 0.577 | 0.732 | 0.553 | 0.809 | 0.632 | 0.751 | 0.581 | 0.827 | 0.617 | 0.890 | 0.692 | 0.787 | 0.600 | 0.812 | 0.602 | 0.752 | 0.570 |
| | 720 | 0.380 | 0.471 | 0.681 | 0.586 | 1.181 | 0.784 | 1.070 | 0.764 | 2.475 | 1.226 | 0.814 | 0.676 | 0.956 | 0.722 | 0.998 | 0.717 | 1.062 | 0.785 | 0.978 | 0.713 | 1.064 | 0.764 | 1.162 | 0.834 | 1.021 | 0.737 | 1.113 | 0.776 | 1.054 | 0.746 |
| | Avg. | 0.183 | 0.284 | 0.372 | 0.376 | 0.631 | 0.535 | 0.571 | 0.460 | 1.463 | 0.861 | 0.425 | 0.429 | 0.493 | 0.428 | 0.512 | 0.413 | 0.559 | 0.473 | 0.514 | 0.430 | 0.574 | 0.464 | 0.623 | 0.523 | 0.542 | 0.441 | 0.575 | 0.454 | 0.537 | 0.431 |
| Exp. 500-550 | 96 | 0.034 | 0.105 | 0.144 | 0.211 | 0.079 | 0.197 | 0.039 | 0.083 | 0.107 | 0.237 | 0.031 | 0.110 | 0.032 | 0.081 | 0.028 | 0.064 | 0.048 | 0.110 | 0.029 | 0.079 | 0.041 | 0.087 | 0.056 | 0.131 | 0.032 | 0.072 | 0.033 | 0.073 | 0.029 | 0.066 |
| | 192 | 0.112 | 0.189 | 0.489 | 0.388 | 0.290 | 0.384 | 0.181 | 0.210 | 0.286 | 0.424 | 0.146 | 0.240 | 0.160 | 0.204 | 0.150 | 0.178 | 0.184 | 0.225 | 0.155 | 0.205 | 0.185 | 0.214 | 0.227 | 0.301 | 0.165 | 0.191 | 0.168 | 0.197 | 0.155 | 0.181 |
| | 336 | 0.239 | 0.313 | 0.943 | 0.591 | 0.726 | 0.627 | 0.616 | 0.481 | 0.676 | 0.663 | 0.418 | 0.453 | 0.564 | 0.473 | 0.552 | 0.437 | 0.618 | 0.503 | 0.561 | 0.467 | 0.624 | 0.492 | 0.709 | 0.597 | 0.584 | 0.461 | 0.594 | 0.470 | 0.563 | 0.437 |
| | 720 | 0.570 | 0.563 | 1.633 | 0.926 | 1.412 | 0.930 | 1.267 | 0.842 | 1.573 | 1.020 | 0.974 | 0.745 | 1.171 | 0.800 | 1.201 | 0.805 | 1.249 | 0.859 | 1.193 | 0.811 | 1.264 | 0.844 | 1.362 | 0.916 | 1.238 | 0.829 | 1.289 | 0.849 | 1.212 | 0.812 |
| | Avg. | 0.238 | 0.293 | 0.802 | 0.529 | 0.627 | 0.534 | 0.526 | 0.404 | 0.660 | 0.586 | 0.392 | 0.387 | 0.482 | 0.390 | 0.483 | 0.371 | 0.524 | 0.424 | 0.485 | 0.390 | 0.529 | 0.409 | 0.589 | 0.486 | 0.505 | 0.388 | 0.521 | 0.397 | 0.490 | 0.374 |
| Thr. 5-20(e-4) | 96 | 0.245 | 0.211 | 0.781 | 0.419 | 0.502 | 0.442 | 0.611 | 0.374 | 0.510 | 0.487 | 0.480 | 0.427 | 0.641 | 0.382 | 0.622 | 0.368 | 0.619 | 0.357 | 0.610 | 0.381 | 0.615 | 0.380 | 0.673 | 0.442 | 0.606 | 0.372 | 0.605 | 0.362 | 0.604 | 0.353 |
| | 192 | 0.331 | 0.295 | 1.274 | 0.668 | 0.696 | 0.609 | 1.143 | 0.670 | 0.665 | 0.593 | 0.971 | 0.712 | 1.182 | 0.680 | 1.183 | 0.658 | 1.155 | 0.630 | 1.151 | 0.679 | 1.150 | 0.681 | 1.259 | 0.753 | 1.139 | 0.656 | 1.141 | 0.663 | 1.137 | 0.663 |
| | 336 | 0.424 | 0.396 | 1.769 | 0.862 | 0.752 | 0.668 | 1.762 | 1.008 | 0.887 | 0.714 | 2.063 | 1.090 | 1.796 | 1.008 | 1.826 | 0.997 | 1.768 | 0.995 | 1.771 | 1.010 | 1.764 | 1.010 | 1.892 | 1.086 | 1.761 | 0.999 | 1.764 | 1.006 | 1.758 | 1.003 |
| | 720 | 0.575 | 0.515 | 1.825 | 0.912 | 0.806 | 0.712 | 1.754 | 1.043 | 1.117 | 0.893 | 1.626 | 1.068 | 1.806 | 1.047 | 1.793 | 1.031 | 1.766 | 1.057 | 1.760 | 1.046 | 1.758 | 1.048 | 1.924 | 1.126 | 1.756 | 1.046 | 1.807 | 1.059 | 1.752 | 1.042 |
| | Avg. | 0.394 | 0.354 | 1.412 | 0.715 | 0.689 | 0.608 | 1.317 | 0.774 | 0.795 | 0.672 | 1.285 | 0.824 | 1.356 | 0.779 | 1.356 | 0.764 | 1.327 | 0.760 | 1.323 | 0.779 | 1.322 | 0.779 | 1.437 | 0.852 | 1.315 | 0.768 | 1.329 | 0.772 | 1.313 | 0.765 |
| Thr. 10-40(e-4) | 96 | 0.314 | 0.306 | 1.754 | 0.800 | 0.652 | 0.591 | 1.068 | 0.667 | 0.820 | 0.691 | 0.884 | 0.707 | 1.150 | 0.687 | 1.055 | 0.649 | 1.076 | 0.656 | 1.058 | 0.674 | 1.073 | 0.672 | 1.250 | 0.788 | 1.063 | 0.652 | 1.063 | 0.654 | 1.055 | 0.653 |
| | 192 | 0.472 | 0.452 | 2.767 | 1.165 | 0.699 | 0.643 | 1.608 | 0.992 | 1.094 | 0.786 | 1.452 | 0.995 | 1.764 | 1.019 | 1.596 | 0.972 | 1.616 | 0.985 | 1.600 | 0.990 | 1.613 | 0.993 | 1.851 | 1.093 | 1.598 | 0.981 | 1.606 | 0.982 | 1.592 | 0.979 |
| | 336 | 0.632 | 0.576 | 2.144 | 1.072 | 0.750 | 0.679 | 1.538 | 0.968 | 1.083 | 0.853 | 1.397 | 0.981 | 1.677 | 0.995 | 1.532 | 0.962 | 1.562 | 0.974 | 1.536 | 0.968 | 1.543 | 0.972 | 1.733 | 1.044 | 1.530 | 0.959 | 1.548 | 0.967 | 1.533 | 0.962 |
| | 720 | 0.820 | 0.705 | 2.044 | 1.078 | 0.844 | 0.743 | 1.462 | 0.960 | 1.065 | 0.870 | 1.360 | 0.965 | 1.604 | 0.985 | 1.457 | 0.950 | 1.495 | 0.974 | 1.460 | 0.960 | 1.467 | 0.963 | 1.632 | 1.022 | 1.460 | 0.959 | 1.475 | 0.962 | 1.459 | 0.955 |
| | Avg. | 0.560 | 0.510 | 2.177 | 1.029 | 0.736 | 0.664 | 1.419 | 0.897 | 1.016 | 0.800 | 1.273 | 0.912 | 1.549 | 0.922 | 1.410 | 0.883 | 1.437 | 0.897 | 1.413 | 0.898 | 1.424 | 0.900 | 1.617 | 0.987 | 1.411 | 0.888 | 1.423 | 0.891 | 1.410 | 0.887 |
| Thr. 15-60(e-4) | 96 | 0.412 | 0.403 | 1.760 | 0.887 | 0.644 | 0.606 | 1.261 | 0.815 | 0.815 | 0.705 | 1.144 | 0.832 | 1.235 | 0.793 | 1.243 | 0.791 | 1.253 | 0.819 | 1.247 | 0.806 | 1.268 | 0.823 | 1.430 | 0.911 | 1.245 | 0.794 | 1.257 | 0.809 | 1.244 | 0.790 |
| | 192 | 0.577 | 0.540 | 1.862 | 0.993 | 0.703 | 0.658 | 1.260 | 0.842 | 0.943 | 0.842 | 1.166 | 0.860 | 1.260 | 0.819 | 1.244 | 0.823 | 1.254 | 0.842 | 1.241 | 0.827 | 1.267 | 0.848 | 1.372 | 0.905 | 1.240 | 0.820 | 1.254 | 0.835 | 1.242 | 0.818 |
| | 336 | 0.680 | 0.620 | 1.641 | 0.952 | 0.750 | 0.694 | 1.137 | 0.813 | 0.983 | 0.840 | 1.072 | 0.828 | 1.135 | 0.798 | 1.117 | 0.790 | 1.132 | 0.815 | 1.116 | 0.799 | 1.144 | 0.817 | 1.238 | 0.861 | 1.111 | 0.791 | 1.134 | 0.809 | 1.109 | 0.790 |
| | 720 | 0.825 | 0.724 | 1.589 | 0.990 | 0.852 | 0.763 | 1.205 | 0.872 | 1.002 | 0.857 | 1.155 | 0.877 | 1.200 | 0.864 | 1.183 | 0.854 | 1.209 | 0.880 | 1.193 | 0.866 | 1.213 | 0.876 | 1.314 | 0.919 | 1.183 | 0.855 | 1.201 | 0.867 | 1.193 | 0.866 |
| | Avg. | 0.624 | 0.572 | 1.713 | 0.956 | 0.737 | 0.680 | 1.216 | 0.836 | 0.936 | 0.801 | 1.134 | 0.849 | 1.207 | 0.818 | 1.197 | 0.814 | 1.212 | 0.839 | 1.199 | 0.825 | 1.223 | 0.841 | 1.339 | 0.899 | 1.195 | 0.815 | 1.211 | 0.830 | 1.197 | 0.816 |

Table 10: Full results for the scenario datasets—Sine wave. We compare Hippen with extensive competitive baseline models under different horizon lengths using 3 random seeds. *Avg* is averaged from all four horizon lengths: {96, 192, 336, 720}.

| Dataset | Pred_len | Hippen MSE | Hippen MAE | NST MSE | NST MAE | DLinear MSE | DLinear MAE | RLinear MSE | RLinear MAE | Dish-TS MSE | Dish-TS MAE | SAN MSE | SAN MAE | Leddam MSE | Leddam MAE | TimeMixer MSE | TimeMixer MAE | iTransformer MSE | iTransformer MAE | PatchTST MSE | PatchTST MAE | TiDE MSE | TiDE MAE | TimesNet MSE | TimesNet MAE | CycleNet MSE | CycleNet MAE | Peri-mid MSE | Peri-mid MAE | FRNet MSE | FRNet MAE |
|---|---|---|---|---|---|---|---|---|---|---|---|---|---|---|---|---|---|---|---|---|---|---|---|---|---|---|---|---|---|---|---|
| Sine 2k-3k | 96 | 0.001 | 0.025 | 0.003 | 0.044 | 0.006 | 0.057 | 0.001 | 0.028 | 0.159 | 0.237 | 0.001 | 0.024 | 0.001 | 0.027 | 0.001 | 0.027 | 0.002 | 0.032 | 0.002 | 0.032 | 0.002 | 0.030 | 0.013 | 0.088 | 0.001 | 0.027 | 0.002 | 0.032 | 0.001 | 0.027 |
| | 192 | 0.002 | 0.031 | 0.013 | 0.083 | 0.035 | 0.140 | 0.006 | 0.056 | 0.627 | 0.465 | 0.004 | 0.045 | 0.005 | 0.049 | 0.004 | 0.046 | 0.008 | 0.067 | 0.006 | 0.056 | 0.007 | 0.059 | 0.072 | 0.206 | 0.005 | 0.049 | 0.008 | 0.065 | 0.005 | 0.050 |
| | 336 | 0.003 | 0.043 | 0.051 | 0.154 | 0.140 | 0.283 | 0.034 | 0.128 | 0.665 | 0.567 | 0.028 | 0.111 | 0.025 | 0.106 | 0.021 | 0.095 | 0.057 | 0.167 | 0.026 | 0.109 | 0.037 | 0.134 | 0.226 | 0.371 | 0.024 | 0.104 | 0.050 | 0.156 | 0.024 | 0.105 |
| | 720 | 0.022 | 0.097 | 0.449 | 0.482 | 0.453 | 0.539 | 0.390 | 0.436 | 0.779 | 0.734 | 0.367 | 0.406 | 0.279 | 0.357 | 0.214 | 0.295 | 0.537 | 0.530 | 0.255 | 0.327 | 0.395 | 0.441 | 0.924 | 0.762 | 0.257 | 0.333 | 0.643 | 0.558 | 0.228 | 0.309 |
| | Avg. | 0.007 | 0.049 | 0.129 | 0.191 | 0.159 | 0.255 | 0.108 | 0.162 | 0.557 | 0.500 | 0.100 | 0.147 | 0.077 | 0.135 | 0.060 | 0.116 | 0.151 | 0.199 | 0.072 | 0.131 | 0.110 | 0.166 | 0.309 | 0.357 | 0.072 | 0.128 | 0.176 | 0.203 | 0.065 | 0.123 |
| Sine 3k-4k | 96 | 0.001 | 0.022 | 0.001 | 0.026 | 0.003 | 0.041 | 0.001 | 0.024 | 0.137 | 0.261 | 0.001 | 0.022 | 0.001 | 0.023 | 0.001 | 0.024 | 0.004 | 0.026 | 0.001 | 0.026 | 0.001 | 0.025 | 0.007 | 0.061 | 0.001 | 0.024 | 0.001 | 0.024 | 0.001 | 0.024 |
| | 192 | 0.001 | 0.026 | 0.003 | 0.039 | 0.016 | 0.094 | 0.003 | 0.039 | 0.610 | 0.501 | 0.001 | 0.030 | 0.002 | 0.036 | 0.003 | 0.037 | 0.004 | 0.045 | 0.003 | 0.042 | 0.003 | 0.040 | 0.027 | 0.127 | 0.003 | 0.037 | 0.003 | 0.040 | 0.003 | 0.038 |
| | 336 | 0.002 | 0.037 | 0.011 | 0.073 | 0.069 | 0.196 | 0.013 | 0.081 | 1.134 | 0.697 | 0.007 | 0.061 | 0.011 | 0.071 | 0.012 | 0.073 | 0.023 | 0.107 | 0.012 | 0.078 | 0.013 | 0.083 | 0.095 | 0.236 | 0.011 | 0.074 | 0.016 | 0.088 | 0.011 | 0.074 |
| | 720 | 0.012 | 0.071 | 0.093 | 0.207 | 0.325 | 0.438 | 0.155 | 0.275 | 4.114 | 1.484 | 0.136 | 0.251 | 0.130 | 0.235 | 0.141 | 0.240 | 0.264 | 0.369 | 0.143 | 0.253 | 0.165 | 0.286 | 0.444 | 0.504 | 0.133 | 0.241 | 0.191 | 0.297 | 0.132 | 0.242 |
| | Avg. | 0.004 | 0.039 | 0.027 | 0.086 | 0.103 | 0.192 | 0.043 | 0.105 | 1.499 | 0.736 | 0.036 | 0.091 | 0.036 | 0.092 | 0.039 | 0.093 | 0.073 | 0.137 | 0.040 | 0.100 | 0.045 | 0.109 | 0.143 | 0.232 | 0.037 | 0.094 | 0.053 | 0.113 | 0.037 | 0.094 |
| Sine 4k-5k | 96 | 0.001 | 0.021 | 0.001 | 0.025 | 0.006 | 0.060 | 0.001 | 0.021 | 0.134 | 0.253 | 0.001 | 0.020 | 0.001 | 0.021 | 0.001 | 0.021 | 0.001 | 0.023 | 0.001 | 0.022 | 0.001 | 0.021 | 0.003 | 0.040 | 0.001 | 0.021 | 0.001 | 0.021 | 0.001 | 0.021 |
| | 192 | 0.001 | 0.022 | 0.002 | 0.036 | 0.019 | 0.109 | 0.001 | 0.029 | 1.222 | 0.773 | 0.001 | 0.025 | 0.001 | 0.027 | 0.002 | 0.030 | 0.002 | 0.034 | 0.002 | 0.031 | 0.001 | 0.029 | 0.012 | 0.085 | 0.001 | 0.029 | 0.002 | 0.032 | 0.001 | 0.029 |
| | 336 | 0.001 | 0.029 | 0.007 | 0.058 | 0.058 | 0.187 | 0.005 | 0.054 | 3.052 | 1.369 | 0.003 | 0.040 | 0.005 | 0.048 | 0.006 | 0.055 | 0.011 | 0.076 | 0.005 | 0.054 | 0.005 | 0.053 | 0.044 | 0.160 | 0.005 | 0.052 | 0.007 | 0.060 | 0.005 | 0.053 |
| | 720 | 0.005 | 0.051 | 0.063 | 0.163 | 0.261 | 0.408 | 0.059 | 0.168 | 6.119 | 2.055 | 0.043 | 0.140 | 0.058 | 0.166 | 0.071 | 0.171 | 0.114 | 0.236 | 0.060 | 0.167 | 0.061 | 0.172 | 0.221 | 0.365 | 0.059 | 0.164 | 0.095 | 0.202 | 0.061 | 0.166 |
| | Avg. | 0.002 | 0.031 | 0.018 | 0.070 | 0.086 | 0.191 | 0.017 | 0.068 | 2.632 | 1.112 | 0.012 | 0.056 | 0.016 | 0.066 | 0.020 | 0.069 | 0.032 | 0.092 | 0.017 | 0.068 | 0.017 | 0.069 | 0.070 | 0.163 | 0.016 | 0.066 | 0.026 | 0.079 | 0.017 | 0.067 |
| Sine 5k-6k | 96 | 0.001 | 0.022 | 0.001 | 0.023 | 0.004 | 0.051 | 0.001 | 0.022 | 0.050 | 0.180 | 0.001 | 0.021 | 0.001 | 0.022 | 0.001 | 0.022 | 0.001 | 0.023 | 0.001 | 0.023 | 0.001 | 0.022 | 0.001 | 0.029 | 0.001 | 0.022 | 0.001 | 0.022 | 0.001 | 0.022 |
| | 192 | 0.001 | 0.024 | 0.001 | 0.028 | 0.012 | 0.086 | 0.001 | 0.028 | 0.147 | 0.297 | 0.001 | 0.025 | 0.001 | 0.028 | 0.001 | 0.028 | 0.002 | 0.031 | 0.001 | 0.029 | 0.001 | 0.027 | 0.007 | 0.062 | 0.001 | 0.028 | 0.002 | 0.035 | 0.001 | 0.028 |
| | 336 | 0.002 | 0.029 | 0.002 | 0.037 | 0.034 | 0.141 | 0.004 | 0.048 | 0.262 | 0.402 | 0.002 | 0.036 | 0.004 | 0.048 | 0.004 | 0.048 | 0.008 | 0.066 | 0.004 | 0.050 | 0.004 | 0.046 | 0.027 | 0.119 | 0.004 | 0.048 | 0.010 | 0.074 | 0.004 | 0.049 |
| | 720 | 0.003 | 0.043 | 0.011 | 0.077 | 0.119 | 0.257 | 0.048 | 0.164 | 0.595 | 0.648 | 0.028 | 0.121 | 0.048 | 0.160 | 0.050 | 0.155 | 0.087 | 0.215 | 0.046 | 0.157 | 0.044 | 0.157 | 0.140 | 0.269 | 0.045 | 0.156 | 0.156 | 0.277 | 0.049 | 0.163 |
| | Avg. | 0.002 | 0.029 | 0.004 | 0.041 | 0.042 | 0.134 | 0.014 | 0.066 | 0.263 | 0.382 | 0.008 | 0.051 | 0.013 | 0.064 | 0.014 | 0.063 | 0.024 | 0.084 | 0.013 | 0.065 | 0.012 | 0.063 | 0.044 | 0.120 | 0.013 | 0.064 | 0.042 | 0.102 | 0.014 | 0.065 |
| Sine 6k-7k | 96 | 0.001 | 0.021 | 0.001 | 0.026 | 0.005 | 0.054 | 0.001 | 0.021 | 0.030 | 0.135 | 0.001 | 0.021 | 0.001 | 0.021 | 0.001 | 0.021 | 0.001 | 0.024 | 0.001 | 0.022 | 0.001 | 0.021 | 0.001 | 0.030 | 0.001 | 0.021 | 0.001 | 0.021 | 0.001 | 0.021 |
| | 192 | 0.001 | 0.025 | 0.002 | 0.033 | 0.012 | 0.085 | 0.001 | 0.025 | 0.039 | 0.159 | 0.001 | 0.024 | 0.001 | 0.025 | 0.001 | 0.026 | 0.001 | 0.030 | 0.001 | 0.027 | 0.001 | 0.025 | 0.005 | 0.055 | 0.001 | 0.025 | 0.001 | 0.027 | 0.001 | 0.026 |
| | 336 | 0.001 | 0.028 | 0.004 | 0.050 | 0.029 | 0.132 | 0.003 | 0.039 | 0.142 | 0.309 | 0.002 | 0.037 | 0.002 | 0.037 | 0.003 | 0.039 | 0.005 | 0.054 | 0.003 | 0.040 | 0.003 | 0.039 | 0.018 | 0.101 | 0.003 | 0.039 | 0.003 | 0.044 | 0.003 | 0.039 |
| | 720 | 0.003 | 0.042 | 0.021 | 0.101 | 0.104 | 0.245 | 0.025 | 0.114 | 0.828 | 0.723 | 0.015 | 0.088 | 0.023 | 0.109 | 0.028 | 0.109 | 0.052 | 0.161 | 0.026 | 0.118 | 0.025 | 0.116 | 0.100 | 0.241 | 0.024 | 0.113 | 0.071 | 0.184 | 0.025 | 0.114 |
| | Avg. | 0.001 | 0.029 | 0.007 | 0.053 | 0.037 | 0.129 | 0.007 | 0.050 | 0.260 | 0.331 | 0.005 | 0.041 | 0.007 | 0.048 | 0.008 | 0.049 | 0.015 | 0.067 | 0.008 | 0.052 | 0.007 | 0.050 | 0.031 | 0.107 | 0.007 | 0.050 | 0.019 | 0.069 | 0.007 | 0.050 |

Table 11: Standard deviationss for the scenario datasets–Exponentials, `Threshold`. We compare Hippen with extensive competitive baseline models under different horizon lengths using 3 random seeds. *Avg* is averaged from all four horizon lengths: {96, 192, 336, 720}.

| Dataset | Pred_len | Hippen | | NST | | DLinear | | RLinear | | Dish-TS | | SAN | | Leddam | | TimeMixer | | iTransformer | | PatchTST | | TiDE | | TimesNet | | CycleNet | | Peri-mid | | FRNet | |
|---|---|---|---|---|---|---|---|---|---|---|---|---|---|---|---|---|---|---|---|---|---|---|---|---|---|---|---|---|---|---|---|
| | | MSE | MAE | MSE | MAE | MSE | MAE | MSE | MAE | MSE | MAE | MSE | MAE | MSE | MAE | MSE | MAE | MSE | MAE | MSE | MAE | MSE | MAE | MSE | MAE | MSE | MAE | MSE | MAE | MSE | MAE |
| Exp. 300-350 | 96 | 0.005 | 0.004 | 0.003 | 0.005 | 0.001 | 0.001 | 0.000 | 0.000 | 0.141 | 0.034 | 0.008 | 0.005 | 0.003 | 0.005 | 0.003 | 0.001 | 0.008 | 0.004 | 0.000 | 0.009 | 0.000 | 0.001 | 0.007 | 0.010 | 0.001 | 0.002 | 0.001 | 0.002 | 0.001 | 0.003 |
| | 192 | 0.029 | 0.012 | 0.058 | 0.023 | 0.002 | 0.001 | 0.000 | 0.001 | 0.099 | 0.031 | 0.131 | 0.030 | 0.005 | 0.002 | 0.003 | 0.005 | 0.004 | 0.002 | 0.001 | 0.004 | 0.001 | 0.000 | 0.018 | 0.010 | 0.001 | 0.002 | 0.007 | 0.006 | 0.000 | 0.005 |
| | 336 | 0.035 | 0.010 | 0.300 | 0.088 | 0.002 | 0.001 | 0.001 | 0.001 | 0.349 | 0.034 | 0.153 | 0.062 | 0.012 | 0.010 | 0.001 | 0.000 | 0.006 | 0.003 | 0.003 | 0.001 | 0.002 | 0.001 | 0.008 | 0.002 | 0.002 | 0.001 | 0.019 | 0.008 | 0.004 | 0.003 |
| | 720 | 0.153 | 0.058 | 0.213 | 0.078 | 0.001 | 0.000 | 0.001 | 0.001 | 0.077 | 0.017 | 0.105 | 0.045 | 0.009 | 0.005 | 0.009 | 0.005 | 0.002 | 0.001 | 0.002 | 0.001 | 0.002 | 0.001 | 0.010 | 0.006 | 0.001 | 0.001 | 0.040 | 0.021 | 0.018 | 0.012 |
| Exp. 400-450 | 96 | 0.007 | 0.002 | 0.017 | 0.015 | 0.001 | 0.001 | 0.000 | 0.000 | 0.105 | 0.024 | 0.003 | 0.006 | 0.002 | 0.001 | 0.000 | 0.002 | 0.001 | 0.001 | 0.001 | 0.006 | 0.000 | 0.000 | 0.002 | 0.004 | 0.001 | 0.002 | 0.000 | 0.001 | 0.000 | 0.001 |
| | 192 | 0.018 | 0.013 | 0.039 | 0.023 | 0.002 | 0.003 | 0.000 | 0.002 | 0.068 | 0.027 | 0.035 | 0.013 | 0.005 | 0.005 | 0.002 | 0.004 | 0.011 | 0.006 | 0.002 | 0.004 | 0.001 | 0.002 | 0.009 | 0.012 | 0.001 | 0.001 | 0.001 | 0.002 | 0.002 | 0.009 |
| | 336 | 0.032 | 0.025 | 0.085 | 0.034 | 0.002 | 0.001 | 0.000 | 0.002 | 0.152 | 0.023 | 0.016 | 0.008 | 0.009 | 0.010 | 0.015 | 0.008 | 0.006 | 0.003 | 0.014 | 0.014 | 0.002 | 0.000 | 0.007 | 0.007 | 0.013 | 0.017 | 0.011 | 0.007 | 0.006 | 0.005 |
| | 720 | 0.051 | 0.038 | 0.166 | 0.059 | 0.000 | 0.001 | 0.003 | 0.003 | 0.154 | 0.035 | 0.056 | 0.021 | 0.001 | 0.001 | 0.029 | 0.011 | 0.017 | 0.009 | 0.000 | 0.004 | 0.002 | 0.001 | 0.013 | 0.008 | 0.009 | 0.005 | 0.058 | 0.022 | 0.040 | 0.019 |
| Exp. 500-550 | 96 | 0.007 | 0.024 | 0.045 | 0.037 | 0.002 | 0.002 | 0.000 | 0.001 | 0.014 | 0.033 | 0.002 | 0.006 | 0.001 | 0.001 | 0.001 | 0.003 | 0.002 | 0.002 | 0.000 | 0.003 | 0.000 | 0.000 | 0.001 | 0.004 | 0.000 | 0.000 | 0.000 | 0.001 | 0.000 | 0.002 |
| | 192 | 0.014 | 0.009 | 0.091 | 0.038 | 0.001 | 0.001 | 0.000 | 0.002 | 0.020 | 0.011 | 0.017 | 0.008 | 0.002 | 0.001 | 0.002 | 0.004 | 0.013 | 0.008 | 0.002 | 0.013 | 0.000 | 0.001 | 0.006 | 0.006 | 0.001 | 0.002 | 0.001 | 0.002 | 0.000 | 0.003 |
| | 336 | 0.058 | 0.024 | 0.063 | 0.021 | 0.004 | 0.002 | 0.000 | 0.004 | 0.015 | 0.013 | 0.041 | 0.013 | 0.005 | 0.003 | 0.001 | 0.004 | 0.018 | 0.007 | 0.001 | 0.009 | 0.003 | 0.006 | 0.023 | 0.010 | 0.002 | 0.006 | 0.004 | 0.004 | 0.003 | 0.005 |
| | 720 | 0.111 | 0.060 | 0.096 | 0.025 | 0.003 | 0.001 | 0.000 | 0.001 | 0.057 | 0.011 | 0.050 | 0.012 | 0.002 | 0.001 | 0.012 | 0.012 | 0.002 | 0.002 | 0.005 | 0.006 | 0.003 | 0.003 | 0.016 | 0.008 | 0.011 | 0.012 | 0.094 | 0.034 | 0.019 | 0.011 |
| Thr. 5-20(e-4) | 96 | 0.003 | 0.008 | 0.037 | 0.021 | 0.001 | 0.002 | 0.001 | 0.004 | 0.005 | 0.003 | 0.019 | 0.020 | 0.002 | 0.001 | 0.013 | 0.016 | 0.000 | 0.005 | 0.000 | 0.005 | 0.001 | 0.004 | 0.017 | 0.010 | 0.001 | 0.009 | 0.000 | 0.002 | 0.000 | 0.005 |
| | 192 | 0.006 | 0.005 | 0.054 | 0.018 | 0.000 | 0.001 | 0.000 | 0.002 | 0.013 | 0.006 | 0.019 | 0.017 | 0.018 | 0.017 | 0.019 | 0.010 | 0.003 | 0.004 | 0.008 | 0.029 | 0.004 | 0.008 | 0.032 | 0.011 | 0.000 | 0.005 | 0.003 | 0.004 | 0.001 | 0.009 |
| | 336 | 0.023 | 0.033 | 0.184 | 0.031 | 0.001 | 0.001 | 0.000 | 0.001 | 0.021 | 0.009 | 0.874 | 0.121 | 0.044 | 0.009 | 0.038 | 0.023 | 0.025 | 0.030 | 0.002 | 0.008 | 0.000 | 0.000 | 0.036 | 0.019 | 0.001 | 0.007 | 0.007 | 0.006 | 0.003 | 0.008 |
| | 720 | 0.013 | 0.012 | 0.056 | 0.030 | 0.001 | 0.001 | 0.000 | 0.000 | 0.009 | 0.006 | 0.007 | 0.001 | 0.006 | 0.003 | 0.064 | 0.010 | 0.008 | 0.007 | 0.004 | 0.005 | 0.002 | 0.004 | 0.047 | 0.025 | 0.004 | 0.011 | 0.082 | 0.024 | 0.003 | 0.002 |
| Thr. 10-40(e-4) | 96 | 0.001 | 0.006 | 0.140 | 0.058 | 0.002 | 0.002 | 0.000 | 0.003 | 0.029 | 0.014 | 0.007 | 0.003 | 0.008 | 0.002 | 0.008 | 0.004 | 0.005 | 0.005 | 0.000 | 0.005 | 0.000 | 0.001 | 0.035 | 0.014 | 0.000 | 0.002 | 0.006 | 0.004 | 0.004 | 0.005 |
| | 192 | 0.011 | 0.008 | 0.069 | 0.010 | 0.001 | 0.000 | 0.001 | 0.003 | 0.061 | 0.017 | 0.002 | 0.001 | 0.047 | 0.002 | 0.005 | 0.012 | 0.003 | 0.003 | 0.005 | 0.001 | 0.000 | 0.000 | 0.030 | 0.015 | 0.001 | 0.001 | 0.009 | 0.006 | 0.004 | 0.002 |
| | 336 | 0.011 | 0.006 | 0.096 | 0.016 | 0.001 | 0.000 | 0.000 | 0.000 | 0.035 | 0.007 | 0.003 | 0.003 | 0.006 | 0.004 | 0.004 | 0.007 | 0.002 | 0.001 | 0.001 | 0.001 | 0.000 | 0.001 | 0.030 | 0.004 | 0.000 | 0.000 | 0.022 | 0.005 | 0.002 | 0.004 |
| | 720 | 0.012 | 0.004 | 0.018 | 0.005 | 0.000 | 0.000 | 0.003 | 0.002 | 0.005 | 0.001 | 0.001 | 0.000 | 0.013 | 0.004 | 0.007 | 0.009 | 0.003 | 0.002 | 0.006 | 0.003 | 0.001 | 0.001 | 0.006 | 0.004 | 0.001 | 0.003 | 0.018 | 0.006 | 0.002 | 0.002 |
| Thr. 15-60(e-4) | 96 | 0.004 | 0.003 | 0.031 | 0.006 | 0.000 | 0.000 | 0.000 | 0.000 | 0.043 | 0.023 | 0.003 | 0.002 | 0.001 | 0.005 | 0.002 | 0.005 | 0.000 | 0.006 | 0.002 | 0.002 | 0.000 | 0.000 | 0.013 | 0.011 | 0.000 | 0.004 | 0.003 | 0.003 | 0.001 | 0.004 |
| | 192 | 0.004 | 0.005 | 0.067 | 0.031 | 0.000 | 0.000 | 0.000 | 0.000 | 0.011 | 0.008 | 0.001 | 0.001 | 0.006 | 0.001 | 0.002 | 0.005 | 0.001 | 0.001 | 0.001 | 0.001 | 0.000 | 0.001 | 0.028 | 0.016 | 0.000 | 0.000 | 0.003 | 0.002 | 0.008 | 0.009 |
| | 336 | 0.005 | 0.003 | 0.046 | 0.014 | 0.000 | 0.000 | 0.001 | 0.001 | 0.011 | 0.005 | 0.001 | 0.000 | 0.002 | 0.002 | 0.013 | 0.004 | 0.001 | 0.001 | 0.002 | 0.003 | 0.001 | 0.000 | 0.018 | 0.010 | 0.000 | 0.000 | 0.003 | 0.002 | 0.009 | 0.009 |
| | 720 | 0.003 | 0.000 | 0.026 | 0.005 | 0.000 | 0.000 | 0.000 | 0.000 | 0.002 | 0.001 | 0.002 | 0.001 | 0.002 | 0.000 | 0.001 | 0.001 | 0.002 | 0.001 | 0.002 | 0.001 | 0.000 | 0.000 | 0.024 | 0.006 | 0.001 | 0.000 | 0.005 | 0.002 | 0.001 | 0.002 |

Table 12: Full results for the scenario datasets–Sine wave. We compare Hippen with extensive competitive baseline models under different horizon lengths using 3 random seeds. *Avg* is averaged from all four horizon lengths: {96, 192, 336, 720}. All values are multiplied by 100.

| Dataset | Pred_len | Hippen MSE | Hippen MAE | NST MSE | NST MAE | DLinear MSE | DLinear MAE | RLinear MSE | RLinear MAE | Dish-TS MSE | Dish-TS MAE | SAN MSE | SAN MAE | Leddam MSE | Leddam MAE | TimeMixer MSE | TimeMixer MAE | iTransformer MSE | iTransformer MAE | PatchTST MSE | PatchTST MAE | TiDE MSE | TiDE MAE | TimesNet MSE | TimesNet MAE | CycleNet MSE | CycleNet MAE | Peri-mid MSE | Peri-mid MAE | FRNet MSE | FRNet MAE |
|---|---|---|---|---|---|---|---|---|---|---|---|---|---|---|---|---|---|---|---|---|---|---|---|---|---|---|---|---|---|---|---|
| Sine 2k-3k | 96 | 0.015 | 0.169 | 0.015 | 0.110 | 0.046 | 0.202 | 0.001 | 0.007 | 2.030 | 0.338 | 0.005 | 0.058 | 0.001 | 0.006 | 0.002 | 0.025 | 0.004 | 0.033 | 0.002 | 0.020 | 0.000 | 0.003 | 0.219 | 0.690 | 0.003 | 0.028 | 0.011 | 0.102 | 0.000 | 0.003 |
|  | 192 | 0.028 | 0.188 | 0.094 | 0.376 | 0.033 | 0.089 | 0.002 | 0.013 | 28.399 | 6.696 | 0.012 | 0.081 | 0.003 | 0.020 | 0.024 | 0.204 | 0.019 | 0.053 | 0.020 | 0.114 | 0.004 | 0.014 | 0.719 | 0.962 | 0.006 | 0.034 | 0.058 | 0.231 | 0.002 | 0.009 |
|  | 336 | 0.017 | 0.138 | 0.265 | 0.565 | 0.073 | 0.071 | 0.005 | 0.010 | 7.865 | 2.022 | 0.059 | 0.021 | 0.080 | 0.235 | 0.136 | 0.229 | 0.049 | 0.080 | 0.110 | 0.257 | 0.002 | 0.006 | 1.137 | 0.866 | 0.003 | 0.005 | 0.979 | 1.564 | 0.004 | 0.017 |
|  | 720 | 0.688 | 0.940 | 0.130 | 0.538 | 0.159 | 0.130 | 0.042 | 0.018 | 3.755 | 2.104 | 0.295 | 0.279 | 0.063 | 0.040 | 0.863 | 0.991 | 0.357 | 0.150 | 0.695 | 0.641 | 0.087 | 0.059 | 4.550 | 2.229 | 1.299 | 0.897 | 16.688 | 8.118 | 2.043 | 2.123 |
| Sine 3k-4k | 96 | 0.005 | 0.063 | 0.003 | 0.012 | 0.016 | 0.097 | 0.000 | 0.002 | 0.839 | 1.026 | 0.004 | 0.051 | 0.000 | 0.005 | 0.001 | 0.009 | 0.000 | 0.003 | 0.002 | 0.030 | 0.000 | 0.003 | 0.058 | 0.204 | 0.001 | 0.010 | 0.002 | 0.031 | 0.000 | 0.006 |
|  | 192 | 0.005 | 0.053 | 0.015 | 0.057 | 0.029 | 0.073 | 0.003 | 0.020 | 7.039 | 1.536 | 0.009 | 0.087 | 0.002 | 0.016 | 0.003 | 0.021 | 0.005 | 0.017 | 0.007 | 0.069 | 0.001 | 0.007 | 0.173 | 0.275 | 0.000 | 0.004 | 0.013 | 0.093 | 0.001 | 0.004 |
|  | 336 | 0.009 | 0.094 | 0.059 | 0.158 | 0.056 | 0.125 | 0.020 | 0.076 | 25.388 | 5.294 | 0.050 | 0.142 | 0.022 | 0.128 | 0.027 | 0.082 | 0.017 | 0.043 | 0.050 | 0.159 | 0.015 | 0.061 | 0.182 | 0.113 | 0.022 | 0.106 | 0.340 | 0.867 | 0.019 | 0.091 |
|  | 720 | 0.400 | 1.574 | 0.465 | 0.804 | 0.067 | 0.075 | 0.165 | 0.174 | 27.908 | 5.458 | 0.184 | 0.074 | 0.108 | 0.168 | 0.788 | 0.305 | 0.195 | 0.165 | 0.239 | 0.148 | 0.010 | 0.011 | 1.157 | 0.625 | 0.035 | 0.064 | 4.570 | 3.898 | 0.148 | 0.206 |
| Sine 4k-5k | 96 | 0.003 | 0.043 | 0.005 | 0.057 | 0.033 | 0.179 | 0.000 | 0.001 | 6.652 | 4.957 | 0.001 | 0.013 | 0.000 | 0.002 | 0.000 | 0.001 | 0.001 | 0.015 | 0.001 | 0.020 | 0.000 | 0.002 | 0.029 | 0.216 | 0.001 | 0.015 | 0.002 | 0.028 | 0.000 | 0.005 |
|  | 192 | 0.003 | 0.050 | 0.012 | 0.063 | 0.036 | 0.124 | 0.000 | 0.003 | 11.629 | 2.746 | 0.002 | 0.031 | 0.000 | 0.005 | 0.000 | 0.020 | 0.006 | 0.037 | 0.002 | 0.016 | 0.003 | 0.037 | 0.116 | 0.495 | 0.000 | 0.006 | 0.024 | 0.214 | 0.002 | 0.023 |
|  | 336 | 0.004 | 0.039 | 0.040 | 0.204 | 0.045 | 0.109 | 0.004 | 0.020 | 15.333 | 3.691 | 0.008 | 0.050 | 0.006 | 0.020 | 0.020 | 0.121 | 0.010 | 0.038 | 0.007 | 0.045 | 0.006 | 0.038 | 0.118 | 0.243 | 0.002 | 0.013 | 0.104 | 0.438 | 0.003 | 0.014 |
|  | 720 | 0.039 | 0.201 | 0.171 | 0.212 | 0.112 | 0.136 | 0.023 | 0.039 | 110.890 | 5.533 | 0.066 | 0.077 | 0.198 | 0.278 | 0.108 | 0.242 | 0.056 | 0.045 | 0.033 | 0.003 | 0.016 | 0.018 | 0.724 | 0.776 | 0.036 | 0.088 | 2.758 | 2.624 | 0.390 | 0.427 |
| Sine 5k-6k | 96 | 0.002 | 0.020 | 0.001 | 0.014 | 0.019 | 0.091 | 0.000 | 0.002 | 1.066 | 2.378 | 0.000 | 0.007 | 0.000 | 0.001 | 0.005 | 0.012 | 0.001 | 0.007 | 0.001 | 0.018 | 0.000 | 0.001 | 0.023 | 0.214 | 0.000 | 0.002 | 0.003 | 0.035 | 0.000 | 0.003 |
|  | 192 | 0.004 | 0.058 | 0.012 | 0.157 | 0.038 | 0.130 | 0.000 | 0.003 | 0.201 | 0.060 | 0.001 | 0.009 | 0.000 | 0.005 | 0.005 | 0.055 | 0.003 | 0.027 | 0.005 | 0.055 | 0.000 | 0.001 | 0.083 | 0.439 | 0.002 | 0.019 | 0.049 | 0.419 | 0.001 | 0.015 |
|  | 336 | 0.007 | 0.068 | 0.012 | 0.124 | 0.035 | 0.108 | 0.002 | 0.017 | 3.001 | 3.340 | 0.001 | 0.003 | 0.011 | 0.060 | 0.011 | 0.058 | 0.006 | 0.026 | 0.011 | 0.069 | 0.000 | 0.001 | 0.047 | 0.089 | 0.005 | 0.050 | 0.276 | 1.071 | 0.010 | 0.056 |
|  | 720 | 0.056 | 0.315 | 0.056 | 0.210 | 0.064 | 0.210 | 0.011 | 0.023 | 5.455 | 3.586 | 0.041 | 0.078 | 0.162 | 0.370 | 0.029 | 0.300 | 0.046 | 0.035 | 0.105 | 0.120 | 0.007 | 0.014 | 0.440 | 0.631 | 0.061 | 0.164 | 3.850 | 3.735 | 0.174 | 0.292 |
| Sine 6k-7k | 96 | 0.002 | 0.028 | 0.006 | 0.086 | 0.002 | 0.019 | 0.000 | 0.001 | 0.348 | 0.698 | 0.000 | 0.003 | 0.000 | 0.001 | 0.000 | 0.005 | 0.001 | 0.010 | 0.000 | 0.006 | 0.000 | 0.001 | 0.008 | 0.101 | 0.001 | 0.001 | 0.001 | 0.010 | 0.000 | 0.001 |
|  | 192 | 0.004 | 0.051 | 0.004 | 0.048 | 0.037 | 0.126 | 0.000 | 0.001 | 0.435 | 1.043 | 0.001 | 0.017 | 0.000 | 0.005 | 0.000 | 0.005 | 0.002 | 0.015 | 0.003 | 0.039 | 0.000 | 0.000 | 0.047 | 0.306 | 0.000 | 0.006 | 0.011 | 0.119 | 0.000 | 0.003 |
|  | 336 | 0.001 | 0.009 | 0.027 | 0.183 | 0.020 | 0.062 | 0.000 | 0.001 | 3.380 | 3.823 | 0.003 | 0.028 | 0.004 | 0.043 | 0.006 | 0.038 | 0.003 | 0.018 | 0.008 | 0.062 | 0.000 | 0.001 | 0.031 | 0.153 | 0.004 | 0.031 | 0.091 | 0.587 | 0.002 | 0.014 |
|  | 720 | 0.031 | 0.168 | 0.148 | 0.324 | 0.025 | 0.060 | 0.003 | 0.009 | 23.987 | 9.960 | 0.053 | 0.110 | 0.174 | 0.383 | 0.081 | 0.250 | 0.061 | 0.107 | 0.053 | 0.128 | 0.001 | 0.004 | 1.069 | 1.697 | 0.015 | 0.011 | 2.397 | 3.549 | 0.019 | 0.082 |

Table 13: Results on the S&P 500 dataset. For brevity, only the top 30 stocks in alphabetical order are reported out of the full 500.

| Dataset | Hipeen_4.0 | | Hipeen_2.0 | | Hipeen_1.0 | | DLinear | | Nonstat. Transf. | | TimeMixer | | iTransformer | | PatchTST | | TiDE | | TimesNet | | SAN_DLinear | | CycleNet | | PerimidFormer | | FRNet | | RLinear | | Leddam | | DishTS | |
|---|---|---|---|---|---|---|---|---|---|---|---|---|---|---|---|---|---|---|---|---|---|---|---|---|---|---|---|---|---|---|---|---|---|---|
| | MSE | MAE | MSE | MAE | MSE | MAE | MSE | MAE | MSE | MAE | MSE | MAE | MSE | MAE | MSE | MAE | MSE | MAE | MSE | MAE | MSE | MAE | MSE | MAE | MSE | MAE | MSE | MAE | MSE | MAE | MSE | MAE | MSE | MAE |
| A | 0.575 | 0.586 | 0.619 | 0.615 | 0.626 | 0.631 | 0.705 | 0.673 | 0.810 | 0.677 | 0.746 | 0.666 | 0.624 | 0.629 | 0.602 | 0.599 | 0.566 | 0.592 | 0.709 | 0.652 | 0.602 | 0.612 | 0.651 | 0.628 | 0.705 | 0.650 | 0.640 | 0.612 | 0.775 | 0.680 | 0.601 | 0.611 | 0.999 | 0.791 |
| AAPL | 0.978 | 0.719 | 0.891 | 0.881 | 0.971 | 0.728 | 1.115 | 0.792 | 2.089 | 1.032 | 1.104 | 0.755 | 1.074 | 0.744 | 1.158 | 0.790 | 1.170 | 0.772 | 1.223 | 0.816 | 1.039 | 0.773 | 1.013 | 0.744 | 1.001 | 0.743 | 1.044 | 0.746 | 0.957 | 0.730 | 0.966 | 0.709 | 1.943 | 1.113 |
| ABT | 0.236 | 0.386 | 0.238 | 0.391 | 0.228 | 0.385 | 0.248 | 0.394 | 0.496 | 0.552 | 0.349 | 0.464 | 0.218 | 0.376 | 0.288 | 0.419 | 0.211 | 0.366 | 0.319 | 0.449 | 0.267 | 0.417 | 0.295 | 0.424 | 0.271 | 0.402 | 0.293 | 0.417 | 0.334 | 0.444 | 0.227 | 0.382 | 0.366 | 0.476 |
| ACGL | 1.949 | 0.583 | 1.940 | 0.574 | 2.002 | 0.613 | 2.865 | 0.973 | 5.698 | 1.302 | 2.170 | 0.655 | 2.333 | 0.752 | 2.035 | 0.607 | 2.327 | 0.744 | 1.925 | 0.565 | 2.443 | 0.821 | 2.068 | 0.603 | 2.047 | 0.622 | 2.012 | 0.591 | 1.941 | 0.564 | 2.123 | 0.651 | 2.092 | 0.669 |
| ACN | 1.048 | 0.573 | 0.488 | 0.541 | 0.491 | 0.553 | 0.497 | 0.563 | 0.930 | 0.712 | 0.933 | 0.713 | 0.478 | 0.549 | 0.632 | 0.597 | 0.453 | 0.527 | 1.045 | 0.790 | 0.509 | 0.554 | 0.886 | 0.704 | 0.801 | 0.668 | 0.667 | 0.610 | 0.890 | 0.709 | 0.479 | 0.536 | 0.717 | 0.667 |
| ADBE | 0.598 | 0.569 | 0.652 | 0.602 | 0.670 | 0.631 | 0.965 | 0.766 | 1.922 | 1.057 | 1.338 | 0.852 | 0.745 | 0.679 | 1.338 | 0.876 | 0.636 | 0.627 | 1.507 | 0.943 | 0.815 | 0.689 | 1.178 | 0.811 | 1.436 | 0.894 | 0.811 | 0.674 | 1.653 | 0.965 | 0.606 | 0.605 | 1.179 | 0.842 |
| ADI | 0.420 | 0.489 | 0.421 | 0.495 | 0.436 | 0.511 | 0.575 | 0.600 | 0.605 | 0.594 | 0.487 | 0.536 | 0.501 | 0.555 | 0.426 | 0.503 | 0.480 | 0.535 | 0.428 | 0.506 | 0.437 | 0.512 | 0.433 | 0.505 | 0.428 | 0.502 | 0.406 | 0.488 | 0.410 | 0.489 | 0.460 | 0.526 | 0.446 | 0.513 |
| ADM | 1.818 | 0.972 | 1.719 | 0.947 | 1.764 | 0.969 | 2.282 | 1.079 | 3.297 | 1.375 | 2.610 | 1.153 | 1.783 | 0.964 | 1.771 | 0.971 | 1.797 | 0.974 | 1.882 | 1.015 | 1.749 | 0.950 | 1.831 | 0.994 | 1.743 | 0.950 | 1.743 | 0.958 | 1.753 | 0.953 | 1.715 | 0.942 | 2.285 | 1.097 |
| ADP | 0.247 | 0.373 | 0.232 | 0.360 | 0.229 | 0.358 | 0.275 | 0.398 | 0.400 | 0.478 | 0.391 | 0.482 | 0.269 | 0.388 | 0.294 | 0.414 | 0.255 | 0.373 | 0.375 | 0.479 | 0.236 | 0.365 | 0.355 | 0.455 | 0.331 | 0.440 | 0.312 | 0.422 | 0.330 | 0.441 | 0.258 | 0.374 | 0.302 | 0.425 |
| ADSK | 0.317 | 0.393 | 0.340 | 0.410 | 0.340 | 0.413 | 0.424 | 0.439 | 0.661 | 0.568 | 0.607 | 0.529 | 0.367 | 0.416 | 0.667 | 0.511 | 0.316 | 0.391 | 0.622 | 0.526 | 0.414 | 0.446 | 0.584 | 0.504 | 0.479 | 0.457 | 0.550 | 0.478 | 0.626 | 0.517 | 0.386 | 0.422 | 0.584 | 0.508 |
| AEE | 0.265 | 0.373 | 0.262 | 0.373 | 0.264 | 0.376 | 0.271 | 0.378 | 0.388 | 0.484 | 0.368 | 0.458 | 0.267 | 0.366 | 0.329 | 0.427 | 0.255 | 0.356 | 0.358 | 0.469 | 0.288 | 0.402 | 0.360 | 0.452 | 0.324 | 0.428 | 0.305 | 0.411 | 0.341 | 0.447 | 0.261 | 0.365 | 0.307 | 0.426 |
| AEP | 0.280 | 0.379 | 0.274 | 0.379 | 0.276 | 0.384 | 0.315 | 0.426 | 0.395 | 0.480 | 0.323 | 0.412 | 0.312 | 0.411 | 0.318 | 0.420 | 0.292 | 0.397 | 0.311 | 0.416 | 0.286 | 0.393 | 0.307 | 0.403 | 0.313 | 0.419 | 0.284 | 0.392 | 0.326 | 0.430 | 0.287 | 0.390 | 0.342 | 0.457 |
| AES | 1.946 | 1.085 | 2.013 | 1.112 | 2.093 | 1.139 | 2.046 | 1.076 | 3.303 | 1.405 | 3.829 | 1.465 | 2.098 | 1.139 | 2.728 | 1.256 | 1.989 | 1.100 | 2.195 | 1.161 | 2.070 | 1.109 | 2.642 | 1.262 | 2.098 | 1.136 | 2.168 | 1.155 | 2.172 | 1.155 | 2.062 | 1.129 | 3.115 | 1.384 |
| AFL | 0.506 | 0.513 | 0.531 | 0.527 | 0.583 | 0.551 | 1.142 | 0.774 | 0.688 | 0.595 | 0.932 | 0.658 | 0.683 | 0.598 | 0.700 | 0.594 | 0.655 | 0.584 | 0.586 | 0.548 | 0.678 | 0.603 | 0.540 | 0.522 | 0.624 | 0.563 | 0.575 | 0.538 | 0.552 | 0.529 | 0.623 | 0.565 | 0.689 | 0.614 |
| AIG | 0.213 | 0.361 | 0.217 | 0.367 | 0.219 | 0.370 | 0.342 | 0.466 | 0.479 | 0.513 | 0.329 | 0.413 | 0.253 | 0.399 | 0.256 | 0.378 | 0.253 | 0.392 | 0.264 | 0.392 | 0.252 | 0.397 | 0.257 | 0.388 | 0.241 | 0.385 | 0.255 | 0.383 | 0.241 | 0.386 | 0.237 | 0.379 | 0.359 | 0.481 |
| AIZ | 0.360 | 0.455 | 0.400 | 0.488 | 0.400 | 0.494 | 0.516 | 0.573 | 0.880 | 0.683 | 0.657 | 0.591 | 0.454 | 0.527 | 0.478 | 0.521 | 0.401 | 0.492 | 0.485 | 0.527 | 0.376 | 0.480 | 0.476 | 0.517 | 0.483 | 0.525 | 0.469 | 0.520 | 0.522 | 0.542 | 0.433 | 0.512 | 0.620 | 0.609 |
| AJG | 0.516 | 0.551 | 0.754 | 0.613 | 0.602 | 0.601 | 0.569 | 0.583 | 0.445 | 0.520 | 0.402 | 0.490 | 0.382 | 0.479 | 0.371 | 0.465 | 0.382 | 0.472 | 0.392 | 0.479 | 0.409 | 0.496 | 0.367 | 0.466 | 0.362 | 0.463 | 0.375 | 0.477 | 0.360 | 0.493 | 0.605 | 0.598 | 0.453 | 0.528 |
| AKAM | 0.385 | 0.481 | 0.380 | 0.482 | 0.383 | 0.486 | 0.403 | 0.506 | 0.529 | 0.552 | 0.519 | 0.530 | 0.395 | 0.497 | 0.407 | 0.492 | 0.382 | 0.487 | 0.390 | 0.482 | 0.353 | 0.473 | 0.405 | 0.491 | 0.386 | 0.488 | 0.388 | 0.476 | 0.397 | 0.493 | 0.379 | 0.481 | 0.616 | 0.640 |
| ALB | 2.681 | 1.219 | 2.655 | 1.212 | 2.608 | 1.202 | 2.812 | 1.211 | 4.708 | 1.588 | 3.272 | 1.349 | 2.808 | 1.276 | 3.220 | 1.313 | 2.782 | 1.258 | 3.056 | 1.312 | 2.465 | 1.167 | 2.420 | 1.182 | 2.644 | 1.228 | 2.657 | 1.222 | 2.630 | 1.223 | 2.634 | 1.217 | 3.357 | 1.402 |
| ALGN | 1.224 | 0.722 | 1.235 | 0.725 | 1.132 | 0.707 | 1.027 | 0.738 | 1.147 | 0.842 | 1.467 | 0.793 | 1.073 | 0.729 | 1.260 | 0.749 | 0.964 | 0.681 | 1.136 | 0.719 | 0.995 | 0.679 | 1.184 | 0.725 | 1.053 | 0.702 | 1.017 | 0.683 | 1.092 | 0.712 | 1.059 | 0.696 | 1.142 | 0.749 |
| ALL | 0.309 | 0.415 | 0.315 | 0.420 | 0.348 | 0.446 | 0.485 | 0.526 | 0.996 | 0.732 | 0.395 | 0.477 | 0.417 | 0.487 | 0.331 | 0.428 | 0.379 | 0.461 | 0.328 | 0.431 | 0.370 | 0.459 | 0.335 | 0.439 | 0.348 | 0.445 | 0.337 | 0.435 | 0.336 | 0.438 | 0.347 | 0.440 | 0.440 | 0.509 |
| AMAT | 2.341 | 1.068 | 2.068 | 1.065 | 2.041 | 1.064 | 2.874 | 1.295 | 5.393 | 1.687 | 2.677 | 1.157 | 2.343 | 1.161 | 2.170 | 1.059 | 2.133 | 1.081 | 2.244 | 1.082 | 2.382 | 1.165 | 2.241 | 1.084 | 2.160 | 1.091 | 2.182 | 1.072 | 2.158 | 1.090 | 2.146 | 1.085 | 2.845 | 1.278 |
| AMD | 3.629 | 1.423 | 3.866 | 1.461 | 4.230 | 1.471 | 6.464 | 2.001 | 7.103 | 1.914 | 5.861 | 1.741 | 4.199 | 1.548 | 5.180 | 1.708 | 3.978 | 1.499 | 5.521 | 1.789 | 5.228 | 1.758 | 4.851 | 1.622 | 5.255 | 1.764 | 4.528 | 1.623 | 5.316 | 1.778 | 4.045 | 1.516 | 6.314 | 2.009 |
| AME | 0.316 | 0.445 | 0.312 | 0.446 | 0.316 | 0.454 | 0.447 | 0.534 | 0.309 | 0.438 | 0.340 | 0.463 | 0.382 | 0.498 | 0.314 | 0.440 | 0.361 | 0.472 | 0.370 | 0.494 | 0.328 | 0.455 | 0.337 | 0.461 | 0.362 | 0.483 | 0.340 | 0.465 | 0.350 | 0.476 | 0.346 | 0.469 | 0.453 | 0.503 |
| AMGN | 0.205 | 0.352 | 0.216 | 0.368 | 0.225 | 0.380 | 0.274 | 0.429 | 0.309 | 0.420 | 0.240 | 0.376 | 0.244 | 0.399 | 0.205 | 0.350 | 0.231 | 0.384 | 0.219 | 0.360 | 0.203 | 0.362 | 0.196 | 0.339 | 0.219 | 0.363 | 0.215 | 0.367 | 0.202 | 0.344 | 0.216 | 0.369 | 0.231 | 0.392 |
| AMP | 0.732 | 0.551 | 0.704 | 0.626 | 0.602 | 0.602 | 1.904 | 1.044 | 3.650 | 1.487 | 1.003 | 0.722 | 0.870 | 0.700 | 0.774 | 0.656 | 0.872 | 0.698 | 0.754 | 0.649 | 1.254 | 0.863 | 0.687 | 0.605 | 0.783 | 0.657 | 0.754 | 0.648 | 0.731 | 0.634 | 0.800 | 0.664 | 0.914 | 0.732 |
| AMT | 0.262 | 0.393 | 0.265 | 0.399 | 0.269 | 0.405 | 0.280 | 0.420 | 0.410 | 0.503 | 0.474 | 0.525 | 0.246 | 0.388 | 0.403 | 0.497 | 0.230 | 0.368 | 0.650 | 0.654 | 0.308 | 0.434 | 0.471 | 0.529 | 0.385 | 0.479 | 0.282 | 0.416 | 0.555 | 0.592 | 0.243 | 0.424 | 0.369 | 0.496 |
| AMZN | 0.404 | 0.455 | 0.454 | 0.504 | 0.462 | 0.525 | 0.541 | 0.594 | 1.042 | 0.730 | 0.691 | 0.561 | 0.471 | 0.549 | 0.559 | 0.530 | 0.445 | 0.523 | 0.596 | 0.574 | 0.481 | 0.529 | 0.645 | 0.550 | 0.607 | 0.579 | 0.442 | 0.497 | 0.643 | 0.589 | 0.424 | 0.504 | 0.671 | 0.674 |
| AON | 0.541 | 0.529 | 0.563 | 0.548 | 0.568 | 0.553 | 0.861 | 0.692 | 0.793 | 0.633 | 0.885 | 0.662 | 0.623 | 0.580 | 0.811 | 0.627 | 0.534 | 0.521 | 1.143 | 0.772 | 0.755 | 0.655 | 0.819 | 0.646 | 0.919 | 0.678 | 0.610 | 0.562 | 1.038 | 0.721 | 0.576 | 0.551 | 0.934 | 0.701 |
| AOS | 0.244 | 0.370 | 0.239 | 0.376 | 0.237 | 0.377 | 0.246 | 0.391 | 0.449 | 0.526 | 0.370 | 0.468 | 0.262 | 0.382 | 0.285 | 0.404 | 0.258 | 0.384 | 0.354 | 0.471 | 0.269 | 0.406 | 0.323 | 0.437 | 0.299 | 0.418 | 0.264 | 0.388 | 0.340 | 0.453 | 0.258 | 0.383 | 0.293 | 0.435 |
| APA | 0.186 | 0.323 | 0.180 | 0.328 | 0.182 | 0.337 | 0.246 | 0.406 | 0.210 | 0.309 | 0.291 | 0.386 | 0.223 | 0.382 | 0.179 | 0.313 | 0.197 | 0.351 | 0.202 | 0.331 | 0.187 | 0.346 | 0.201 | 0.330 | 0.202 | 0.352 | 0.188 | 0.329 | 0.209 | 0.351 | 0.191 | 0.340 | 0.259 | 0.401 |
| APD | 0.117 | 0.214 | 0.114 | 0.214 | 0.115 | 0.222 | 0.124 | 0.242 | 0.210 | 0.309 | 0.162 | 0.242 | 0.140 | 0.235 | 0.132 | 0.225 | 0.130 | 0.229 | 0.149 | 0.246 | 0.124 | 0.220 | 0.152 | 0.239 | 0.128 | 0.226 | 0.122 | 0.223 | 0.140 | 0.233 | 0.126 | 0.225 | 0.174 | 0.296 |
| APH | 0.450 | 0.492 | 0.458 | 0.503 | 0.436 | 0.487 | 0.483 | 0.522 | 0.613 | 0.603 | 0.613 | 0.594 | 0.470 | 0.505 | 0.539 | 0.548 | 0.437 | 0.480 | 0.661 | 0.644 | 0.451 | 0.503 | 0.534 | 0.556 | 0.567 | 0.575 | 0.459 | 0.499 | 0.653 | 0.638 | 0.453 | 0.494 | 0.583 | 0.597 |
| ARE | 0.753 | 0.600 | 0.788 | 0.619 | 0.842 | 0.647 | 1.089 | 0.751 | 2.245 | 1.009 | 0.643 | 0.579 | 0.886 | 0.677 | 0.688 | 0.594 | 0.833 | 0.651 | 0.722 | 0.615 | 0.838 | 0.643 | 0.662 | 0.585 | 0.704 | 0.599 | 0.644 | 0.576 | 0.657 | 0.583 | 0.779 | 0.631 | 0.843 | 0.663 |
| ATO | 0.673 | 0.580 | 0.657 | 0.598 | 0.669 | 0.610 | 0.679 | 0.616 | 0.714 | 0.643 | 0.802 | 0.666 | 0.604 | 0.570 | 0.771 | 0.645 | 0.537 | 0.523 | 0.973 | 0.743 | 0.657 | 0.593 | 0.869 | 0.692 | 0.748 | 0.631 | 0.800 | 0.667 | 0.935 | 0.724 | 0.616 | 0.577 | 0.941 | 0.730 |
| AVB | 0.246 | 0.245 | 0.177 | 0.238 | 0.174 | 0.236 | 0.190 | 0.260 | 0.286 | 0.344 | 0.187 | 0.267 | 0.207 | 0.274 | 0.187 | 0.263 | 0.188 | 0.256 | 0.191 | 0.275 | 0.172 | 0.241 | 0.179 | 0.246 | 0.186 | 0.263 | 0.184 | 0.252 | 0.184 | 0.261 | 0.180 | 0.248 | 0.189 | 0.243 |
| AVGO | 0.404 | 0.455 | 0.403 | 0.460 | 0.403 | 0.469 | 0.405 | 0.503 | 0.466 | 0.517 | 0.479 | 0.510 | 0.436 | 0.508 | 0.448 | 0.501 | 0.397 | 0.481 | 0.461 | 0.509 | 0.357 | 0.458 | 0.436 | 0.481 | 0.472 | 0.510 | 0.406 | 0.466 | 0.486 | 0.517 | 0.404 | 0.478 | 0.608 | 0.595 |
| AVY | 2.727 | 1.141 | 2.677 | 1.143 | 3.084 | 1.243 | 4.124 | 1.467 | 3.402 | 1.327 | 1.936 | 0.993 | 4.181 | 1.487 | 2.175 | 1.050 | 3.104 | 1.256 | 2.464 | 1.136 | 3.294 | 1.301 | 1.936 | 0.991 | 2.951 | 1.223 | 2.053 | 1.003 | 2.228 | 1.073 | 2.887 | 1.202 | 3.434 | 1.342 |
| AWK | 0.238 | 0.372 | 0.249 | 0.384 | 0.258 | 0.392 | 0.317 | 0.437 | 0.407 | 0.496 | 0.356 | 0.440 | 0.285 | 0.413 | 0.285 | 0.400 | 0.256 | 0.390 | 0.293 | 0.415 | 0.251 | 0.390 | 0.273 | 0.401 | 0.285 | 0.405 | 0.260 | 0.390 | 0.289 | 0.407 | 0.255 | 0.388 | 0.381 | 0.472 |
| | 0.224 | 0.349 | 0.240 | 0.370 | 0.238 | 0.369 | 0.259 | 0.388 | 0.435 | 0.527 | 0.379 | 0.472 | 0.222 | 0.346 | 0.290 | 0.413 | 0.199 | 0.318 | 0.414 | 0.518 | 0.260 | 0.384 | 0.247 | 0.375 | 0.249 | 0.367 | 0.221 | 0.345 | 0.356 | 0.472 | 0.203 | 0.325 | 0.357 | 0.479 |

Table 14: **MAE (Mean Absolute Error)** results on the Nifty50 dataset, averaged over three random seeds. For brevity, stocks are represented by the first three letters of their alphabetical names.

| Stock | Horizon | Hipeen | DLinear | RLinear | SAN | Leddam | DDN | FAN | TimeMixer | PatchTST | TiDE | TimesNet | CycleNet | Peri-mid. | FRNet | Smamba | STF |
|---|---|---|---|---|---|---|---|---|---|---|---|---|---|---|---|---|---|
| ADA. | 12 | **0.083** | 0.190 | 0.095 | 0.087 | 0.103 | 0.117 | 0.120 | 0.094 | 0.099 | 0.101 | 0.176 | 0.104 | 0.088 | 0.096 | 0.421 | 0.190 |
| ADA. | 24 | **0.122** | 0.213 | 0.141 | 0.122 | 0.169 | 0.169 | 0.152 | 0.134 | 0.154 | 0.141 | 0.216 | 0.160 | 0.127 | 0.141 | 0.403 | 0.216 |
| ADA. | 48 | **0.180** | 0.232 | 0.197 | 0.180 | 0.208 | 0.215 | 0.195 | 0.206 | 0.214 | 0.196 | 0.200 | 0.217 | 0.194 | 0.202 | 0.385 | 0.266 |
| ADA. | 96 | 0.212 | 0.235 | 0.228 | 0.228 | 0.226 | 0.262 | **0.206** | 0.258 | 0.258 | 0.232 | 0.248 | 0.257 | 0.230 | 0.230 | 0.355 | 0.329 |
| ADA. | Avg. | **0.149** | 0.218 | 0.165 | 0.154 | 0.177 | 0.191 | 0.168 | 0.173 | 0.181 | 0.168 | 0.210 | 0.185 | 0.160 | 0.167 | 0.391 | 0.250 |
| BAJ. | 12 | 0.160 | 0.313 | 0.165 | **0.158** | 0.183 | 0.260 | 0.204 | 0.171 | 0.165 | 0.188 | 0.264 | 0.174 | 0.166 | 0.167 | 0.823 | 0.605 |
| BAJ. | 24 | **0.220** | 0.366 | 0.232 | 0.225 | 0.244 | 0.296 | 0.257 | 0.236 | 0.254 | 0.267 | 0.292 | 0.240 | 0.236 | 0.233 | 0.847 | 0.622 |
| BAJ. | 48 | **0.308** | 0.470 | 0.327 | 0.327 | 0.321 | 0.392 | 0.348 | 0.330 | 0.327 | 0.378 | 0.369 | 0.337 | 0.333 | 0.329 | 0.897 | 0.718 |
| BAJ. | 96 | **0.401** | 0.597 | 0.428 | 0.423 | 0.436 | 0.442 | 0.434 | 0.452 | 0.421 | 0.464 | 0.486 | 0.453 | 0.437 | 0.432 | 0.948 | 0.823 |
| BAJ. | Avg. | **0.272** | 0.437 | 0.288 | 0.283 | 0.296 | 0.347 | 0.311 | 0.297 | 0.292 | 0.324 | 0.353 | 0.301 | 0.293 | 0.290 | 0.878 | 0.692 |
| HDF. | 12 | **0.084** | 0.166 | 0.087 | 0.092 | 0.090 | 0.125 | 0.107 | 0.095 | 0.092 | 0.095 | 0.125 | 0.097 | 0.086 | 0.089 | 0.548 | 0.467 |
| HDF. | 24 | **0.113** | 0.196 | 0.116 | 0.124 | 0.120 | 0.154 | 0.161 | 0.129 | 0.121 | 0.123 | 0.154 | 0.126 | 0.115 | 0.121 | 0.527 | 0.439 |
| HDF. | 48 | **0.158** | 0.256 | 0.167 | 0.188 | 0.170 | 0.192 | 0.207 | 0.181 | 0.171 | 0.173 | 0.187 | 0.180 | 0.166 | 0.170 | 0.527 | 0.450 |
| HDF. | 96 | **0.211** | 0.336 | 0.239 | 0.269 | 0.245 | 0.257 | 0.307 | 0.259 | 0.254 | 0.244 | 0.265 | 0.257 | 0.239 | 0.242 | 0.536 | 0.346 |
| HDF. | Avg. | **0.141** | 0.238 | 0.152 | 0.168 | 0.156 | 0.182 | 0.196 | 0.166 | 0.160 | 0.159 | 0.183 | 0.165 | 0.151 | 0.155 | 0.535 | 0.426 |
| HER. | 12 | 0.150 | 0.253 | 0.149 | 0.147 | 0.155 | 0.200 | 0.191 | 0.149 | 0.149 | 0.163 | 0.192 | 0.152 | **0.145** | 0.150 | 1.557 | 1.208 |
| HER. | 24 | 0.201 | 0.287 | 0.196 | 0.196 | 0.199 | 0.234 | 0.244 | 0.198 | 0.202 | 0.209 | 0.228 | 0.203 | **0.195** | 0.197 | 1.581 | 1.369 |
| HER. | 48 | 0.263 | 0.339 | 0.258 | 0.264 | 0.262 | 0.281 | 0.305 | 0.260 | 0.269 | 0.277 | 0.288 | 0.258 | **0.256** | 0.258 | 1.625 | 1.512 |
| HER. | 96 | 0.353 | 0.437 | 0.352 | 0.363 | 0.349 | 0.374 | 0.410 | **0.346** | 0.355 | 0.358 | 0.379 | 0.350 | 0.353 | 0.352 | 1.634 | 1.613 |
| HER. | Avg. | 0.242 | 0.329 | 0.239 | 0.243 | 0.241 | 0.272 | 0.288 | 0.238 | 0.244 | 0.252 | 0.272 | 0.241 | **0.237** | 0.239 | 1.599 | 1.426 |
| HIN. | 12 | **0.025** | 0.052 | 0.026 | 0.027 | 0.026 | 0.041 | 0.038 | 0.026 | 0.025 | 0.029 | 0.039 | 0.028 | 0.026 | 0.026 | 0.075 | 0.132 |
| HIN. | 24 | 0.036 | 0.065 | **0.034** | 0.035 | 0.036 | 0.043 | 0.044 | 0.035 | 0.037 | 0.037 | 0.042 | 0.036 | 0.034 | 0.035 | 0.083 | 0.142 |
| HIN. | 48 | 0.050 | 0.070 | 0.048 | 0.049 | 0.048 | 0.060 | 0.056 | 0.052 | 0.049 | 0.050 | 0.052 | 0.050 | **0.046** | 0.048 | 0.087 | 0.142 |
| HIN. | 96 | 0.072 | 0.084 | 0.063 | 0.065 | 0.063 | 0.070 | 0.069 | 0.065 | 0.066 | 0.076 | 0.071 | 0.067 | **0.062** | 0.063 | 0.095 | 0.144 |
| HIN. | Avg. | 0.046 | 0.068 | 0.043 | 0.044 | 0.043 | 0.053 | 0.052 | 0.045 | 0.044 | 0.048 | 0.051 | 0.045 | **0.042** | 0.043 | 0.085 | 0.140 |
| LT | 12 | **0.067** | 0.128 | 0.070 | 0.069 | 0.076 | 0.106 | 0.087 | 0.074 | 0.069 | 0.073 | 0.108 | 0.077 | 0.069 | 0.069 | 0.219 | 0.135 |
| LT | 24 | 0.104 | 0.155 | **0.100** | 0.102 | 0.108 | 0.131 | 0.114 | 0.113 | 0.106 | 0.104 | 0.128 | 0.113 | 0.101 | 0.100 | 0.233 | 0.169 |
| LT | 48 | **0.141** | 0.195 | 0.143 | 0.147 | 0.156 | 0.167 | 0.159 | 0.152 | 0.161 | 0.146 | 0.165 | 0.168 | 0.143 | 0.142 | 0.248 | 0.183 |
| LT | 96 | 0.207 | 0.262 | **0.185** | 0.197 | 0.195 | 0.202 | 0.244 | 0.194 | 0.206 | 0.244 | 0.207 | 0.218 | 0.191 | 0.185 | 0.286 | 0.243 |
| LT | Avg. | 0.130 | 0.185 | 0.125 | 0.129 | 0.134 | 0.152 | 0.151 | 0.133 | 0.136 | 0.142 | 0.152 | 0.144 | 0.126 | **0.124** | 0.247 | 0.183 |
| MAR. | 12 | **0.278** | 0.624 | 0.292 | 0.306 | 0.306 | 0.531 | 0.408 | 0.297 | 0.285 | 0.311 | 0.409 | 0.294 | 0.285 | 0.287 | 4.323 | 3.828 |
| MAR. | 24 | **0.377** | 0.719 | 0.394 | 0.405 | 0.401 | 0.521 | 0.539 | 0.425 | 0.412 | 0.407 | 0.514 | 0.401 | 0.387 | 0.394 | 4.401 | 4.095 |
| MAR. | 48 | **0.543** | 0.864 | 0.565 | 0.599 | 0.572 | 0.668 | 0.749 | 0.619 | 0.567 | 0.581 | 0.609 | 0.563 | 0.568 | 0.575 | 4.506 | 4.380 |
| MAR. | 96 | **0.751** | 1.141 | 0.786 | 0.933 | 0.777 | 0.838 | 1.245 | 0.798 | 0.789 | 0.789 | 0.845 | 0.757 | 0.840 | 0.785 | 4.628 | 4.684 |
| MAR. | Avg. | **0.487** | 0.837 | 0.509 | 0.561 | 0.514 | 0.639 | 0.735 | 0.535 | 0.513 | 0.522 | 0.595 | 0.504 | 0.520 | 0.510 | 4.465 | 4.247 |
| NTP. | 12 | 0.138 | 0.188 | 0.116 | 0.117 | 0.117 | 0.167 | 0.155 | 0.120 | 0.115 | 0.121 | 0.156 | **0.110** | 0.114 | 0.118 | 0.449 | 0.241 |
| NTP. | 24 | 0.160 | 0.226 | 0.152 | 0.171 | 0.150 | 0.171 | 0.215 | 0.154 | 0.152 | 0.158 | 0.181 | **0.149** | 0.154 | 0.152 | 0.478 | 0.270 |
| NTP. | 48 | 0.217 | 0.279 | 0.195 | 0.238 | 0.190 | 0.211 | 0.276 | 0.201 | 0.202 | 0.201 | 0.212 | 0.194 | 0.196 | 0.198 | 0.516 | 0.386 |
| NTP. | 96 | 0.302 | 0.385 | 0.256 | 0.318 | 0.255 | 0.284 | 0.407 | 0.294 | 0.295 | 0.263 | 0.280 | 0.256 | 0.266 | 0.266 | 0.592 | 0.641 |
| NTP. | Avg. | 0.204 | 0.269 | 0.180 | 0.211 | 0.178 | 0.208 | 0.263 | 0.192 | 0.191 | 0.186 | 0.207 | **0.177** | 0.182 | 0.183 | 0.509 | 0.385 |
| POW. | 12 | **0.203** | 0.332 | 0.216 | 0.208 | 0.225 | 0.305 | 0.291 | 0.216 | 0.211 | 0.237 | 0.301 | 0.216 | 0.214 | 0.218 | 1.417 | 0.807 |
| POW. | 24 | **0.269** | 0.368 | 0.280 | 0.274 | 0.290 | 0.359 | 0.345 | 0.285 | 0.281 | 0.308 | 0.345 | 0.282 | 0.284 | 0.282 | 1.438 | 1.065 |
| POW. | 48 | **0.357** | 0.421 | 0.368 | 0.368 | 0.405 | 0.441 | 0.459 | 0.370 | 0.372 | 0.387 | 0.393 | 0.368 | 0.369 | 0.368 | 1.501 | 1.314 |
| POW. | 96 | **0.443** | 0.469 | 0.445 | 0.444 | 0.465 | 0.484 | 0.548 | 0.450 | 0.450 | 0.470 | 0.476 | 0.451 | 0.445 | 0.448 | 1.584 | 1.588 |
| POW. | Avg. | **0.318** | 0.397 | 0.327 | 0.323 | 0.346 | 0.397 | 0.411 | 0.330 | 0.329 | 0.351 | 0.379 | 0.329 | 0.328 | 0.329 | 1.485 | 1.193 |
| TAT. | 12 | **0.113** | 0.222 | 0.126 | 0.118 | 0.132 | 0.193 | 0.147 | 0.128 | 0.128 | 0.136 | 0.170 | 0.131 | 0.125 | 0.124 | 0.352 | 0.202 |
| TAT. | 24 | **0.161** | 0.258 | 0.178 | 0.172 | 0.185 | 0.232 | 0.203 | 0.187 | 0.186 | 0.188 | 0.212 | 0.181 | 0.177 | 0.176 | 0.368 | 0.259 |
| TAT. | 48 | **0.242** | 0.321 | 0.262 | 0.250 | 0.259 | 0.304 | 0.277 | 0.285 | 0.264 | 0.263 | 0.290 | 0.262 | 0.264 | 0.267 | 0.391 | 0.352 |
| TAT. | 96 | **0.364** | 0.400 | 0.390 | 0.388 | 0.386 | 0.471 | 0.379 | 0.413 | 0.383 | 0.405 | 0.410 | 0.395 | 0.393 | 0.393 | 0.415 | 0.435 |
| TAT. | Avg. | **0.220** | 0.300 | 0.239 | 0.232 | 0.241 | 0.300 | 0.252 | 0.253 | 0.240 | 0.248 | 0.271 | 0.242 | 0.240 | 0.240 | 0.381 | 0.312 |
| TEC. | 12 | **0.044** | 0.094 | 0.048 | 0.047 | 0.052 | 0.071 | 0.060 | 0.047 | 0.046 | 0.053 | 0.084 | 0.050 | 0.047 | 0.047 | 0.158 | 0.177 |
| TEC. | 24 | **0.063** | 0.115 | 0.071 | 0.066 | 0.072 | 0.089 | 0.081 | 0.067 | 0.077 | 0.079 | 0.099 | 0.069 | 0.068 | 0.066 | 0.168 | 0.188 |
| TEC. | 48 | 0.101 | 0.134 | 0.104 | 0.099 | 0.104 | 0.115 | 0.111 | 0.103 | **0.096** | 0.113 | 0.108 | 0.103 | 0.099 | 0.101 | 0.175 | 0.248 |
| TEC. | 96 | 0.148 | 0.181 | 0.162 | 0.155 | 0.172 | 0.166 | 0.152 | 0.160 | 0.149 | 0.169 | 0.160 | 0.157 | 0.161 | 0.155 | 0.181 | 0.295 |
| TEC. | Avg. | **0.089** | 0.131 | 0.096 | 0.092 | 0.100 | 0.110 | 0.101 | 0.094 | 0.092 | 0.104 | 0.113 | 0.095 | 0.094 | 0.092 | 0.170 | 0.227 |
| TIT. | 12 | **0.045** | 0.088 | 0.053 | 0.058 | 0.050 | 0.080 | 0.061 | 0.053 | 0.049 | 0.061 | 0.084 | 0.059 | 0.049 | 0.050 | 0.184 | 0.403 |
| TIT. | 24 | **0.059** | 0.105 | 0.075 | 0.080 | 0.072 | 0.100 | 0.074 | 0.079 | 0.065 | 0.079 | 0.114 | 0.080 | 0.072 | 0.072 | 0.194 | 0.365 |
| TIT. | 48 | **0.083** | 0.128 | 0.108 | 0.120 | 0.111 | 0.133 | 0.095 | 0.112 | 0.098 | 0.105 | 0.115 | 0.135 | 0.109 | 0.111 | 0.195 | 0.271 |
| TIT. | 96 | **0.114** | 0.166 | 0.151 | 0.167 | 0.144 | 0.166 | 0.152 | 0.158 | 0.149 | 0.150 | 0.173 | 0.183 | 0.150 | 0.145 | 0.221 | 0.310 |
| TIT. | Avg. | **0.075** | 0.122 | 0.096 | 0.106 | 0.094 | 0.120 | 0.096 | 0.100 | 0.090 | 0.099 | 0.121 | 0.114 | 0.095 | 0.094 | 0.199 | 0.337 |

Table 15: **MAPE (Mean Absolute Percentage Error)** results on the Nifty50 dataset, averaged over three random seeds. For brevity, stocks are represented by the first three letters of their alphabetical names.

| Stock | Horizon | Hipeen | DLinear | RLinear | SAN | Leddam | DDN | FAN | TimeMixer | PatchTST | TiDE | TimesNet | CycleNet | Peri-mid. | FRNet | Smamba | STF |
|---|---|---|---|---|---|---|---|---|---|---|---|---|---|---|---|---|---|
| BAJ. | 12 | 0.154 | 0.282 | 0.15 | **0.139** | 0.182 | 0.23 | 0.193 | 0.157 | 0.16 | 0.162 | 0.25 | 0.166 | 0.151 | 0.148 | 0.482 | 0.352 |
| BAJ. | 24 | 0.22 | 0.322 | 0.223 | **0.212** | 0.247 | 0.279 | 0.253 | 0.23 | 0.242 | 0.247 | 0.283 | 0.237 | 0.227 | 0.227 | 0.498 | 0.384 |
| BAJ. | 48 | 0.308 | 0.382 | 0.313 | **0.3** | 0.317 | 0.361 | 0.324 | 0.328 | 0.32 | 0.346 | 0.349 | 0.324 | 0.317 | 0.317 | 0.529 | 0.441 |
| BAJ. | 96 | 0.405 | 0.444 | 0.404 | **0.379** | 0.418 | 0.417 | 0.391 | 0.422 | 0.411 | 0.423 | 0.452 | 0.41 | 0.411 | 0.408 | 0.571 | 0.503 |
| BAJ. | avg | 0.272 | 0.358 | 0.273 | **0.257** | 0.291 | 0.322 | 0.29 | 0.284 | 0.283 | 0.295 | 0.334 | 0.284 | 0.276 | 0.275 | 0.52 | 0.42 |
| HDF. | 12 | **0.08** | 0.157 | 0.082 | 0.086 | 0.085 | 0.114 | 0.099 | 0.09 | 0.088 | 0.089 | 0.117 | 0.092 | 0.082 | 0.084 | 0.468 | 0.484 |
| HDF. | 24 | **0.107** | 0.183 | 0.109 | 0.116 | 0.113 | 0.151 | 0.147 | 0.122 | 0.115 | 0.117 | 0.144 | 0.119 | 0.109 | 0.113 | 0.44 | 0.46 |
| HDF. | 48 | **0.15** | 0.232 | 0.156 | 0.172 | 0.159 | 0.184 | 0.188 | 0.17 | 0.16 | 0.163 | 0.176 | 0.167 | 0.156 | 0.159 | 0.435 | 0.479 |
| HDF. | 96 | **0.198** | 0.294 | 0.219 | 0.24 | 0.224 | 0.24 | 0.266 | 0.238 | 0.229 | 0.224 | 0.244 | 0.232 | 0.219 | 0.223 | 0.442 | 0.318 |
| HDF. | avg | **0.134** | 0.216 | 0.142 | 0.153 | 0.145 | 0.172 | 0.175 | 0.155 | 0.148 | 0.148 | 0.17 | 0.153 | 0.141 | 0.145 | 0.446 | 0.436 |
| HER. | 12 | 0.063 | 0.11 | 0.063 | **0.061** | 0.066 | 0.084 | 0.082 | 0.062 | 0.062 | 0.067 | 0.083 | 0.063 | 0.061 | 0.061 | 0.528 | 0.4 |
| HER. | 24 | 0.087 | 0.126 | **0.084** | 0.084 | 0.087 | 0.1 | 0.106 | 0.084 | 0.087 | 0.089 | 0.1 | 0.087 | 0.084 | 0.084 | 0.536 | 0.457 |
| HER. | 48 | 0.117 | 0.149 | 0.114 | 0.116 | 0.117 | 0.124 | 0.134 | 0.118 | 0.118 | 0.121 | 0.128 | **0.113** | 0.113 | 0.114 | 0.551 | 0.509 |
| HER. | 96 | 0.156 | 0.19 | 0.156 | 0.159 | 0.154 | 0.165 | 0.179 | **0.152** | 0.155 | 0.158 | 0.167 | 0.154 | 0.156 | 0.155 | 0.555 | 0.545 |
| HER. | avg | 0.106 | 0.144 | 0.104 | 0.105 | 0.106 | 0.118 | 0.125 | **0.103** | 0.106 | 0.109 | 0.119 | 0.104 | 0.103 | 0.104 | 0.542 | 0.478 |
| HIN. | 12 | **0.052** | 0.11 | 0.054 | 0.055 | 0.054 | 0.086 | 0.084 | 0.054 | 0.054 | 0.06 | 0.082 | 0.056 | 0.054 | 0.053 | 0.147 | 0.305 |
| HIN. | 24 | 0.073 | 0.134 | 0.071 | 0.072 | 0.073 | 0.086 | 0.094 | 0.072 | 0.077 | 0.075 | 0.087 | 0.074 | **0.07** | 0.071 | 0.163 | 0.311 |
| HIN. | 48 | 0.1 | 0.14 | 0.1 | 0.099 | 0.099 | 0.124 | 0.115 | 0.103 | 0.1 | 0.103 | 0.104 | 0.1 | **0.096** | 0.098 | 0.17 | 0.295 |
| HIN. | 96 | 0.139 | 0.159 | 0.125 | 0.127 | 0.128 | 0.139 | 0.136 | 0.131 | 0.132 | 0.145 | 0.143 | 0.135 | **0.124** | 0.126 | 0.182 | 0.291 |
| HIN. | avg | 0.091 | 0.136 | 0.088 | 0.088 | 0.088 | 0.109 | 0.107 | 0.09 | 0.091 | 0.096 | 0.104 | 0.091 | **0.086** | 0.087 | 0.165 | 0.3 |
| LT | 12 | **0.173** | 0.299 | 0.188 | 0.181 | 0.204 | 0.272 | 0.214 | 0.201 | 0.19 | 0.194 | 0.281 | 0.209 | 0.181 | 0.191 | 0.527 | 0.329 |
| LT | 24 | **0.256** | 0.345 | 0.267 | 0.265 | 0.29 | 0.359 | 0.272 | 0.309 | 0.286 | 0.273 | 0.329 | 0.311 | 0.265 | 0.277 | 0.561 | 0.419 |
| LT | 48 | **0.333** | 0.41 | 0.383 | 0.362 | 0.418 | 0.464 | 0.354 | 0.427 | 0.448 | 0.396 | 0.445 | 0.463 | 0.382 | 0.388 | 0.571 | 0.447 |
| LT | 96 | **0.432** | 0.484 | 0.494 | 0.507 | 0.525 | 0.505 | 0.482 | 0.533 | 0.57 | 0.593 | 0.553 | 0.611 | 0.507 | 0.497 | 0.565 | 0.448 |
| LT | avg | **0.299** | 0.385 | 0.333 | 0.329 | 0.359 | 0.4 | 0.331 | 0.367 | 0.373 | 0.364 | 0.402 | 0.399 | 0.334 | 0.338 | 0.556 | 0.411 |
| MAR. | 12 | **0.049** | 0.113 | 0.051 | 0.054 | 0.054 | 0.094 | 0.073 | 0.052 | 0.05 | 0.055 | 0.072 | 0.051 | 0.05 | 0.05 | 0.688 | 0.603 |
| MAR. | 24 | **0.067** | 0.131 | 0.069 | 0.071 | 0.071 | 0.092 | 0.097 | 0.076 | 0.073 | 0.071 | 0.092 | 0.071 | 0.068 | 0.069 | 0.701 | 0.648 |
| MAR. | 48 | **0.098** | 0.158 | 0.101 | 0.108 | 0.103 | 0.121 | 0.136 | 0.111 | 0.101 | 0.103 | 0.11 | 0.101 | 0.101 | 0.102 | 0.719 | 0.696 |
| MAR. | 96 | **0.139** | 0.211 | 0.145 | 0.171 | 0.144 | 0.154 | 0.227 | 0.147 | 0.145 | 0.145 | 0.155 | 0.14 | 0.155 | 0.144 | 0.745 | 0.754 |
| MAR. | avg | **0.088** | 0.153 | 0.091 | 0.101 | 0.093 | 0.115 | 0.134 | 0.096 | 0.092 | 0.094 | 0.107 | 0.091 | 0.093 | 0.091 | 0.713 | 0.675 |
| NTP. | 12 | 0.569 | 0.792 | 0.508 | **0.45** | 0.459 | 0.778 | 0.586 | 0.457 | 0.511 | 0.505 | 0.706 | 0.455 | 0.478 | 0.504 | 1.95 | 0.709 |
| NTP. | 24 | 0.685 | 0.943 | 0.67 | 0.69 | **0.643** | 0.747 | 0.818 | 0.649 | 0.686 | 0.675 | 0.833 | 0.666 | 0.697 | 0.655 | 2.166 | 0.91 |
| NTP. | 48 | 1.06 | 1.126 | 0.989 | 1.051 | **0.945** | 1.094 | 1.143 | 0.998 | 0.979 | 0.968 | 1.031 | 0.973 | 0.984 | 1.002 | 2.479 | 1.317 |
| NTP. | 96 | 1.371 | 1.345 | 1.28 | 1.328 | **1.287** | 1.423 | 1.442 | 1.463 | 1.501 | 1.324 | 1.342 | 1.294 | 1.345 | 1.363 | 3.014 | 2.666 |
| NTP. | avg | 0.921 | 1.052 | 0.862 | 0.88 | **0.834** | 1.01 | 0.997 | 0.892 | 0.919 | 0.868 | 0.978 | 0.847 | 0.876 | 0.881 | 2.402 | 1.4 |
| POW. | 12 | **0.085** | 0.144 | 0.091 | 0.087 | 0.095 | 0.129 | 0.124 | 0.091 | 0.088 | 0.1 | 0.128 | 0.091 | 0.09 | 0.092 | 0.514 | 0.293 |
| POW. | 24 | **0.114** | 0.16 | 0.119 | 0.116 | 0.123 | 0.151 | 0.151 | 0.12 | 0.12 | 0.131 | 0.148 | 0.12 | 0.119 | 0.119 | 0.524 | 0.388 |
| POW. | 48 | **0.155** | 0.184 | 0.158 | 0.158 | 0.176 | 0.187 | 0.207 | 0.159 | 0.159 | 0.166 | 0.172 | 0.158 | 0.158 | 0.158 | 0.555 | 0.486 |
| POW. | 96 | 0.198 | 0.205 | 0.197 | **0.195** | 0.204 | 0.213 | 0.249 | 0.2 | 0.2 | 0.209 | 0.21 | 0.2 | 0.197 | 0.198 | 0.599 | 0.602 |
| POW. | avg | **0.138** | 0.173 | 0.141 | 0.139 | 0.15 | 0.17 | 0.182 | 0.142 | 0.142 | 0.152 | 0.165 | 0.142 | 0.141 | 0.142 | 0.548 | 0.442 |
| TAT. | 12 | **0.557** | 1.111 | 0.709 | 0.637 | 0.71 | 0.984 | 0.74 | 0.783 | 0.616 | 0.824 | 0.833 | 0.75 | 0.744 | 0.688 | 1.423 | 1.184 |
| TAT. | 24 | 0.937 | 1.282 | 1.052 | 0.965 | 0.937 | 1.127 | 1.366 | 1.066 | **0.936** | 1.11 | 1.365 | 0.937 | 1.041 | 1.023 | 1.46 | 1.673 |
| TAT. | 48 | 1.484 | **1.348** | 1.593 | 1.684 | 1.456 | 1.974 | 1.774 | 1.369 | 1.587 | 1.567 | 1.64 | 1.446 | 1.661 | 1.668 | 1.54 | 2.189 |
| TAT. | 96 | 1.838 | 1.816 | 2.075 | 2.129 | **1.748** | 2.969 | 2.006 | 2.166 | 1.967 | 2.332 | 2.21 | 1.961 | 2.052 | 2.071 | 1.807 | 2.355 |
| TAT. | avg | **1.204** | 1.389 | 1.357 | 1.354 | 1.213 | 1.764 | 1.472 | 1.346 | 1.277 | 1.458 | 1.512 | 1.273 | 1.374 | 1.363 | 1.557 | 1.85 |
| TEC. | 12 | 0.554 | 1.012 | 0.561 | 0.601 | 0.537 | 0.7 | 0.625 | **0.52** | 0.543 | 0.612 | 0.894 | 0.586 | 0.523 | 0.531 | 1.185 | 4.244 |
| TEC. | 24 | 0.716 | 1.273 | 0.775 | **0.638** | 0.711 | 0.899 | 0.795 | 0.676 | 0.896 | 0.916 | 1.108 | 0.695 | 0.679 | 0.674 | 1.407 | 3.887 |
| TEC. | 48 | 0.956 | 1.485 | 1.28 | 1.005 | 1.166 | 1.16 | 1.05 | **0.811** | 0.879 | 1.132 | 0.83 | 1.045 | 1.021 | 0.915 | 1.634 | 3.5 |
| TEC. | 96 | 1.308 | 2.171 | 2.434 | 1.88 | 2.302 | 1.923 | 1.385 | 1.04 | 1.032 | 2.262 | 1.311 | 2.071 | 2.313 | 1.092 | 1.871 | 2.99 |
| TEC. | avg | 0.884 | 1.485 | 1.262 | 1.031 | 1.179 | 1.171 | 0.964 | **0.762** | 0.837 | 1.23 | 1.036 | 1.099 | 1.134 | 0.803 | 1.524 | 3.655 |
| TIT. | 12 | **0.171** | 0.325 | 0.195 | 0.207 | 0.182 | 0.29 | 0.236 | 0.198 | 0.18 | 0.226 | 0.304 | 0.219 | 0.183 | 0.184 | 0.667 | 1.193 |
| TIT. | 24 | **0.226** | 0.376 | 0.274 | 0.277 | 0.263 | 0.361 | 0.296 | 0.289 | 0.244 | 0.287 | 0.413 | 0.291 | 0.263 | 0.263 | 0.698 | 1.127 |
| TIT. | 48 | **0.311** | 0.441 | 0.395 | 0.417 | 0.41 | 0.493 | 0.376 | 0.413 | 0.365 | 0.385 | 0.417 | 0.489 | 0.399 | 0.404 | 0.729 | 0.845 |
| TIT. | 96 | **0.418** | 0.549 | 0.569 | 0.599 | 0.54 | 0.637 | 0.552 | 0.562 | 0.523 | 0.566 | 0.645 | 0.66 | 0.566 | 0.544 | 0.812 | 0.911 |
| TIT. | avg | **0.282** | 0.423 | 0.358 | 0.375 | 0.349 | 0.445 | 0.365 | 0.365 | 0.328 | 0.366 | 0.444 | 0.415 | 0.353 | 0.349 | 0.726 | 1.019 |

Table 16: **RMSE (Root Mean Squared Error)** results on the Nifty50 dataset, averaged over three random seeds. For brevity, stocks are represented by the first three letters of their alphabetical names.

| Stock | Horizon | Hipeen | DLinear | RLinear | SAN | Leddam | DDN | FAN | TimeMixer | PatchTST | TiDE | TimesNet | CycleNet | Peri-mid. | FRNet | Smamba | STF |
|---|---|---|---|---|---|---|---|---|---|---|---|---|---|---|---|---|---|
| ADA. | 12 | **0.126** | 0.282 | 0.144 | 0.132 | 0.149 | 0.167 | 0.175 | 0.134 | 0.139 | 0.151 | 0.258 | 0.146 | 0.131 | 0.142 | 0.558 | 0.258 |
| ADA. | 24 | 0.182 | 0.324 | 0.205 | **0.178** | 0.241 | 0.252 | 0.225 | 0.19 | 0.23 | 0.196 | 0.324 | 0.222 | 0.192 | 0.206 | 0.544 | 0.295 |
| ADA. | 48 | 0.276 | 0.368 | 0.302 | **0.265** | 0.307 | 0.307 | 0.302 | 0.309 | 0.312 | 0.29 | 0.282 | 0.306 | 0.296 | 0.304 | 0.513 | 0.359 |
| ADA. | 96 | 0.343 | 0.38 | 0.366 | 0.359 | 0.361 | 0.384 | **0.342** | 0.379 | 0.387 | 0.364 | 0.373 | 0.374 | 0.369 | 0.361 | 0.474 | 0.445 |
| ADA. | avg | **0.232** | 0.339 | 0.254 | 0.234 | 0.264 | 0.278 | 0.261 | 0.253 | 0.267 | 0.25 | 0.309 | 0.262 | 0.247 | 0.253 | 0.522 | 0.339 |
| BAJ. | 12 | 0.235 | 0.425 | 0.234 | **0.222** | 0.26 | 0.358 | 0.286 | 0.24 | 0.239 | 0.257 | 0.362 | 0.243 | 0.236 | 0.231 | 1.01 | 0.856 |
| BAJ. | 24 | 0.327 | 0.493 | 0.336 | **0.323** | 0.353 | 0.414 | 0.371 | 0.347 | 0.362 | 0.372 | 0.407 | 0.342 | 0.341 | 0.337 | 1.029 | 0.848 |
| BAJ. | 48 | **0.446** | 0.607 | 0.465 | 0.456 | 0.461 | 0.535 | 0.487 | 0.473 | 0.47 | 0.515 | 0.501 | 0.475 | 0.471 | 0.468 | 1.072 | 0.912 |
| BAJ. | 96 | **0.572** | 0.742 | 0.592 | 0.577 | 0.601 | 0.598 | 0.58 | 0.632 | 0.587 | 0.621 | 0.644 | 0.607 | 0.602 | 0.596 | 1.096 | 0.97 |
| BAJ. | avg | **0.395** | 0.567 | 0.407 | 0.394 | 0.419 | 0.476 | 0.431 | 0.423 | 0.415 | 0.441 | 0.478 | 0.417 | 0.412 | 0.408 | 1.052 | 0.897 |
| HDF. | 12 | **0.124** | 0.221 | 0.126 | 0.13 | 0.128 | 0.172 | 0.15 | 0.136 | 0.131 | 0.135 | 0.173 | 0.134 | 0.125 | 0.128 | 0.716 | 0.556 |
| HDF. | 24 | **0.169** | 0.262 | 0.171 | 0.177 | 0.175 | 0.222 | 0.224 | 0.187 | 0.174 | 0.179 | 0.215 | 0.177 | 0.17 | 0.177 | 0.688 | 0.52 |
| HDF. | 48 | **0.24** | 0.336 | 0.247 | 0.251 | 0.251 | 0.279 | 0.291 | 0.266 | 0.251 | 0.253 | 0.267 | 0.26 | 0.246 | 0.252 | 0.687 | 0.522 |
| HDF. | 96 | **0.325** | 0.429 | 0.348 | 0.369 | 0.355 | 0.362 | 0.408 | 0.371 | 0.368 | 0.353 | 0.379 | 0.367 | 0.348 | 0.351 | 0.692 | 0.447 |
| HDF. | avg | **0.215** | 0.312 | 0.223 | 0.235 | 0.227 | 0.259 | 0.268 | 0.24 | 0.231 | 0.23 | 0.259 | 0.234 | 0.222 | 0.227 | 0.696 | 0.511 |
| HER. | 12 | 0.194 | 0.317 | 0.193 | 0.192 | 0.201 | 0.255 | 0.244 | 0.193 | 0.196 | 0.207 | 0.246 | 0.197 | **0.188** | 0.196 | 1.698 | 1.364 |
| HER. | 24 | 0.264 | 0.361 | 0.256 | **0.255** | 0.263 | 0.301 | 0.312 | 0.259 | 0.266 | 0.27 | 0.296 | 0.264 | 0.255 | 0.256 | 1.725 | 1.529 |
| HER. | 48 | 0.344 | 0.429 | 0.338 | 0.346 | 0.343 | 0.364 | 0.39 | 0.354 | 0.357 | 0.357 | 0.381 | **0.337** | 0.337 | 0.338 | 1.775 | 1.669 |
| HER. | 96 | 0.467 | 0.55 | 0.464 | 0.479 | 0.461 | 0.484 | 0.525 | **0.455** | 0.461 | 0.469 | 0.502 | 0.461 | 0.465 | 0.463 | 1.793 | 1.781 |
| HER. | avg | 0.317 | 0.414 | 0.313 | 0.318 | 0.317 | 0.351 | 0.368 | 0.312 | 0.319 | 0.326 | 0.356 | 0.315 | **0.311** | 0.313 | 1.748 | 1.586 |
| HIN. | 12 | **0.034** | 0.069 | 0.036 | 0.036 | 0.036 | 0.057 | 0.054 | 0.036 | 0.035 | 0.039 | 0.053 | 0.037 | 0.036 | 0.035 | 0.095 | 0.186 |
| HIN. | 24 | 0.05 | 0.086 | **0.048** | 0.049 | 0.049 | 0.058 | 0.062 | 0.049 | 0.051 | 0.051 | 0.059 | 0.05 | 0.048 | 0.049 | 0.105 | 0.182 |
| HIN. | 48 | 0.069 | 0.094 | 0.069 | 0.069 | 0.068 | 0.083 | 0.079 | 0.072 | 0.069 | 0.071 | 0.072 | 0.07 | **0.067** | 0.068 | 0.11 | 0.167 |
| HIN. | 96 | 0.097 | 0.111 | 0.092 | 0.093 | 0.093 | 0.101 | 0.096 | 0.094 | 0.095 | 0.108 | 0.104 | 0.097 | **0.091** | 0.093 | 0.119 | 0.17 |
| HIN. | avg | 0.062 | 0.09 | 0.061 | 0.062 | 0.061 | 0.075 | 0.073 | 0.063 | 0.063 | 0.067 | 0.072 | 0.064 | **0.06** | 0.061 | 0.107 | 0.176 |
| LT | 12 | **0.094** | 0.169 | 0.098 | 0.096 | 0.105 | 0.142 | 0.12 | 0.103 | 0.099 | 0.102 | 0.144 | 0.108 | 0.097 | 0.098 | 0.272 | 0.18 |
| LT | 24 | 0.143 | 0.205 | 0.139 | **0.138** | 0.148 | 0.184 | 0.158 | 0.156 | 0.15 | 0.142 | 0.17 | 0.158 | 0.139 | 0.14 | 0.291 | 0.226 |
| LT | 48 | 0.198 | 0.258 | 0.198 | **0.197** | 0.211 | 0.232 | 0.219 | 0.211 | 0.221 | 0.202 | 0.221 | 0.224 | 0.197 | 0.198 | 0.316 | 0.241 |
| LT | 96 | 0.288 | 0.353 | 0.27 | 0.27 | 0.278 | 0.295 | 0.323 | 0.275 | 0.289 | 0.332 | 0.287 | 0.293 | 0.274 | **0.269** | 0.378 | 0.328 |
| LT | avg | 0.181 | 0.246 | 0.176 | **0.175** | 0.185 | 0.213 | 0.205 | 0.186 | 0.19 | 0.195 | 0.206 | 0.196 | 0.177 | 0.176 | 0.314 | 0.244 |
| MAR. | 12 | **0.376** | 0.821 | 0.393 | 0.416 | 0.412 | 0.689 | 0.548 | 0.398 | 0.397 | 0.409 | 0.541 | 0.394 | 0.383 | 0.386 | 4.484 | 4.02 |
| MAR. | 24 | **0.519** | 0.948 | 0.539 | 0.552 | 0.548 | 0.686 | 0.727 | 0.577 | 0.562 | 0.548 | 0.684 | 0.542 | 0.531 | 0.533 | 4.562 | 4.278 |
| MAR. | 48 | **0.75** | 1.138 | 0.778 | 0.821 | 0.774 | 0.883 | 1.005 | 0.828 | 0.773 | 0.784 | 0.817 | 0.771 | 0.78 | 0.784 | 4.669 | 4.56 |
| MAR. | 96 | **1.005** | 1.478 | 1.069 | 1.237 | 1.035 | 1.102 | 1.616 | 1.083 | 1.055 | 1.06 | 1.092 | 1.029 | 1.146 | 1.055 | 4.789 | 4.854 |
| MAR. | avg | **0.662** | 1.096 | 0.695 | 0.757 | 0.692 | 0.84 | 0.974 | 0.722 | 0.697 | 0.7 | 0.783 | 0.684 | 0.71 | 0.689 | 4.626 | 4.428 |
| NTP. | 12 | 0.177 | 0.239 | 0.155 | 0.158 | 0.16 | 0.21 | 0.198 | 0.163 | 0.16 | 0.162 | 0.203 | 0.154 | **0.153** | 0.159 | 0.545 | 0.314 |
| NTP. | 24 | 0.211 | 0.285 | 0.201 | 0.226 | 0.202 | 0.222 | 0.272 | 0.207 | 0.206 | 0.207 | 0.235 | 0.203 | 0.202 | **0.201** | 0.574 | 0.344 |
| NTP. | 48 | 0.28 | 0.348 | 0.256 | 0.307 | **0.254** | 0.274 | 0.345 | 0.264 | 0.269 | 0.262 | 0.276 | 0.258 | 0.258 | 0.259 | 0.61 | 0.492 |
| NTP. | 96 | 0.376 | 0.47 | **0.326** | 0.398 | 0.327 | 0.353 | 0.491 | 0.367 | 0.372 | 0.333 | 0.352 | 0.33 | 0.336 | 0.337 | 0.682 | 0.763 |
| NTP. | avg | 0.261 | 0.336 | **0.235** | 0.272 | 0.236 | 0.265 | 0.326 | 0.25 | 0.252 | 0.241 | 0.266 | 0.236 | 0.237 | 0.239 | 0.603 | 0.478 |
| POW. | 12 | **0.28** | 0.422 | 0.297 | 0.288 | 0.305 | 0.393 | 0.372 | 0.296 | 0.294 | 0.318 | 0.39 | 0.296 | 0.292 | 0.298 | 1.54 | 0.93 |
| POW. | 24 | **0.364** | 0.469 | 0.377 | 0.375 | 0.388 | 0.471 | 0.438 | 0.386 | 0.381 | 0.402 | 0.449 | 0.378 | 0.381 | 0.38 | 1.557 | 1.184 |
| POW. | 48 | **0.475** | 0.547 | 0.486 | 0.486 | 0.515 | 0.584 | 0.581 | 0.487 | 0.498 | 0.506 | 0.511 | 0.487 | 0.488 | 0.485 | 1.614 | 1.433 |
| POW. | 96 | **0.593** | 0.622 | 0.593 | 0.594 | 0.612 | 0.638 | 0.721 | 0.608 | 0.6 | 0.62 | 0.629 | 0.598 | 0.594 | 0.597 | 1.687 | 1.69 |
| POW. | avg | **0.428** | 0.515 | 0.438 | 0.436 | 0.455 | 0.522 | 0.528 | 0.444 | 0.443 | 0.462 | 0.494 | 0.44 | 0.439 | 0.44 | 1.599 | 1.309 |
| TAT. | 12 | **0.159** | 0.287 | 0.176 | 0.166 | 0.185 | 0.252 | 0.202 | 0.182 | 0.182 | 0.189 | 0.224 | 0.187 | 0.174 | 0.172 | 0.494 | 0.274 |
| TAT. | 24 | **0.228** | 0.335 | 0.245 | 0.239 | 0.257 | 0.304 | 0.27 | 0.256 | 0.262 | 0.255 | 0.281 | 0.256 | 0.241 | 0.243 | 0.505 | 0.344 |
| TAT. | 48 | **0.335** | 0.419 | 0.355 | 0.339 | 0.355 | 0.399 | 0.366 | 0.38 | 0.36 | 0.356 | 0.386 | 0.358 | 0.357 | 0.36 | 0.529 | 0.456 |
| TAT. | 96 | **0.494** | 0.535 | 0.519 | 0.507 | 0.522 | 0.601 | 0.496 | 0.546 | 0.51 | 0.531 | 0.543 | 0.529 | 0.523 | 0.524 | 0.558 | 0.556 |
| TAT. | avg | **0.304** | 0.394 | 0.323 | 0.313 | 0.33 | 0.389 | 0.333 | 0.341 | 0.329 | 0.333 | 0.359 | 0.333 | 0.324 | 0.325 | 0.521 | 0.407 |
| TEC. | 12 | **0.064** | 0.125 | 0.068 | 0.065 | 0.072 | 0.098 | 0.083 | 0.067 | 0.064 | 0.073 | 0.111 | 0.067 | 0.067 | 0.065 | 0.193 | 0.248 |
| TEC. | 24 | **0.093** | 0.153 | 0.1 | 0.094 | 0.1 | 0.124 | 0.112 | 0.094 | 0.105 | 0.109 | 0.133 | 0.095 | 0.097 | 0.093 | 0.204 | 0.254 |
| TEC. | 48 | 0.143 | 0.179 | 0.144 | 0.137 | 0.143 | 0.157 | 0.153 | 0.147 | **0.134** | 0.155 | 0.147 | 0.141 | 0.14 | 0.141 | 0.212 | 0.308 |
| TEC. | 96 | 0.2 | 0.233 | 0.213 | 0.202 | 0.222 | 0.217 | 0.206 | 0.215 | **0.199** | 0.22 | 0.208 | 0.204 | 0.212 | 0.213 | 0.22 | 0.356 |
| TEC. | avg | **0.125** | 0.173 | 0.131 | 0.125 | 0.134 | 0.149 | 0.139 | 0.131 | 0.126 | 0.139 | 0.15 | 0.127 | 0.129 | 0.128 | 0.207 | 0.291 |
| TIT. | 12 | **0.066** | 0.111 | 0.073 | 0.079 | 0.07 | 0.102 | 0.084 | 0.073 | 0.069 | 0.081 | 0.106 | 0.08 | 0.069 | 0.071 | 0.237 | 0.596 |
| TIT. | 24 | **0.088** | 0.133 | 0.1 | 0.108 | 0.097 | 0.126 | 0.103 | 0.103 | 0.092 | 0.105 | 0.139 | 0.105 | 0.097 | 0.097 | 0.248 | 0.522 |
| TIT. | 48 | **0.116** | 0.162 | 0.139 | 0.155 | 0.14 | 0.165 | 0.127 | 0.141 | 0.13 | 0.136 | 0.146 | 0.165 | 0.14 | 0.141 | 0.249 | 0.364 |
| TIT. | 96 | **0.152** | 0.212 | 0.189 | 0.213 | 0.181 | 0.205 | 0.194 | 0.199 | 0.196 | 0.188 | 0.209 | 0.23 | 0.189 | 0.182 | 0.28 | 0.401 |
| TIT. | avg | **0.105** | 0.155 | 0.125 | 0.139 | 0.122 | 0.149 | 0.127 | 0.129 | 0.122 | 0.128 | 0.15 | 0.145 | 0.124 | 0.122 | 0.254 | 0.471 |

Table 17: **Trading-based metrics** results on the Nifty50 dataset, averaged over {12, 24, 48,96} horizon lengths and three random seeds. The scales differ substantially across horizons, and due to the nature of risk-adjusted return metrics—which can sometimes yield infinite values—we report the results using rank-based averaging. For brevity, stock names are abbreviated to the first three letters of their alphabetical names.

| Stock | Metric | Hipeen | DLinear | RLinear | SAN | Leddam | DDN | FAN | TimeMixer | PatchTST | TiDE | TimesNet | CycleNet | Peri-mid. | FRNet | Smamba | STF |
|---|---|---|---|---|---|---|---|---|---|---|---|---|---|---|---|---|---|
| ADA. | Revenue | **3.3** | 10.5 | 11.3 | 4.5 | 9.3 | 7.5 | 5.0 | 10.8 | 6.5 | 5.5 | 14.0 | 11.8 | 9.5 | 10.5 | 8.5 | 7.8 |
| ADA. | Drawdown | **2.5** | 11.5 | 8.8 | 6.3 | 6.3 | 11.5 | 4.3 | 12.3 | 11.5 | 5.0 | 10.8 | 13.5 | 9.8 | 8.5 | 7.0 | 6.8 |
| ADA. | Sharpe | **3.3** | 8.8 | 11.0 | 3.5 | 11.3 | 8.3 | 4.5 | 10.5 | 7.3 | 5.8 | 14.3 | 12.3 | 9.3 | 11.8 | 6.5 | 8.0 |
| ADA. | Sortino | **3.3** | 8.8 | 11.3 | 3.8 | 11.0 | 7.8 | 4.5 | 10.8 | 7.5 | 5.8 | 14.5 | 12.3 | 9.5 | 10.8 | 6.8 | 8.0 |
| ADA. | Calmar | **3.5** | 9.0 | 11.0 | 4.0 | 10.0 | 7.0 | 5.8 | 10.3 | 6.3 | 5.3 | 14.3 | 12.8 | 9.0 | 10.3 | 10.0 | 7.8 |
| BAJ. | Revenue | **1.3** | 11.0 | 7.5 | 4.8 | 8.5 | 11.0 | 3.0 | 11.5 | 6.5 | 13.0 | 9.5 | 9.8 | 11.8 | 8.3 | 10.3 | 8.5 |
| BAJ. | Drawdown | **3.0** | 11.8 | 5.8 | 6.3 | 3.5 | 10.3 | 5.8 | 11.5 | 9.3 | 13.0 | 7.3 | 9.3 | 10.8 | 7.0 | 12.5 | 9.3 |
| BAJ. | Sharpe | **1.8** | 9.3 | 7.0 | 4.8 | 6.5 | 10.5 | 3.0 | 11.5 | 8.3 | 14.3 | 9.3 | 9.8 | 13.0 | 9.0 | 9.5 | 8.8 |
| BAJ. | Sortino | **1.5** | 9.5 | 7.0 | 4.8 | 6.3 | 11.3 | 3.0 | 11.5 | 8.3 | 14.3 | 9.5 | 9.5 | 12.5 | 9.0 | 9.5 | 8.8 |
| BAJ. | Calmar | **1.3** | 10.0 | 7.3 | 5.5 | 7.5 | 11.0 | 3.3 | 11.3 | 6.8 | 14.3 | 10.5 | 10.0 | 12.0 | 7.5 | 9.8 | 8.3 |
| HDF. | Revenue | 2.8 | 15.5 | 8.8 | 9.8 | 6.5 | 3.0 | 10.8 | 12.3 | 8.0 | 9.3 | 9.8 | 9.8 | 8.5 | 10.0 | 10.3 | **1.3** |
| HDF. | Drawdown | 6.8 | 16.0 | 8.3 | 9.8 | 4.0 | 4.3 | 13.3 | 11.5 | 9.5 | 7.3 | 10.3 | 8.5 | 7.0 | 6.5 | 11.3 | **2.0** |
| HDF. | Sharpe | 2.8 | 14.3 | 9.8 | 10.3 | 6.3 | 3.0 | 9.5 | 12.0 | 8.3 | 10.8 | 10.0 | 10.8 | 8.5 | 10.5 | 8.3 | **1.3** |
| HDF. | Sortino | 2.8 | 15.3 | 9.5 | 10.3 | 6.3 | 3.0 | 10.0 | 11.8 | 8.3 | 10.8 | 9.8 | 10.3 | 8.5 | 10.3 | 8.3 | **1.3** |
| HDF. | Calmar | 2.8 | 15.3 | 10.0 | 10.0 | 6.5 | 3.0 | 10.3 | 11.0 | 7.5 | 11.0 | 10.0 | 9.3 | 8.5 | 11.0 | 8.8 | **1.3** |
| HER. | Revenue | 11.3 | 15.3 | 7.5 | 13.5 | 7.5 | 12.5 | 13.8 | **2.0** | 4.5 | 10.3 | 10.5 | 6.3 | 6.8 | 4.5 | 7.3 | 2.8 |
| HER. | Drawdown | 8.5 | 11.8 | 8.8 | 11.8 | 5.5 | 8.0 | 14.3 | **3.8** | 7.5 | 6.8 | 6.5 | 4.3 | 8.8 | 5.5 | 13.3 | 11.3 |
| HER. | Sharpe | 10.8 | 14.8 | 7.0 | 13.3 | 7.3 | 13.3 | 13.8 | **2.0** | 5.0 | 11.0 | 10.8 | 6.0 | 6.5 | 4.8 | 6.8 | 3.3 |
| HER. | Sortino | 11.0 | 15.0 | 7.3 | 13.5 | 7.5 | 13.3 | 14.0 | **2.0** | 5.0 | 10.5 | 10.8 | 6.3 | 6.5 | 4.3 | 6.5 | 2.8 |
| HER. | Calmar | 11.0 | 15.3 | 7.8 | 13.3 | 7.3 | 12.8 | 13.8 | **1.3** | 5.0 | 10.5 | 10.3 | 5.5 | 7.0 | 5.0 | 7.5 | 3.0 |
| HIN. | Revenue | 3.3 | 8.0 | 14.8 | 8.3 | 9.8 | 10.3 | 11.0 | 6.0 | 4.8 | 12.0 | 14.0 | 5.5 | 10.5 | 13.3 | **1.3** | 3.5 |
| HIN. | Drawdown | 10.3 | 10.8 | 7.5 | 11.5 | 2.8 | 9.8 | 13.3 | 7.8 | 6.3 | 9.8 | 10.8 | 4.8 | 6.0 | 10.5 | **2.0** | 12.5 |
| HIN. | Sharpe | 3.3 | 7.8 | 15.0 | 8.8 | 9.5 | 10.8 | 9.3 | 6.5 | 4.0 | 12.8 | 13.8 | 5.0 | 10.5 | 13.5 | **1.3** | 4.5 |
| HIN. | Sortino | 3.3 | 7.8 | 15.3 | 8.5 | 10.0 | 10.8 | 9.3 | 6.3 | 3.8 | 12.5 | 14.0 | 5.5 | 10.5 | 13.5 | **1.3** | 4.0 |
| HIN. | Calmar | 4.3 | 8.3 | 14.8 | 9.0 | 9.8 | 10.3 | 11.3 | 4.8 | 4.0 | 11.8 | 13.8 | 5.3 | 10.3 | 13.5 | **1.3** | 4.0 |
| LT | Revenue | 8.0 | 4.8 | 10.3 | **1.8** | 14.5 | 13.3 | 4.5 | 9.8 | 6.3 | 12.8 | 12.5 | 14.5 | 10.5 | 6.8 | 3.3 | 2.8 |
| LT | Drawdown | 15.3 | 9.0 | 4.5 | 5.8 | 10.8 | 10.0 | 10.5 | 11.0 | 6.3 | 6.5 | 7.3 | 14.0 | 11.5 | 6.0 | 4.8 | **3.0** |
| LT | Sharpe | 7.8 | 4.8 | 11.3 | 2.0 | 13.8 | 12.5 | 4.5 | 10.0 | 6.3 | 13.5 | 13.5 | 13.8 | 10.0 | 6.8 | 3.3 | 2.5 |
| LT | Sortino | 7.5 | 4.8 | 10.8 | **1.8** | 14.0 | 12.8 | 4.5 | 9.8 | 6.3 | 14.0 | 13.5 | 13.5 | 10.0 | 7.0 | 3.3 | 2.8 |
| LT | Calmar | 7.8 | 5.0 | 11.5 | **1.5** | 14.5 | 13.0 | 6.3 | 9.8 | 5.5 | 12.0 | 13.0 | 14.0 | 10.0 | 6.8 | 2.8 | 2.8 |
| MAR. | Revenue | 8.3 | 13.5 | 8.8 | 12.8 | **3.5** | 8.5 | 15.5 | 6.3 | 6.3 | 10.0 | 8.0 | 8.0 | 6.5 | 5.8 | 7.3 | 7.3 |
| MAR. | Drawdown | 6.0 | 14.5 | 9.5 | 13.5 | **4.0** | 8.5 | 13.8 | 7.8 | 5.3 | 7.0 | 5.8 | 8.0 | 7.5 | 5.8 | 10.0 | 10.0 |
| MAR. | Sharpe | 9.5 | 13.5 | 8.0 | 12.5 | **3.5** | 8.5 | 15.0 | 5.8 | 6.3 | 11.3 | 8.3 | 8.5 | 6.8 | 5.8 | 6.5 | 6.5 |
| MAR. | Sortino | 9.5 | 13.8 | 8.0 | 12.8 | **3.5** | 8.5 | 15.0 | 5.8 | 6.3 | 11.0 | 8.3 | 8.3 | 6.8 | 5.8 | 6.5 | 6.5 |
| MAR. | Calmar | 8.5 | 13.5 | 8.3 | 12.8 | **3.5** | 8.5 | 15.3 | 5.8 | 6.3 | 10.0 | 8.3 | 9.0 | 6.3 | 5.8 | 7.3 | 7.3 |
| NTPC | Revenue | 13.3 | 9.5 | 5.3 | 15.8 | **2.5** | 9.5 | 14.3 | 7.0 | 6.8 | 8.0 | 8.0 | 2.8 | 8.0 | 7.8 | 11.8 | 6.0 |
| NTPC | Drawdown | 13.5 | 13.0 | 3.3 | 15.8 | **1.8** | 9.0 | 13.8 | 8.5 | 8.8 | 5.8 | 7.5 | 5.5 | 5.3 | 4.0 | 12.8 | 6.0 |
| NTPC | Sharpe | 13.0 | 9.5 | 5.0 | 15.3 | **2.5** | 10.5 | 13.8 | 8.0 | 6.8 | 8.8 | 8.3 | 3.0 | 7.0 | 7.5 | 11.3 | 6.0 |
| NTPC | Sortino | 13.0 | 9.5 | 4.5 | 15.3 | **2.8** | 10.3 | 13.5 | 7.5 | 7.0 | 8.5 | 8.5 | 3.5 | 6.8 | 8.0 | 11.5 | 6.0 |
| NTPC | Calmar | 13.0 | 9.8 | 4.8 | 15.5 | **2.0** | 9.3 | 14.0 | 7.5 | 7.8 | 7.8 | 8.5 | 3.0 | 6.8 | 7.8 | 12.3 | 6.5 |
| POW. | Revenue | **2.0** | 5.8 | 5.0 | 9.0 | 9.3 | 9.8 | 11.3 | 8.3 | 8.3 | 12.8 | 3.8 | 5.3 | 10.3 | 4.5 | 16.0 | 15.0 |
| POW. | Drawdown | 4.3 | 7.8 | 4.8 | 7.8 | 6.3 | 9.5 | 13.0 | 9.3 | 9.5 | 8.3 | 4.5 | 5.5 | 11.5 | **3.3** | 16.0 | 15.0 |
| POW. | Sharpe | 3.5 | 7.8 | 4.8 | 9.0 | 6.3 | 10.8 | 12.3 | 8.0 | 8.8 | 11.5 | 4.3 | 5.0 | 11.0 | **2.3** | 16.0 | 15.0 |
| POW. | Sortino | 4.3 | 8.3 | 4.0 | 8.5 | 6.5 | 11.5 | 12.5 | 8.0 | 8.3 | 11.3 | 4.5 | 4.8 | 10.5 | **2.3** | 16.0 | 15.0 |
| POW. | Calmar | 4.3 | 7.3 | 3.3 | 8.5 | 7.3 | 10.3 | 12.8 | 7.5 | 9.5 | 11.5 | 4.3 | 5.5 | 10.5 | **2.8** | 16.0 | 15.0 |
| TAT. | Revenue | 3.3 | 10.3 | 10.8 | 5.0 | 10.8 | 10.5 | 4.8 | 12.8 | 8.0 | 10.0 | 12.8 | 8.3 | 11.0 | 9.5 | 6.3 | **2.3** |
| TAT. | Drawdown | 2.8 | 12.0 | 7.3 | 6.5 | 5.8 | 12.3 | 6.0 | 13.5 | 14.5 | 11.0 | 10.3 | 6.5 | 8.5 | 8.3 | **4.5** | 6.5 |
| TAT. | Sharpe | 3.3 | 10.3 | 12.3 | 5.3 | 12.3 | 9.0 | 4.3 | 13.3 | 6.8 | 12.0 | 13.3 | 9.0 | 10.0 | 9.3 | 3.3 | **2.8** |
| TAT. | Sortino | 3.0 | 10.0 | 12.3 | 5.5 | 11.5 | 9.3 | 3.8 | 13.3 | 7.3 | 11.8 | 13.8 | 9.0 | 10.0 | 9.0 | 4.0 | **2.8** |
| TAT. | Calmar | 3.5 | 9.8 | 11.8 | 5.3 | 12.5 | 9.0 | 5.0 | 13.0 | 7.8 | 10.0 | 13.5 | 8.8 | 9.5 | 9.3 | 5.0 | **2.5** |
| TEC. | Revenue | 6.8 | 5.0 | 14.0 | 6.8 | 14.8 | 9.5 | 7.5 | 5.0 | 2.8 | 11.8 | 11.8 | 4.3 | 13.5 | 9.3 | **1.8** | 11.8 |
| TEC. | Drawdown | 11.8 | 6.5 | 9.3 | 4.8 | 13.5 | 8.5 | 12.0 | 5.3 | 5.5 | 10.0 | 8.8 | 3.5 | 12.3 | 10.0 | **1.5** | 13.0 |
| TEC. | Sharpe | 7.5 | 4.8 | 13.8 | 7.0 | 15.5 | 9.3 | 7.3 | 4.3 | 3.3 | 13.3 | 10.3 | 3.5 | 14.0 | 9.0 | **1.8** | 11.8 |
| TEC. | Sortino | 7.5 | 5.0 | 14.3 | 7.0 | 15.5 | 9.0 | 7.3 | 4.5 | 3.0 | 13.5 | 10.8 | 3.3 | 13.5 | 9.0 | **1.8** | 11.3 |
| TEC. | Calmar | 6.0 | 6.0 | 13.5 | 7.3 | 15.0 | 8.5 | 8.3 | 5.0 | 3.3 | 12.3 | 10.0 | 4.3 | 13.8 | 9.8 | **1.5** | 11.8 |
| TIT. | Revenue | **1.3** | 6.3 | 14.0 | 9.3 | 11.5 | 11.5 | 2.3 | 7.0 | 2.5 | 13.3 | 11.8 | 6.3 | 15.8 | 12.0 | 6.0 | 5.5 |
| TIT. | Drawdown | **1.8** | 8.3 | 15.0 | 10.0 | 10.5 | 12.0 | 3.0 | 7.0 | 2.3 | 12.3 | 11.8 | 5.0 | 16.0 | 11.8 | 5.5 | 4.0 |
| TIT. | Sharpe | **1.3** | 6.3 | 13.3 | 9.0 | 12.5 | 11.0 | 2.3 | 7.3 | 2.5 | 15.0 | 12.3 | 6.0 | 14.0 | 12.3 | 6.0 | 5.3 |
| TIT. | Sortino | **1.3** | 7.0 | 13.0 | 8.8 | 13.5 | 10.8 | 2.3 | 7.5 | 2.5 | 14.8 | 11.3 | 6.0 | 14.8 | 11.5 | 6.0 | 5.3 |
| TIT. | Calmar | **1.3** | 6.3 | 14.0 | 9.0 | 12.0 | 10.8 | 2.5 | 7.0 | 2.5 | 13.5 | 11.8 | 6.5 | 15.8 | 11.8 | 6.0 | 5.5 |

Table 18: Benchmark results (MSE, MAE) across standard time-series datasets and prediction lengths. Each model occupies two columns (MSE, MAE).

| Dataset | Pred_len | Hipeen MSE | Hipeen MAE | NST MSE | NST MAE | DLinear MSE | DLinear MAE | RLinear MSE | RLinear MAE | Dish-TS MSE | Dish-TS MAE | SAN MSE | SAN MAE | Leddam MSE | Leddam MAE |
|---|---|---|---|---|---|---|---|---|---|---|---|---|---|---|---|
| electricity | 96 | 0.173 | 0.270 | 0.169 | 0.273 | 0.210 | 0.302 | 0.176 | 0.281 | 0.212 | 0.320 | 0.253 | 0.353 | 0.166 | 0.271 |
| | 192 | 0.181 | 0.279 | 0.182 | 0.286 | 0.210 | 0.305 | 0.191 | 0.293 | 0.235 | 0.344 | 0.259 | 0.353 | 0.176 | 0.284 |
| | 336 | 0.197 | 0.297 | 0.200 | 0.304 | 0.223 | 0.319 | 0.206 | 0.305 | 0.244 | 0.353 | 0.270 | 0.366 | 0.192 | 0.294 |
| | 720 | 0.235 | 0.328 | 0.222 | 0.321 | 0.258 | 0.350 | 0.238 | 0.330 | 0.256 | 0.359 | 0.297 | 0.385 | 0.232 | 0.328 |
| | Avg. | 0.197 | 0.294 | 0.193 | 0.296 | 0.225 | 0.319 | 0.203 | 0.302 | 0.237 | 0.344 | 0.270 | 0.364 | 0.191 | 0.294 |
| ETTh1 | 96 | 0.383 | 0.385 | 0.513 | 0.491 | 0.397 | 0.412 | 0.381 | 0.402 | 0.497 | 0.496 | 0.496 | 0.480 | 0.379 | 0.398 |
| | 192 | 0.435 | 0.416 | 0.534 | 0.504 | 0.446 | 0.441 | 0.428 | 0.429 | 0.592 | 0.555 | 0.555 | 0.507 | 0.434 | 0.432 |
| | 336 | 0.480 | 0.441 | 0.588 | 0.535 | 0.489 | 0.467 | 0.465 | 0.448 | 0.668 | 0.595 | 0.601 | 0.535 | 0.482 | 0.454 |
| | 720 | 0.509 | 0.484 | 0.643 | 0.616 | 0.513 | 0.510 | 0.494 | 0.478 | 0.696 | 0.634 | 0.666 | 0.584 | 0.498 | 0.478 |
| | Avg. | 0.452 | 0.431 | 0.570 | 0.537 | 0.461 | 0.458 | 0.442 | 0.439 | 0.613 | 0.570 | 0.579 | 0.527 | 0.448 | 0.441 |
| ETTh2 | 96 | 0.298 | 0.339 | 0.476 | 0.458 | 0.340 | 0.394 | 0.332 | 0.366 | 1.871 | 0.959 | 0.315 | 0.366 | 0.300 | 0.346 |
| | 192 | 0.387 | 0.394 | 0.512 | 0.493 | 0.482 | 0.479 | 0.404 | 0.411 | 3.715 | 1.369 | 0.395 | 0.411 | 0.388 | 0.400 |
| | 336 | 0.430 | 0.431 | 0.552 | 0.551 | 0.591 | 0.541 | 0.448 | 0.449 | 4.001 | 1.406 | 0.440 | 0.448 | 0.424 | 0.433 |
| | 720 | 0.624 | 0.547 | 0.562 | 0.560 | 0.839 | 0.661 | 0.454 | 0.462 | 3.118 | 1.259 | 0.430 | 0.453 | 0.428 | 0.446 |
| | Avg. | 0.435 | 0.428 | 0.526 | 0.516 | 0.563 | 0.519 | 0.410 | 0.422 | 3.176 | 1.248 | 0.395 | 0.420 | 0.385 | 0.406 |
| ETTm1 | 96 | 0.317 | 0.349 | 0.386 | 0.398 | 0.346 | 0.374 | 0.357 | 0.371 | 0.406 | 0.438 | 0.346 | 0.371 | 0.328 | 0.359 |
| | 192 | 0.364 | 0.376 | 0.459 | 0.444 | 0.382 | 0.391 | 0.363 | 0.384 | 0.455 | 0.466 | 0.382 | 0.391 | 0.364 | 0.381 |
| | 336 | 0.397 | 0.400 | 0.495 | 0.464 | 0.415 | 0.415 | 0.393 | 0.404 | 0.505 | 0.501 | 0.412 | 0.410 | 0.396 | 0.403 |
| | 720 | 0.473 | 0.446 | 0.585 | 0.516 | 0.473 | 0.451 | 0.459 | 0.440 | 0.633 | 0.578 | 0.474 | 0.445 | 0.471 | 0.447 |
| | Avg. | 0.388 | 0.393 | 0.481 | 0.456 | 0.404 | 0.408 | 0.393 | 0.400 | 0.500 | 0.496 | 0.404 | 0.404 | 0.390 | 0.397 |
| ETTm2 | 96 | 0.176 | 0.257 | 0.192 | 0.274 | 0.193 | 0.293 | 0.175 | 0.259 | 0.679 | 0.551 | 0.182 | 0.273 | 0.177 | 0.258 |
| | 192 | 0.247 | 0.308 | 0.280 | 0.339 | 0.284 | 0.361 | 0.247 | 0.315 | 0.830 | 0.616 | 0.248 | 0.319 | 0.249 | 0.307 |
| | 336 | 0.313 | 0.338 | 0.334 | 0.361 | 0.382 | 0.429 | 0.302 | 0.349 | 1.372 | 0.826 | 0.304 | 0.353 | 0.313 | 0.346 |
| | 720 | 0.467 | 0.423 | 0.417 | 0.413 | 0.558 | 0.525 | 0.408 | 0.407 | 2.573 | 1.125 | 0.402 | 0.416 | 0.418 | 0.405 |
| | Avg. | 0.301 | 0.332 | 0.306 | 0.347 | 0.354 | 0.402 | 0.283 | 0.333 | 1.364 | 0.779 | 0.284 | 0.340 | 0.289 | 0.329 |
| exchange_rate | 96 | 0.087 | 0.206 | 0.111 | 0.237 | 0.088 | 0.218 | 0.098 | 0.218 | 0.116 | 0.258 | 0.087 | 0.216 | 0.087 | 0.207 |
| | 192 | 0.197 | 0.314 | 0.219 | 0.335 | 0.176 | 0.315 | 0.195 | 0.314 | 0.242 | 0.385 | 0.171 | 0.317 | 0.179 | 0.301 |
| | 336 | 0.344 | 0.422 | 0.346 | 0.476 | 0.313 | 0.427 | 0.359 | 0.434 | 0.380 | 0.487 | 0.344 | 0.408 | 0.361 | 0.433 |
| | 720 | 0.711 | 0.645 | 1.092 | 0.769 | 0.839 | 0.695 | 0.997 | 0.756 | 1.305 | 0.899 | 0.718 | 0.650 | 0.967 | 0.738 |
| | Avg. | 0.335 | 0.397 | 0.461 | 0.454 | 0.354 | 0.414 | 0.412 | 0.431 | 0.511 | 0.507 | 0.330 | 0.398 | 0.398 | 0.420 |
| solar_AL | 96 | 0.179 | 0.247 | 0.321 | 0.380 | 0.290 | 0.378 | 0.222 | 0.275 | 0.186 | 0.278 | 0.274 | 0.318 | 0.222 | 0.270 |
| | 192 | 0.205 | 0.260 | 0.346 | 0.369 | 0.320 | 0.398 | 0.252 | 0.298 | 0.218 | 0.286 | 0.310 | 0.340 | 0.263 | 0.279 |
| | 336 | 0.220 | 0.265 | 0.357 | 0.387 | 0.353 | 0.415 | 0.277 | 0.317 | 0.218 | 0.292 | 0.334 | 0.350 | 0.271 | 0.289 |
| | 720 | 0.218 | 0.258 | 0.375 | 0.424 | 0.357 | 0.413 | 0.288 | 0.326 | 0.212 | 0.288 | 0.333 | 0.344 | 0.261 | 0.285 |
| | Avg. | 0.205 | 0.257 | 0.350 | 0.390 | 0.330 | 0.401 | 0.260 | 0.304 | 0.208 | 0.286 | 0.313 | 0.338 | 0.254 | 0.281 |
| traffic | 96 | 0.607 | 0.309 | 0.612 | 0.338 | 0.650 | 0.396 | 0.580 | 0.384 | 0.611 | 0.418 | 0.582 | 0.368 | 0.549 | 0.365 |
| | 192 | 0.614 | 0.310 | 0.613 | 0.340 | 0.598 | 0.370 | 0.587 | 0.377 | 0.595 | 0.405 | 0.586 | 0.371 | 0.550 | 0.369 |
| | 336 | 0.627 | 0.314 | 0.618 | 0.328 | 0.605 | 0.373 | 0.601 | 0.384 | 0.619 | 0.420 | 0.608 | 0.374 | 0.574 | 0.376 |
| | 720 | 0.671 | 0.336 | 0.653 | 0.355 | 0.645 | 0.394 | 0.638 | 0.399 | 0.653 | 0.425 | 0.638 | 0.391 | 0.609 | 0.390 |
| | Avg. | 0.630 | 0.317 | 0.624 | 0.340 | 0.625 | 0.383 | 0.601 | 0.386 | 0.619 | 0.417 | 0.604 | 0.376 | 0.571 | 0.375 |
| weather | 96 | 0.149 | 0.198 | 0.173 | 0.223 | 0.195 | 0.252 | 0.158 | 0.204 | 0.164 | 0.239 | 0.181 | 0.239 | 0.154 | 0.202 |
| | 192 | 0.191 | 0.238 | 0.245 | 0.285 | 0.237 | 0.295 | 0.206 | 0.249 | 0.208 | 0.283 | 0.220 | 0.275 | 0.203 | 0.247 |
| | 336 | 0.243 | 0.280 | 0.321 | 0.338 | 0.282 | 0.331 | 0.266 | 0.292 | 0.260 | 0.323 | 0.268 | 0.312 | 0.261 | 0.289 |
| | 720 | 0.313 | 0.330 | 0.414 | 0.410 | 0.345 | 0.382 | 0.348 | 0.325 | 0.326 | 0.369 | 0.335 | 0.359 | 0.343 | 0.390 |
| | Avg. | 0.224 | 0.261 | 0.288 | 0.314 | 0.265 | 0.315 | 0.244 | 0.268 | 0.239 | 0.303 | 0.251 | 0.296 | 0.240 | 0.270 |

Table 19: Ablation study on the bias term conducted on the ETTh1, Exchange, and Weather datasets. We compare Hipeen without a bias term, with a bias applied along the N-dimension ($N \times 1$), and along the H-dimension ($1 \times H$).

| Horizon | | 96 | | 192 | | 336 | | 720 | |
|---|---|---|---|---|---|---|---|---|---|
| Metric | | MSE | MAE | MSE | MAE | MSE | MAE | MSE | MAE |
| ETTh1 | Ours | **0.383** | **0.385** | **0.435** | **0.416** | **0.480** | **0.441** | **0.509** | **0.486** |
| | N-dim | 0.387 | 0.399 | 0.444 | 0.434 | 0.493 | 0.464 | 0.530 | 0.510 |
| | H-dim | 0.407 | 0.415 | 0.506 | 0.485 | 0.671 | 0.592 | 0.892 | 0.702 |
| | No bias | 0.808 | 0.698 | 1.043 | 0.816 | 1.062 | 0.820 | 1.247 | 0.877 |
| Exchange | Ours | 0.087 | 0.206 | 0.196 | 0.314 | **0.332** | **0.417** | 0.705 | 0.643 |
| | N-dim | **0.081** | **0.201** | **0.166** | **0.294** | 0.356 | 0.441 | **0.635** | **0.615** |
| | H-dim | 0.090 | 0.209 | 0.307 | 0.378 | 0.542 | 0.520 | 1.259 | 0.849 |
| | No bias | 0.185 | 0.279 | 1.319 | 0.732 | 1.765 | 0.902 | 11.270 | 2.543 |
| Weather | Ours | **0.149** | **0.198** | **0.191** | **0.238** | **0.243** | **0.280** | 0.313 | **0.330** |
| | N-dim | 0.153 | 0.203 | 0.195 | 0.249 | 0.250 | 0.295 | 0.320 | 0.340 |
| | H-dim | 0.150 | 0.202 | 0.196 | 0.250 | 0.244 | 0.291 | **0.312** | 0.336 |
| | No bias | 0.153 | 0.213 | 0.203 | 0.265 | 0.260 | 0.319 | 0.332 | 0.373 |

Table 20: Training and inference efficiency comparison across models. Reported are average time per step (ms) and maximum VRAM usage (MB), with corresponding ranks. *With an extra ensemble dimension of 1, the method scales only the batch size without adding learnable parameters, yielding high efficiency.

| Model | Train Avg. Time (ms) | | Infer Avg. Time (ms) | | Train Max VRAM (MB) | | Infer Max VRAM (MB) | |
|---|---|---|---|---|---|---|---|---|
| | Value | Rank | Value | Rank | Value | Rank | Value | Rank |
| CycleNet | 2.7 | 4 | 0.7 | 4 | 21.7 | 4 | 20.3 | 4 |
| DLinear | 2.0 | 1 | 0.7 | 3 | 19.0 | 1 | 18.4 | 2 |
| FEDformer | 261.1 | 17 | 62.9 | 15 | 2071.4 | 15 | 469.9 | 14 |
| FRNet | 14.1 | 8 | 3.5 | 8 | 53.9 | 7 | 34.6 | 8 |
| iTransformer | 10.6 | 6 | 2.5 | 5 | 26.5 | 5 | 21.2 | 6 |
| NST | 63.5 | 12 | 24.0 | 13 | 2093.2 | 16 | 828.0 | 17 |
| PatchTST | 16.5 | 9 | 4.6 | 9 | 427.5 | 11 | 213.5 | 12 |
| PerimidFormer | 88.3 | 14 | 65.1 | 16 | 969.7 | 13 | 433.2 | 13 |
| TiDE | 27.4 | 10 | 8.1 | 11 | 193.2 | 10 | 60.0 | 10 |
| TimeMixer | 31.6 | 11 | 7.0 | 10 | 80.5 | 8 | 33.0 | 7 |
| TimesNet | 134.7 | 16 | 18.7 | 12 | 582.9 | 12 | 62.2 | 11 |
| RLinear | 2.5 | 3 | 0.6 | 2 | 19.2 | 3 | 18.1 | 1 |
| DishTS | 110.2 | 15 | 94.8 | 17 | 3418.1 | 17 | 709.1 | 16 |
| SAN | 2.1 | 2 | 0.6 | 1 | 19.0 | 1 | 18.4 | 2 |
| Leddam | 11.8 | 7 | 3.0 | 6 | 91.2 | 9 | 57.8 | 9 |
| Hipeen* | 5.1 | 5 | 3.3 | 7 | 52.6 | 6 | 20.5 | 5 |

