# OpenReview forum: "Hierarchical Periodic Stationarization for Non-stationary Time Series Forecasting"
_ICLR.cc/2026/Conference — Submitted to ICLR 2026_

### Official Review · Reviewer_JNzr · 2025-10-29

**Soundness:** 2
**Presentation:** 2
**Contribution:** 2
**Rating:** 4
**Confidence:** 5

**Summary:**

The paper indicates that many state-of-the-art forecasting models still fail to predict even a simple long-period sine wave, largely because existing datasets underrepresent the non-stationary characteristics commonly found in real-world time series—leading to misleading forecasting. To address this, the authors introduce controlled datasets to reveal the potential information loss caused by the widely used z-score normalization method. Furthermore, they propose Hipeen, a hierarchical periodic stationarization technique that decomposes time series into multiple periodic components, thereby reducing information loss during stationarization. The proposed approach is thoroughly validated on synthetic, stock, and long-horizon forecasting datasets, demonstrating superior performance in MSE and MAE metrics.

**Strengths:**

1. The paper conducts extensive experiments across diverse datasets and scenarios, and further constructs tailored datasets to validate the motivation and observed phenomena.
2. The proposed approach offers insightful perspectives on handling non-stationarity, differing from traditional methods by emphasizing the importance of preserving critical information during the stationarization process.
3. The empirical validation is clear and comprehensive, with rich experimental evidence supporting the claims

**Weaknesses:**

1. Figure 1 appears overly crowded, which affects readability, and both axis labels are too small.
2. The current baselines are not well-suited for the stock forecasting task; I suggest including additional baselines specifically designed for financial or stock data.
3. Using only MSE and MAE for stock prediction lacks sufficient persuasiveness—metrics such as drawdown or risk-adjusted returns would better reflect real-world forecasting quality.
4. On the long-horizon forecasting benchmarks (Table 4), the improvements are not significant; notably, Leddam* performs even better, especially considering the low predictability of the Exchange dataset.
5. The paper could include more discussion of post-SAN works such as FAN and DDN, which also incorporate normalization and frequency-domain operations.

**Questions:**

see weeknesses.

---

> ### Author Response · Authors · 2025-11-21
> **Official Comment by Authors (1/2)**
>
> We thank the reviewer for recognizing our contributions in both experimental validation and methodological innovation, as our approach provides a novel perspective on handling non-stationarity by preserving critical information, distinguishing it from traditional methods.
>
> ---
> ### [W1]
>
> We have thoroughly revised Figure 1 to make it easier to interpret. We adjusted the sizes of the axis labels and legends. Please refer to the revised manuscript PDF.
>
> Additionally, we revised the Related Works section and the “Motivation behind Hipeen” in the Method section (lines 170–195) to provide a more in-depth discussion of prior work and to better highlight the distinctions of Hipeen. We also improved the overall readability throughout the manuscript. The major revisions are marked in blue.
>
> ---
> ### [W2,3]
>
> Thank you for providing such valuable ideas.
> Following your suggestions, we added **two stock-forecasting–specialized models** and simulated stock trading to include **Revenue, Drawdown, Sharpe, Sortino, and Calmar scores** (risk-adjusted return metrics). We conducted experiments with prediction horizons of 12, 24, 48, and 96, and evaluated performance using MAE, MAPE, and RMSE. These results are presented in the main Table 3, and details are in Appendix E.2.
>
> Our original baselines consisted of models commonly used for multivariate long/short-term forecasting [1,2,3], and existing TSF models are generally not validated on stock data. Following your suggestion, we added **MambaStock (SMamba)** [4] and **StockTransformer (STF)** [5] as additional baselines (we excluded PMANet [6] because the GitHub code was not usable).
>
> A summarized version of Table 3 is provided below. Here, “R.” denotes the average rank, averaging the model’s rank across 144 scenarios (12 stocks × 4 horizons × 3 seeds).
>
> >|Models|Hipeen|DLin.|RLin.|SAN|Leddam|DDN|FAN|TMixer|PTST|TNet|CNet|PeriMF|FRNet|SMamba|STF|
> >|-|-|-|-|-|-|-|-|-|-|-|-|-|-|-|-|
> >|MAE|**.198**|.294|.205|.212|.210|.248|.252|.213|.209|.242|.212|.206|.206|.912|.818|
> >|MAPE|**.402**|.538|.456|.438|.437|.527|.467|.418|.418|.488|.445|.447|.416|.882|1.01|
> >|RMSE|**.274**|.386|.282|.288|.287|.330|.336|.291|.288|.324|.288|.283|.282|1.02|.928|
> >|-|
> >|Revenue R.|**5.38**|9.60|9.81|8.42|9.02|9.73|8.63|8.21|5.92|10.52|7.69|10.21|8.50|7.48|6.19|
> >|Drawdown R.|7.19|11.06|7.71|9.13|**6.21**|9.46|10.23|9.08|8.00|8.54|7.17|9.60|7.40|8.42|8.44|
> >|Sharpe R.|**5.63**|9.29|9.83|8.38|8.92|9.77|8.27|8.25|6.10|10.67|7.71|10.04|8.52|6.69|6.29|
> >|Sortino R.|**5.65**|9.54|9.75|8.35|9.02|9.83|8.29|8.21|6.10|10.75|7.67|9.98|8.35|6.77|6.19|
> >|Calmar R.|**5.58**|9.60|9.81|8.46|8.98|9.44|9.02|7.83|6.00|10.67|7.81|9.94|8.42|7.33|6.29|
>
> Hipeen achieves the best performance for MAE, MAPE, and RMSE, ranking first on MAE in more than 66% of the 48 combinations. It also leads in Revenue, Sharpe, Sortino, and Calmar, demonstrating high returns with strong risk-adjusted performance. Since Drawdown measures peak-to-trough decline, lower-return models can appear better [7] (no trades would obtain the best drawdown). Considering that Hipeen achieved the highest returns, its third-place performance in Drawdown is solid.
>
> For more details, please refer to the PDF.
>
> ---

---

> ### Author Response · Authors · 2025-11-21
> **Official Comment by Authors (2/2)**
>
> ### [W4]
>
> We apologize for not providing sufficient explanation regarding the long-horizon forecasting benchmark experiments. We have revised the manuscript to clarify this point.
>
> 1. In Table 4, we now explicitly indicate the **inherent parameters introduced by each stationarization method** in addition to the backbone. A summary of Table 4 is provided below. (DDN and FAN have also been added [8,9].)
>
> >|Model|Hipeen|NST|Dlinear|Rlinear*|DishTS*+|SAN*+|Leddam*+|DDN*+|FAN*+|
> >|-|-|-|-|-|-|-|-|-|-|
> >|Avg. value|**0.349**|0.414|0.400|0.363|0.690|0.383|0.354|0.391|0.381|
> >|Avg. Rank|**2.39**|6.56|6.39|3.67|7.39|4.89|2.50|6.06|4.94|
> >|Inh. Param.|0|0|0|0|15081k|114k|3415k|5539k|59k|
>
> 2. We have added a detailed analysis. The last row indicates the inherent learnable parameters of the stationarization module, excluding the backbone. **Hipeen has no internal parameters and outperforms other learning-free approaches such as NST, DLinear, and RLinear (RevIN)**. Although the same linear backbone was applied to eliminate architectural influence, some methods introduce multiple layers and non-linear activations within their stationarization modules, gaining architectural advantages that hinder fair comparison. Notably, Leddam contains 180 times more parameters than DLinear, raising practicality concerns. Even so, Hipeen achieves strong performance, demonstrating its effectiveness on relatively stationary signals (lines 447–454).
>
> ---
> ### [W5]
>
> Thank you for introducing the latest post-SAN stationarization methods. They are highly relevant to our work and fit well within our manuscript. We have incorporated them as follows:
>
> 1. We added descriptions of **DDN[8] and FAN[9] in the Related Works section**, and highlighted their differences from Hipeen (lines 471–161).
>
> 2. We included DDN and FAN in the controlled dataset experiments (Table 1), stock dataset experiments (Table 3), and benchmark experiments (Table 4).
>
>  a. A summary of Table 1 (showing only average ranks) is as follows:
>
> |Model|Metric|Exponential|Threshold|Sine wave|
> |-|-|-|-|-|
> |DDN|MSE Rank|13.7|11.7|14|
> ||MAE Rank|13.7|10.7|13.8|
> |FAN|MSE Rank|13.0|2.3|9.8|
> ||MAE Rank|12.7|2.3|10.0|
> |Hipeen|MSE Rank|**1.0**|**1.0**|**1.0**|
> ||MAE Rank|**1.0**|**1.0**|**1.0**|
>
> We observe that Hipeen consistently outperforms both DDN and FAN across all cases. **These two models also suffer from critical information loss during their stationarization processes**.
>
> b. Summaries of the results for Tables 3 and 4 were provided in our previous responses (W2,3 and 4).
> Hipeen outperforms both DDN and FAN on the **stock forecasting benchmarks (every metrics) as well as the long-horizon forecasting benchmarks**.
>
> ---
> Once again, thank you for your valuable contributions. Thanks to your input, our manuscript has been significantly strengthened.
>
> Please feel free to share any additional questions or comments.
>
> ---
> >[1] Timemixer: Decomposable multiscale mixing for time series forecasting. ICLR (2024)
> >[2] Peri-midformer: Periodic pyramid transformer for time series analysis.. NeurIPS (2024)
> >[3] TimeMixer++: A General Time Series Pattern Machine for Universal Predictive Analysis. ICLR. 2025.
> >[4] MambaStock: Selective state space model for stock prediction. arXiv (2024); 31 citations
> >[5] Predictive modeling of stock prices using transformer model ICMLT(2024)
> >[6] PMANet: a time series forecasting model for Chinese stock price prediction. Scientific Reports (2024)
> >[7] On inefficiency of markowitz-style investment strategies when drawdown is important. IEEE (2017)
> >[8] DDN: Dual-domain dynamic normalization for non-stationary time series forecasting. NeurIPS (2024)
> >[9] Frequency adaptive normalization for non-stationary time series forecasting. NeurIPS (2024)

---

> ### Author Response · Authors · 2025-11-26
> **Kind Request for Assessment Following Revisions**
>
> ---
> ---
> Dear reviewer,
>
> Thank you again for your insightful comments and constructive suggestions.
> Following your feedback, we have revised the manuscript substantially:
>
> - Figure 1 readability, axis labels, and visual clarity have been improved.
> - Additional stock-specific baselines (MambaStock, StockTransformer), as well as simulated trading evaluations (Revenue, Drawdown, Sharpe, Sortino, Calmar), have been added.
> - We included post-SAN methods (DDN and FAN) across all experiments and expanded related work accordingly.
> - Benchmark analysis has been clarified, including parameter fairness, module complexity discussion, and additional experimental insights.
>
> Since the discussion period ends in one week, we kindly ask if you could update your assessment after reviewing the revised manuscript.
>
> If you have any further questions or would like clarification on any part of the revision, please feel free to let us know — we would be happy to respond.
>
> Warm regards,
> The authors
>
> ---
> ---

---

### Official Review · Reviewer_fNqm · 2025-10-30

**Soundness:** 3
**Presentation:** 2
**Contribution:** 3
**Rating:** 6
**Confidence:** 4

**Summary:**

This paper addresses the challenge of stationarization for non-stationary time series forecasting. Instead of relying on conventional z-normalization-based approaches, the authors propose a novel framework named Hippen, which represents each time series value as a combination of multiple periodic components. The authors conduct extensive experiments on both controlled synthetic datasets and real-world benchmarks across diverse data scenarios. The results demonstrate that Hippen achieves superior performance compared to existing stationarization methods.

**Strengths:**

- The paper introduces a novel and conceptually interesting approach to stationarization for non-stationary time series forecasting. By decomposing each value into multiple periodic components rather than applying standard z-normalization, the proposed Hippen framework aims to retain essential information typically lost in conventional preprocessing.
- Extensive experiments are conducted on both synthetic and real-world datasets, and the proposed method shows consistent and significant improvements across different data conditions and forecasting models.

**Weaknesses:**

- Some claims need further clarification:
  - One of the central claims of this paper is that gradients and absolute values are essential for non-stationary time series forecasting but are discarded by conventional stationarization methods. This argument requires further justification. For example, methods like RevIN restore both the mean and standard deviation after normalization, which effectively recover much of the original information. The paper should clarify in what specific sense such information is “discarded” and why Hippen preserves it better.

  - The mechanism by which the proposed transformation retains gradients and absolute values, and how this process alleviates the impact of non-stationarity, needs a clearer intuitive explanation. Currently, the experimental evidence is convincing, but the theoretical or conceptual reasoning behind the improvement is underdeveloped.

  - What does the hierarchical periodicity metioned in line 255 refer to? And why the cosine distance loss function help to account for it?

- It is recommended to include more investigation experiments:
  - In Appendix C.2.2, the ensemble mechanism with $E=16$ is mentioned. It would be informative to analyze the effect of different ensemble sizes $E$ on performance, to assess the robustness and scalability of the approach.

  - It would also be valuable to investigate whether Hippen can serve as a model-agnostic plugin, similar to RevIN, that can enhance the performance of various backbone forecasting models.

**Questions:**

Please refer to the weaknesses part.

---

> ### Author Response · Authors · 2025-11-21
> **Official Comment by Authors (1/2)**
>
> We thank the reviewer for recognizing Hipeen's conceptual novelty in decomposing values into multiple periodic components to preserve essential information, as well as its consistent improvements across synthetic and real-world datasets. We sincerely appreciate the reviewer taking the time to read our paper carefully and for providing such thoughtful feedback.
>
> ---
> ## [Some claims need further clarification 1,2]
>
> We thank the reviewer for raising this important point and for the opportunity to clarify our argument.
>
> ### 1. What it means for normalization-based stationarization to “discard information”
>
> “Information loss” refers to the fact that many normalization-based
> stationarization procedures are **many-to-one mappings**. That is, once the input
> signal is normalized, it cannot be uniquely recovered without additional information
> that is not provided to the backbone model.
>
> Formally, for an input sequence $X$, a stationarization operator $S(\cdot)$ is
> information-preserving only if it is injective:
>
> $ S(X_1) = S(X_2) \Rightarrow X_1 = X_2. $
>
> This means that if two different inputs $X_1$ and $X_2$ are mapped to the same
> normalized output, some information about the original inputs is lost.
>
> For example, consider **RevIN**:
>
> $\tilde{X} = S_{\mathrm{RevIN}}(X)
> = \frac{X - \mu(X)}{\sigma(X)},
> $
>
> To recover the original \(X\), one would need:
>
> $X = \tilde{X}\cdot \sigma(X) + \mu(X)$
>
> Here, $\tilde{X}$ alone is insufficient; It requires additional statistics
> $\mu(X)$ and $\sigma(X)$ that are stored inside RevIN but not provided to the
> backbone model.
>
> Thus, the backbone receives a representation that is information-reduced. In this
> sense, normalization-based stationarization discards (i.e., hides) raw value,
> scale, and local gradient information.
>
> ### 2. Why this matters for non-stationary forecasting
>
> Let the input be decomposed as $X = (A, B)$, where:
>
> - $A$ encodes information such as absolute value, scale, and local gradients
> (all of which normalization removes or rescales unpredictably),
> - $B$ contains the shape information that remains after normalization.
>
> If the target depends primarily on $A$ (i.e., $Y = f(A)$), then methods such as
> RevIN degrade performance because the backbone model receives only $B$.
>
> Conversely, if $Y$ depends primarily on $B$, RevIN performs well.
>
> Our *Controlled dataset* explicitly constructs cases where $Y$ is a function of $A$,
> and conventional normalization-based models indeed perform poorly in these settings.
>
> ### 3. How Hipeen preserves information
>
> In contrast, the proposed Hipeen projection is one-to-one.
> We show that the Hipeen projection $H(\cdot)$ admits an explicit inverse via the
> Hipeen estimator $H^{-1}(\cdot)$, satisfying:
>
> $H^{-1}(H(X)) = X.$
>
> Thus, no information is lost at any stage of the transformation. As a result, the
> backbone model inside Hipeen receives the full information of $X$, including
> absolute values and gradients.
>
> This explains why Hipeen avoids the failure modes of normalization-based approaches
> under non-stationarity.
>
> ### 4. Why Hipeen achieves distribution alignment (stationarization) without information loss
>
> Achieving distributional alignment while maintaining injectivity is inherently
> challenging. Most prior methods rely on removing non-stationary statistics (mean,
> variance, etc.) to match distributions—an intuitive but lossy approach.
>
> Feeding raw non-stationary values to a model typically induces distributional shift
> and hurts performance. This pattern is consistent with models such as the
> Nonstationary Transformer, which shows degraded performance even when mean/variance
> statistics are passed through auxiliary heads.
>
> Hipeen achieves alignment in a fundamentally different manner: it leverages
> hierarchical periodicity to project heterogeneous non-stationary inputs into a
> shared representation space without discarding any information.
>
> As shown in Figure 2 (D1, D2), the Hipeen projection produces well-aligned
> distributions while remaining invertible.
>
> The conceptual explanation behind this mechanism is provided in lines 181–192 of the
> revised manuscript.
>
> ### 5. Revision to the manuscript
>
> Due to space limitations, we will include the following content in the Method section in the final version:
>
> 1. The sense in which normalization-based stationarization is many-to-one and discards information.
> 2. How Hipeen preserves information through its one-to-one mapping.
> 3. How Hipeen achieves stationarization.
> ---

---

> ### Author Response · Authors · 2025-11-21
> **Official Comment by Authors (2/2)**
>
> ## [Some claims need further clarification 3]
>
> By "hierarchical periodicity," we refer to the structure in which the $2H$-dimensional representation is composed of $H$ pairs of $(\sin, \cos)$ components, each corresponding to a sub-period.
> Rather than optimizing all $2H$ dimensions jointly, we optimize each pair independently using the cosine distance (angular loss). This ensures that the phase of each sub-period is correctly aligned, effectively capturing the hierarchical periodic structure.
>
> The cosine distance is used as a loss to align the angular phase of each hierarchical component. As mentioned in lines 264--265, this loss can be approximated by the squared angular difference.
>
> We have revised the main text (lines 255--263) to clarify the above explanation.
>
> ---
> ## [Investigation experiments 1]
> We appreciate the important suggestion.
> **In Appendix C.2.2 (lines 1026–1073)**, we reported experimental results for Exponentials, Threshold, and Sine wave by increasing the dimension of the model’s **extra ensemble from 1 to 32** and presenting the results as graphs.
>
> A summary of the results (MSE only) is as follows:
>
> >|Extra Ensemble|1|2|4|8|16|32|2nd Best|
> >|-|-|-|-|-|-|-|-|
> >|Exponential|0.255|0.238|0.219|0.209|**0.188**|0.249|0.372|
> >|Threshold|0.542|0.507|**0.500**|0.513|0.559|0.653|0.736|
> >|Sine wave|.0071|.0033|.0022|**.0018**|.0024|.0035|.0118|
>
> Across all datasets, Hipeen consistently outperforms the second-best model. Its performance curve shows a U-shape, decreasing at first and then improving as the ensemble size increases. Selecting 4–8 ensembles offers a good balance. Note that these extra ensembles add no learnable parameters and are efficiently computed along the batch dimension.
>
> ---
> ## [Investigation experiments 2]
> Since Hipeen projection converts a scalar into a vector of size H, a simple plug-in requires additional modifications to the model.
>
> Following the reviewer’s suggestion, we conducted a pilot study to explore whether Hipeen can further improve the performance of the original model when the H dimension is simply modified as a channel.
>
> We conducted experiments on 4 ETT datasets, integrating iTransformer as a backbone for Hipeen.
> The following table presents the results on the ETTh1 subset.
>
> >|Horizon|<Hipeen|+iTF>|<Hip|een>|<iTrans|former>
> >|-|-|-|-|-|-|-|
> >|ETTh1|MSE|MAE|MSE|MAE|MSE|MAE|
> >|96|**0.381**|**0.384**|0.383|0.385|0.386|0.405|
> >|192|**0.433**|0.421|0.435|**0.416**|0.441|0.512|
> >|336|**0.480**|**0.441**|0.480|0.441|0.487|0.458|
> >|720|**0.482**|**0.460**|0.509|0.484|0.503|0.491|
> >|Avg.|**0.444**|**0.426**|0.452|0.431|0.454|0.467|
>
>
> Hipeen+iTransformer outperformed both Hipeen and iTransformer in 56% of the cases, and outperformed at least one of them in 94% of the cases. Notably, a significant performance improvement was observed for the 720-length horizon. Furthermore, when Hipeen and iTransformer showed comparable performance, their combination tended to outperform both models.
>
> Theoretically, with a large H value, Hipeen passes values close to the raw input to the internal backbone, making it possible for Hipeen to improve performance as a plug-in.
>
> Exploring modifications that are well-suited for Hipeen projection will be an important topic for future research.
>
> ---
> We sincerely thank the reviewer for carefully reading our paper and providing valuable feedback.
> Should there be any remaining questions or further comments, we would greatly appreciate your sharing them.

---

> > ### Comment · Reviewer_fNqm · 2025-11-25
> >
> > Thank you for your detailed rebuttal, which helps address my concerns, especially regarding the claims requiring further clarification.
> >
> > In addition, I noticed that the setting of $E$ appears to have a substantial impact on performance in the controlled datasets. Could you also provide experimental results on real-world datasets to further validate this effect?

---

> ### Author Response · Authors · 2025-11-26
>
> Thank you for your thoughtful follow-up comment — we are glad that our clarification helped address your concerns.
>
> Following your suggestion, we conducted additional experiments on real-world datasets by varying the extra ensemble dimension $\(E\)$, and we now report the results below.
>
> ---
>
> ### **# Nifty50 Dataset (12 stocks × {12, 24, 48, 96} horizons × 3 seeds)**
> (Results aligned with Table 3 in the manuscript)
>
> | E | 1 | 2 | 4 | 8 | 16 | 32 | 2nd Best |
> |---|---|---|---|---|---|----|----------|
> | **MAE** | 0.204 | 0.202 | 0.202 | 0.199 | **0.198** | 0.200 | 0.205 |
> | **MAPE** | 0.407 | 0.402 | 0.399 | **0.393** | 0.401 | 0.399 | 0.416 |
> | **RMSE** | 0.278 | 0.278 | 0.275 | **0.273** | 0.274 | 0.274 | 0.282 |
>
> ---
>
> ### **# Weather Benchmark Dataset**
>
> | E | 1 | 2 | 4 | 8 | 16 | 32 | 2nd Best |
> |---|---|---|---|---|---|----|----------|
> | **MSE** | 0.227 | 0.223 | **0.220** | 0.221 | 0.224 | 0.229 | 0.239 |
> | **MAE** | 0.268 | 0.264 | 0.262 | **0.261** | **0.261** | 0.262 | 0.268 |
>
> ---
>
> ### **Summary**
>
> These findings demonstrate that varying $\(E\)$ produces a measurable difference in real-world datasets, resulting in approximately **1–3% performance variation** on average across evaluation metrics.
>
> Even with \(E=1\), the model consistently outperforms the second-best baseline. However, setting $\(E\)$ within the range of **4–8** can provide further performance gains.
>
> > *Unlike other experiments where the learning rate was fixed, the benchmark experiments followed prior work and performed hyperparameter search on the learning rate. However, for these additional experiments, we reused the learning rate selected at \(E=16\) to reduce computation time. Additional tuning may yield further improvements.*
>
> ---
>
> We appreciate your helpful suggestion and welcome any further questions or comments you may have.

---

> > ### Comment · Reviewer_fNqm · 2025-11-27
> >
> > Thank you for your additional experiments. I will keep my positive evaluation.

---

### Official Review · Reviewer_NASy · 2025-10-30

**Soundness:** 2
**Presentation:** 2
**Contribution:** 2
**Rating:** 4
**Confidence:** 4

**Summary:**

This paper points out that current time series forecasting models struggle to predict even simple periodic functions such as sine waves. The authors argue that this arises from a gap between real-world non-stationarity and the stationary assumptions often used in training, and that standard stationarization techniques inevitably lead to information loss. To address this, the paper introduces new controlled datasets and proposes a hierarchical periodic stationarization method, Hipeen. Experiments show that applying this method to existing backbone and MLP models yields notable performance gains.

**Strengths:**

1. Clear and reasonable motivation, easy to understand.
2. Extensive experiments, including new datasets.

**Weaknesses:**

1. The writing and structure could better highlight the differences from prior work and the paper’s unique contributions. Related work discussion should also be more comprehensive.
2. Applying long-horizon forecasting techniques to short-term financial stock prediction is questionable, as most long-horizon TSF methods are not well-suited for such tasks.
3. The use of MSE/MAE metrics shows only moderate advantages on long-term tasks, making it difficult to fully demonstrate the superiority of the proposed method.

**Questions:**

See weaknesses.

**Details Of Ethics Concerns:**

nan

---

> ### Author Response · Authors · 2025-11-21
>
> We thank the reviewer for recognizing the clear and well-motivated nature of our work, as well as the comprehensiveness of our experiments, including evaluations on newly constructed datasets.
>
> ---
> ### [W1]
> We have substantially revised the Related Works (Section 2) and the Motivation behind Hipeen (lines 170–195) to improve the overall readability. The Related Works section is now more comprehensive, including the **latest stationarization methods DDN [1] and FAN [2]**, and we emphasize Hipeen’s intuition and its distinctions from prior work. Please refer to the revised manuscript PDF.
>
> Additionally, Appendix A (Additional Related Works) now includes content on **stock forecasting models**. Figure 1 has been fully revised for clarity. Major changes are highlighted in blue.
>
> ---
> ### [W2]
> We agree with your comment. We have substantially strengthened the Nifty50 experiments as follows:
>
> 1. Added two stock-forecasting-specific models, **MambaStock (SMamba) [3] and StockTransformer (STF) [4]**, as baselines.
> 2. Expanded the prediction horizon from 96 to {12, 24, 48, 96}.
> 3. Changed the primary evaluation metrics to **MAE, MAPE, and RMSE**, commonly used in multivariate short-term forecasting.
> 4. Added advanced stock-specific metrics: trading-based evaluation, reporting **Revenue, Drawdown, Sharpe, Sortino, and Calmar scores** to better assess real-world applicability.
>
> The results are shown in **Table 3**. A summarized version of Table 3 is provided below, where “R.” denotes the average rank across 144 scenarios (12 stocks × 4 horizons × 3 seeds).
>
> >|Models|Hipeen|DLin.|RLin.|SAN|Leddam|DDN|FAN|TMixer|PTST|TNet|CNet|PeriMF|FRNet|SMamba|STF|
> >|-|-|-|-|-|-|-|-|-|-|-|-|-|-|-|-|
> >|MAE|**.198**|.294|.205|.212|.210|.248|.252|.213|.209|.242|.212|.206|.206|.912|.818|
> >|MAPE|**.402**|.538|.456|.438|.437|.527|.467|.418|.418|.488|.445|.447|.416|.882|1.01|
> >|RMSE|**.274**|.386|.282|.288|.287|.330|.336|.291|.288|.324|.288|.283|.282|1.02|.928|
> >|-|
> >|Revenue R.|**5.38**|9.60|9.81|8.42|9.02|9.73|8.63|8.21|5.92|10.52|7.69|10.21|8.50|7.48|6.19|
> >|Drawdown R.|7.19|11.06|7.71|9.13|**6.21**|9.46|10.23|9.08|8.00|8.54|7.17|9.60|7.40|8.42|8.44|
> >|Sharpe R.|**5.63**|9.29|9.83|8.38|8.92|9.77|8.27|8.25|6.10|10.67|7.71|10.04|8.52|6.69|6.29|
> >|Sortino R.|**5.65**|9.54|9.75|8.35|9.02|9.83|8.29|8.21|6.10|10.75|7.67|9.98|8.35|6.77|6.19|
> >|Calmar R.|**5.58**|9.60|9.81|8.46|8.98|9.44|9.02|7.83|6.00|10.67|7.81|9.94|8.42|7.33|6.29|
>
> Hipeen achieves the best performance for MAE, MAPE, and RMSE, ranking first on MAE in more than 66% of the 48 combinations. It also leads in Revenue, Sharpe, Sortino, and Calmar, demonstrating high returns with strong risk-adjusted performance. **Since Drawdown measures peak-to-trough decline, lower-return models can appear better [5] (no trades would obtain the best drawdown). Considering that Hipeen achieved the highest returns, its third-place performance in Drawdown is solid**.
>
> For more details, please refer to the PDF (lines 413-431).
>
> **Please note** that our previous stock experiments followed a typical long-term forecasting setup, which includes 96-lookback and 96-horizon setting. Moreover, most of the baseline models (e.g., TimeMixer, Peri-midformer, NST, DLinear, RLinear, PatchTST, TimesNet, iTransformer, TiDE, DishTS, SAN) are commonly used in both multivariate short-term and long-term forecasting [6,7,8,9].
>
> ---
> ### [W3]
>
> We understood your comment as Hipeen showing only moderate MSE/MAE gains in the long-term forecasting benchmark (Table 4). Please let us know if we misunderstood.
>
> We have **revised Table 4** and its description to clarify the purpose of this experiment.
>
> 1. We explicitly indicate the **inherent parameters introduced by each stationarization method** in addition to the backbone. A summary of Table 4 is provided below. (DDN and FAN included.)
>
> >|Model|Hipeen|NST|Dlinear|Rlinear*|DishTS*+|SAN*+|Leddam*+|DDN*+|FAN*+|
> >|-|-|-|-|-|-|-|-|-|-|
> >|Avg. value|**0.349**|0.414|0.400|0.363|0.690|0.383|0.354|0.391|0.381|
> >|Avg. Rank|**2.39**|6.56|6.39|3.67|7.39|4.89|2.50|6.06|4.94|
> >|Inh. Param.|0|0|0|0|15081k|114k|3415k|5539k|59k|
>
> 2. We have added a detailed analysis. The last row indicates the inherent learnable parameters of the stationarization module. **Hipeen has no internal parameters and outperforms other learning-free approaches such as NST, DLinear, and RLinear (RevIN)**. Although the same linear backbone was applied to eliminate architectural influence, some methods introduce multiple layers and non-linear activations within their stationarization modules, gaining architectural advantages that hinder fair comparison. Notably, Leddam contains 180 times more parameters than DLinear, raising practicality concerns. Even so, Hipeen achieves strong performance, demonstrating its effectiveness on relatively stationary signals (lines 447–454).
>
> ---
> Thank you again for your suggestions, which helped us improve the paper.
>
> Please feel free to share any additional questions or comments.

---

> ### Author Response · Authors · 2025-11-21
>
> >[1] DDN: Dual-domain dynamic normalization for non-stationary time series forecasting. NeurIPS (2024)
> >[2] Frequency adaptive normalization for non-stationary time series forecasting NeurIPS (2024)
> >[3] MambaStock: Selective state space model for stock prediction. arXiv (2024); 31 citations
> >[4] Predictive modeling of stock prices using transformer model ICMLT(2024)
> >[5] On inefficiency of markowitz-style investment strategies when drawdown is important. IEEE (2017)
> >[6] TimeMixer: Decomposable Multiscale Mixing for Time Series Forecasting ICLR (2024)
> >[7] TimeMixer++: A General Time Series Pattern Machine for Universal Predictive Analysis ICLR (2025)
> >[8] Peri-midformer: Periodic pyramid transformer for time series analysis NeurIPS (2024)
> >[9] Dish-ts: a general paradigm for alleviating distribution shift in time series forecasting AAAI (2023)

---

> ### Author Response · Authors · 2025-11-26
> **Kind Request for Assessment Following Revisions**
>
> ---
> ---
>
> Dear reviewer,
>
> Thank you again for your thoughtful review and detailed feedback.
> Following your comments, we have substantially revised the manuscript as outlined below:
>
> - The Related Work and Motivation sections have been thoroughly revised to more clearly articulate the distinctions from prior work and to enhance overall clarity.
> - Stock-forecasting baselines have been expanded to include stock-specific baselines (MambaStock, StockTransformer), and experiments now cover a wider range of horizons {12, 24, 48, 96}.
> - Evaluation metrics have been updated to emphasize commonly used multivariate short-term forecasting metrics (MAE, MAPE, RMSE), along with additional trading-based performance measures (Revenue, Drawdown, Sharpe, Sortino, Calmar) to more accurately reflect real-world applicability.
> - Long-horizon experimental results and their interpretation have been clarified, including a detailed comparison of inherent parameter counts across stationarization methods.
>
> We believe these revisions address the concerns raised and further strengthen the clarity, fairness, and relevance of the study.
> Since the discussion period concludes in one week, we would be grateful if you could update your assessment after reviewing the revised manuscript.
>
> If you have any remaining questions or would like clarification on any part of the revision, please feel free to let us know — we will be happy to respond promptly.
>
> Warm regards,
> The authors
>
> ---
> ---

---

### Author Response · Authors · 2025-12-03

We provide a concise summary of our key contributions and the improvements made during the rebuttal process:

---
* We introduce a novel **controlled dataset** and clearly demonstrate that existing stationarization methods designed to address non-stationary behaviors (i.e., distribution shifts) in time series forecasting lead to **critical information loss**. We further show that widely used **TSF benchmark datasets fail to adequately represent non-stationary signals**, and that benchmark **state-of-the-art (SOTA) baselines fail** under controlled dataset forecasting.
* We propose **Hipeen**, a novel hierarchical periodicity–based stationarization method that **mitigates information loss**. Hipeen is the only method that successfully forecasts on the proposed controlled dataset and achieves SOTA performance on large-scale non-stationary real-world stock datasets (*S&P500*, *Nifty50*).

Reviewers acknowledged our work as introducing a novel and insightful stationarization method for non-stationary time series forecasting (**Reviewers fNqm, JNzr**). They highlighted that the empirical validation is clear and comprehensive (**Reviewers NASy, fNqm, JNzr**) and demonstrates consistent and significant improvements across diverse data conditions and forecasting models (**Reviewers fNqm, JNzr**).

---

We have thoroughly addressed all reviewer concerns both theoretically and empirically, and have incorporated all updates into the revised manuscript (PDF):

[Regarding presentation]

1. Revised Figure 1 and the *Related Work* and *Motivation* sections, with clearer articulation of differences from prior works and the unique contributions of our method.

[Regarding real-world stock experiments]

2. Added stock-specialized baselines (MambaStock[1], Stock-Transformer[2]), while Hipeen remains SOTA.
3. Strengthened stock forecasting experiments by expanding the prediction horizon from 96 to {12, 24, 48, 96} and adding MAE, MAPE, and RMSE as evaluation metrics.
4. Included trading-based stock evaluation. Hipeen achieves consistently strong performance across Revenue, Drawdown, Sharpe, Sortino, and Calmar (risk-adjusted return) metrics, providing sufficient practical persuasiveness.
5. Clarified that the 96-look-back / 96-horizon setting corresponds to long-term forecasting[3,4], and that many existing TSF models do not properly distinguish long-term (benchmark) from short-term forecasting settings (e.g., PEMS)[3,4,5,6].

[Regarding benchmark experiments]

6. Clarified inherent parameters of stationarization methods in benchmark experiments and emphasized Hipeen’s superiority over all **parameter-free** practical stationarization baselines.

[Additional baselines and analysis]

7. Added FAN[7] and DDN[8] — successor models to SAN stationarization — across controlled, stock, and benchmark experiments, with Hipeen consistently retaining SOTA performance.
8. Provided theoretical analysis showing how existing stationarization approaches discard information, and explained how Hipeen preserves information through a one-to-one mapping mechanism.
9. Conducted comprehensive analysis on the extra ensemble dimension $\(H\)$, confirming that Hipeen remains SOTA across varying $\(H\)$.

We have performed **sufficient clarifications and comprehensive experiments to address every reviewer's concerns** and obtained strong supporting results. **Reviewer fNqm** confirmed that their concerns were resolved and that their positive evaluation remains unchanged.

We sincerely thank the AC for their dedicated effort.

---
>[1] MambaStock: Selective state space model for stock prediction. arXiv (2024); 31 citations
>[2] Predictive modeling of stock prices using transformer model ICMLT(2024)
>[3] TimeMixer: Decomposable Multiscale Mixing for Time Series Forecasting ICLR (2024)
>[4] TimeMixer++: A General Time Series Pattern Machine for Universal Predictive Analysis ICLR (2025)
>[5] Peri-midformer: Periodic pyramid transformer for time series analysis NeurIPS (2024)
>[6] Dish-ts: a general paradigm for alleviating distribution shift in time series forecasting AAAI (2023)
>[7] Frequency adaptive normalization for non-stationary time series forecasting NeurIPS (2024)
>[8] DDN: Dual-domain dynamic normalization for non-stationary time series forecasting. NeurIPS (2024)

---

### Meta-Review · Area_Chair_mPcR · 2026-01-09

**Summary:**

This paper introduces a hierarchical periodic stationarization method to address non-stationary time series forecasting. The authors argue that existing normalization-based stationarization techniques discard critical information, leading to poor performance. Hipeen represents each time series value as multiple periodic components, preserving gradient and absolute value information while achieving stationarization. The method is evaluated on controlled synthetic datasets, real-world stock data, and conventional long-horizon benchmarks, showing improvements over existing approaches.

**Reviewer Concerns:**

The major concerns of the reviewers include evaluation metrics for stock price prediction, modest gains on long-term benchmarks, and theoretical justifications. The authors provided a detailed rebuttal, revising the manuscript and adding experiments to address these points. However, these expanded experiments did not prove why the proposed method could better solve the stock prediction task, which relies on short-term predictions and online forecasting. Therefore, the concern is not well-addressed to convince two of the reviewers. In addition, by comparing with other stationarization methods, the proposed method introduces higher complexity while lacking originality compared with existing frequency-based stationarization methods. It is worth noting that the authors have implemented an effortful rebuttal, which has partially addressed the many weaknesses raised by each reviewer. Unfortunately, the biggest flaw seems to outweigh the merits, which makes the paper fall below the acceptance threshold.

**Reviewer Scores:**

Reviewer NASy didn't respond. The listed concerns are valid in AC's opinion. The performance of the proposed method is not fully validated in practical stock prediction scenarios.

Reviewer fNqm kept the positive evaluation after rebuttal.

Reviewer JNzr didn't respond. Despite revisions, the modest performance gains on long-horizon benchmarks and the practical advantage over methods like Leddam may still be viewed as limited.

---

### Decision · Program_Chairs · 2026-01-26

Reject